# Coral Lipids

**DOI:** 10.3390/md21100539

**Published:** 2023-10-15

**Authors:** Andrey B. Imbs, Valery M. Dembitsky

**Affiliations:** 1A.V. Zhirmunsky National Scientific Center of Marine Biology, 17 Palchevsky Str., 690041 Vladivostok, Russia; 2Centre for Applied Research, Innovation and Entrepreneurship, Lethbridge College, 3000 College Drive South, Lethbridge, AB T1K 1L6, Canada

**Keywords:** coral, zooxanthellae, lipids, fatty acids, oxilipins, chemotaxonomy, biosynthesis

## Abstract

Reef-building corals, recognized as cornerstone species in marine ecosystems, captivate with their unique duality as both symbiotic partners and autotrophic entities. Beyond their ecological prominence, these corals produce a diverse array of secondary metabolites, many of which are poised to revolutionize the domains of pharmacology and medicine. This exhaustive review delves deeply into the multifaceted world of coral-derived lipids, highlighting both ubiquitous and rare forms. Within this spectrum, we navigate through a myriad of fatty acids and their acyl derivatives, encompassing waxes, sterol esters, triacylglycerols, mono-akyl-diacylglycerols, and an array of polar lipids such as betaine lipids, glycolipids, sphingolipids, phospholipids, and phosphonolipids. We offer a comprehensive exploration of the intricate biochemical variety of these lipids, related fatty acids, prostaglandins, and both cyclic and acyclic oxilipins. Additionally, the review provides insights into the chemotaxonomy of these compounds, illuminating the fatty acid synthesis routes inherent in corals. Of particular interest is the symbiotic bond many coral species nurture with dinoflagellates from the Symbiodinium group; their lipid and fatty acid profiles are also detailed in this discourse. This exploration accentuates the vast potential and intricacy of coral lipids and underscores their profound relevance in scientific endeavors.


*We lost Professor Andrey B. Imbs, who dedicated his life to researching the lipids of corals found in the waters of Vietnam, the Great Barrier Reef in Australia, and the Bering Sea.*


## 1. Introduction

Corals, celebrated for their elaborate structures and ecological roles, belong to the marine invertebrate category, specifically the class Anthozoa within the phylum Cnidaria. Intriguingly, what we often recognize as a single coral unit is actually a collection of numerous identical individual polyps. Every polyp contributes to the formation of the coral’s framework, producing an exoskeleton at its base that can extend over several meters [1,2,3].

Delving into their taxonomy, reef-building or hard corals are identified by their robust calcareous exoskeletons and are part of the subclass Hexacorallia. Conversely, soft corals, devoid of this firm exoskeleton, belong to the subclass Octocorallia. Here, we observe a distinction between alcyonarians and gorgonians, the latter often referred to as “horny coral”. Augmenting the coral mosaic are the hydrocorals, frequently seen enhancing the reef settings. They are particularly associated with the genus Millepora and are members of the class Hydrozoa. A defining feature of many hard corals, and a substantial number of soft corals, is their intricate bond with endocellular symbiotic dinoflagellates, primarily those of the Symbiodinium group, commonly known as zooxanthellae [4,5,6,7].

The world of corals, along with its varied symbionts, is a rich repository of biologically active secondary metabolites. These compounds, encompassing fatty acids, steroids, triterpenoids, distinct lipids, alkaloids, and low-molecular-weight terpenoids, have captivated the scientific community [8,9,10,11,12,13,14,15,16,17,18,19,20,21,22,23,24]. While it is recognized that corals host an array of symbionts, including microalgae, fungal endophytes, bacteria, and other microorganisms, there remains uncertainty regarding the precise origin of certain compounds—be it the corals or their microbial allies [25,26,27,28,29]. Until this mystery is unraveled, the prevailing hypothesis holds that all metabolites derived from corals are the results of the corals’ own biochemical processes [8,9,10,11,12,13,14,15,16,19,20,21,22,23].

In this review, we set forth a comprehensive exploration, assimilating the abundant knowledge on lipids, polar lipids, fatty acids, and other lipophilic compounds found in corals and selected symbionts.

## 2. The Content of Total Lipids in Corals

The quantity of total lipids serves as a primary benchmark when assessing the chemical composition of a biological specimen. Typically, total lipids are extracted from biological tissues using a blend of chloroform and methanol, often employing variations of the Folch method [30]. When extracting from coral tissues, other hydrophobic substances such as HC and ST also migrate into the total lipids fraction. While the ST composition of corals is not addressed in this book, the proportions of HC and ST are significant within the total lipids fraction. Thus, both HC and ST are taken into account when examining the content and makeup of total lipids and in the chemotaxonomy of corals.

Corals predominantly form colonies. The majority of a coral’s total lipids reside within its polyps (corallites). These polyps are embedded within the surface layer of a substantial base composed largely of inorganic substances that possess minimal lipid content. The ratio between the mass of polyps and their supporting structure varies, and is largely influenced by the size of the colony gathered for analysis. This variability can pose challenges when determining total lipid content, leading to data inconsistencies. In some instances, the percentage of total lipids has been expressed in relation to the entire colony mass. For hard corals and alcyonarians, lipid content is most commonly determined in desiccated tissue post the removal of the carbonate exoskeleton and spicules.

The soft tissues of hard corals are lipid rich. Hosai [11] discovered that the lipid content in *Fungia actiniformis* amounts to 25.7% of the dry weight (DW) of coral tissues. For *Pocillipora capitata*, the lipid content approximated 35% of DW, translating to 2.9 mg/g of coral tissue [31]. Meyers [32] assessed the fatty acid (FA) content, which corresponds to lipid content, in the soft tissues of 45 samples spanning 27 Caribbean scleractinian species from 10 different genera (Appendix A). Due to the intricate procedure for FA extraction, his data variation extended from 0.4 to 324 mg FA per 1 g DW. Based on Patton’s research, which analyzed 37 hard coral species, 1 g of a coral colony ranged from containing 1.3 mg (in *Favia stelligera*) to 45.2 mg (in *Goniopora gracilis*) of total lipids (as shown in Appendix A) [33].

Given the voluminous data, Appendix A, are provided in Appendix A. The calculation method deployed showed that the percentage of total lipids in coral colonies with larger corallites, such as *G. gracilis*, *Tubastraea coccinea*, and *Dendrophyllia* c.f. *micranthus*, was noticeably greater than that in colonies with smaller corallites (refer to Appendix A). Lipid content in *Porites porites*, *Montastrea annularis*, and *Siderastrea siderea*, sourced from near Barbados Island, regstered at 8–11%, 24–31%, and 25–34% DW, respectively [34]. Meanwhile, the lipid ratio in corals such as *Pocillopora verrucosa*, *Stylophora pistillata*, and *Goniastrea retiformis*, gathered from the Red Sea, was somewhat diminished (ranging between 11 and 17%), as illustrated in Table 1 [35].

According to data from Yamashiro and his colleagues [36], the lipid content in 12 scleractinian species, gathered near Okinawa Island in December, had an average of 24.2% (DW). This ranged from 14% in *Pocillopora verrucosa* to 37% in *Galaxea fascicularis* (as referenced in Table 2). An examination of the chemical composition of hermatypic corals, namely *Stylophora pistillata*, *Lobophyllia corymbosa*, and *Echinopora gemmacea*, from the Red Sea, revealed lipid contents of 1.90 mg/g, 8.58 mg/g, and 1.32 mg/g of coral DW, respectively [37]. The lipid content in undisturbed colonies of the hard coral *Stylophora subseriata* was between 8.9 and 12.8% of DW, while zooxanthellae extracted from these colonies exhibited a lipid content between 11.7 and 16.0% of DW [38].

The lipid content for *Astrangia danae*, *Montastrea annularis*, *Pocillopora damicornis*, and *Acropora formosa* was measured at 3.0, 1.8, 0.17, and 2.28 mg/cm^2^ of surface area, respectively [39,40,41,42]. Some species of the genus Acropora are shown in Figure 1, stony corals from the family Dendrophyllidae are shown in Figure 2, and stony corals from the Faviidae family are shown in Figure 3. Soft corals have been more extensively studied for their lipid content compared to hard corals. Based on Patton’s research [33], 1 g of the soft coral colony *Tubipora musica* had 83 mg of total lipids, as observed in Australia (see Appendix A). For Caribbean gorgonians *Eunecia tourneforte* and *Plexaura homomalla*, lipid content represented 5.2% and 22.0% of dry weight, respectively [43]. Meanwhile, in Vietnamese gorgonians *Bebryce indica* and *Mopsella aurantia*, lipid percentages stood at 1.24% and 3.80% of DW, respectively [44]. Illustrations 1, 2, and 3 depict various corals from the South Pacific Ocean and the Great Barrier Reef in Australia (Figure 1, Figure 2 and Figure 3).

In *Gorgonia mariae* and *G. ventalina* from Puerto Rico, the lipid proportions were up to 9.1% and 3.2% of wet weight (WW) of coral tissue, respectively [45]. On average, eight gorgonian species from Vietnam contained 8.2 mg/g of lipids, while four alcyonarian species had 16.8 mg/g WW (refer to Table 3) [46]. The lipid content in *Lobophytum crassum* from Okinawa Island reached 29.7% of dry weight (DW) [36]. Antarctic shallow-water soft corals, such as *Alcyonium paessleri*, *Clavularia frankliniana*, and *Gersemia antarctica*, held between 5.2 and 12.6% lipids [47]. The cold-water alcyonarian, *Gersemia rubiformis*, collected from Avachinsky Bay in the Bering Sea at depths of 4–18 m in August, had a total lipid content of 2.2% WW [48]. A review focused on the lipids of marine and estuarine invertebrates highlighted that gorgonians typically have a higher lipid content compared to Hydrozoa and Scyphozoa [43].

### 2.1. Variation of the Total Lipid Content According to Environmental Factors

The total lipid content in corals fluctuates based on various factors, including the stage of the reproduction cycle, season, habitat depth, light exposure, and other environmental conditions. A primary reason for the lipid levels’ sensitivity to light intensity—which can change throughout the year and with depth—is its impact on the lipid biosynthesis in symbiotic microalgae called zooxanthellae. These microalgae are present in the cells of most coral species. A decrease in light reduces lipid synthesis in the zooxanthellae, which in turn affects the transfer of lipids from the symbionts to the host coral and the rate at which these lipids are incorporated into the host [49].

Lipids serve as the primary energy reserve and source for corals. The amounts of lipids in coral tissues can vary based on the energy provided by the zooxanthellae and the energy expended by the coral during respiration, cellular renewal, and the release of reproductive material [50,51]. Stimson’s study on six species of Hawaiian hermatypic corals—including *Pocillopora meandrina*, *Pocillopora damicornis*, *Cyphastrea ocellina*, *Montipora verrucosa*, *Porites compressa*, and *Porites lobata*—revealed that the lipid content of individual samples from a single species collected in different seasons, and among different species collected within the same season, showed significant differences (*p* < 0.005) [52]. For instance, the lipid content in the “Y” type of *Pocillopora damicornis* exhibited a cyclical variation (±5% of DW) over a lunar month (see Table 4). In this species the lowest observed lipid content was 21% of the tissue dry weight, while the highest reached 58% [52].

The lipid content in deep-water cnidarians—including three species of alcyonarians, five species of gorgonians, Antipatharia, and two species of sea pens—varied between 2.4 and 38.8%, with an average of 12.2 ± 7.7% [53]. When comparing lipid percentages across these groups, the order of increase was gorgonians < alcyonarians/sea pens < Antipatharia. Interestingly, the lipid content of these deep-water species was not vastly different from that of their shallow-water counterparts, whose lipid content ranged from 6 to 47% [35,54]. It is noteworthy that extreme depths of habitat (exceeding 400 m) and the lack of phototrophic food sources seemed to have little influence on the coral’s lipid content [53]. However, more recent and detailed studies have indicated that the balance between storage and structural lipid fractions in corals such as *Seriatopora hystrix* and *Pachyseris speciosa* (found between 3 and 60 m depths in Scott Reef of Northwest Australia in the Indian Ocean) fluctuates with depth and is influenced by the type of symbiont [55].

In *Montastrea annularis* from Barbados Island, consistent lipid levels (24–31% DW) were observed across a depth range of 3 to 30 m [34]. Interestingly, *S. siderea* exhibited higher lipid content (25–34% DW) in deeper waters. In contrast, the lipid content in *P. porites* declined with depth, registering at 9–12% DW (as illustrated in Table 5) [34]. Researchers postulate that these variations in lipid levels are influenced by each coral species’ unique nutritional strategy. The contribution from various food sources, including products from zooxanthellate photosynthesis, zooplankton, and dissolved organic substances, differs for each species. This results in distinct trends in lipid content corresponding to habitat depth (Table 6).

The hard coral *Goniastrea aspera*, sampled monthly over a year near Okinawa Island, showed variations in its total lipid content (% DW). The lipid levels were lowest in the winter months (December–January at 21–25%) and peaked during the summer (June–September, 35–42%). A strong positive correlation (r = 0.9) was identified between the lipid level, water temperature, and light intensity [56]. The researchers suggest that the summer surge in lipid levels can be attributed to oocyte maturation and a rise in metabolic rate due to increased water temperatures. However, no notable shifts in this metric were observed post the summer spawning.

There has also been interest in the seasonal lipid dynamics of soft corals. For instance, the soft coral *Heteroxenia fuscescens* from the Red Sea, which contains zooxanthellae, exhibited an average lipid content of 11.0 ± 3.5% DW over a three-year monthly analysis [57]. *H. fuscescens* saw lipid content fluctuations ranging from 7% in winter to 20% in summer, correlating with variations in food availability and light levels. In contrast, the soft coral species *Corallium rubrum* did not display any discernible season-based lipid content variations [58]. However, lipid content disparities were observed between female and male specimens of *Paramuricea clavata* during the spring season.

In populations of the hard coral *Pocillopora damicornis* from Rottnest Island, Western Australia, two distinct organismal groups were identified [51]. Although they shared similar morphological features, they exhibited different reproductive strategies. One group, which produced both oocytes and sperm during gametogenesis and subsequently released larvae, had a higher lipid content (about 43% DW) in January–February compared to the other group, which only produced sperm (36–40% DW). By March, right before spawning, lipid levels in both groups were roughly equivalent, ranging between 38 and 39%. Over the subsequent months, this percentage gradually decreased, averaging 33% and reaching its lowest point in April. The authors attributed these fluctuations to variances in the ratio of reserve to structural lipids in different *P. damicornis* groups before and after spawning. However, the initial data outlining lipid composition by class was not provided.

Though the study spanned only half a year, its findings concerning the lowest lipid content in the spring for both *P. damicornis* and *H. fuscescens* align well with other research [57]. The drop in *P. damicornis* lipid content during this season is likely influenced more by seasonal changes than by spawning events.

Harsh environmental conditions or diseases can significantly affect the lipid content in corals. Coral reef bleaching, which involves the loss of zooxanthellae due to abnormal spikes in water temperature, as experienced at Shikoku Island, Japan in 1998, resulted in a pronounced decline in lipid content within hard corals [59].

In healthy colonies of seven coral species—*Stylophora pistillata*, *Porites cylindrica*, *Montipora aequituberculata*, *Goniastrea aspera*, *Fungia fungites*, *Montipora digitata*, and *Montipora informis*—the lipid content ranged from 19% to 33% of the dry weight of decalcified tissues. However, when these same species underwent bleaching, their lipid content notably dropped, with levels ranging from 3% to 17%. Only one coral species, *Galaxea fascicularis*, exhibited a minor decline in lipid content, moving from 37% to 32%.

The extent of lipid reduction seemed to be influenced by the colony morphology. Generally, species with bulkier colony forms retained more lipids and rebounded faster post-bleaching compared to species with branching colony structures. A positive correlation was noted (r = 0.674, *p* < 0.01) between the lipid content and zooxanthellae density in bleached corals. This relationship underscores the critical role of zooxanthellae in providing lipids to the host, aiding in both the survival and recovery of affected colonies [59].

Colonies of the symbiotic gorgonian *Eunicella singularis* were subjected to four different nutritional diets at 18 °C for over two months: autotrophy alone, autotrophy with inorganic nitrogen addition, autotrophy coupled with heterotrophy, and heterotrophy on its own [60]. Unlike many other anthozoans, adding inorganic nitrogen or food (heterotrophy) to autotrophy had no impact on lipid content. In every scenario, a temperature rise from 18 to 26 °C resulted in a reduction in lipid content.

The hard coral *Montipora informis*, located in Shikoku Island, Japan, displayed hemispherical protrusions, or tumors. These tumors had a lipid content of 10.6% DW of tissues, which was considerably lower (*p* < 0.05) than that of healthy coral tissues, which stood at 32.2% [61]. Colonies of the Australian scleractinian coral, *Pocillopora damicornis*, were artificially fragmented, yet no significant changes in lipid content (ranging from 29–46%) were observed after this intervention [50].

When two species of hard corals, *Porites cylindrica* and *Stylophora pistillata*, collected from the Great Barrier Reef, were exposed to heat stress at 32 °C, interesting results emerged. Over a 10-day period, *P. cylindrica* lost half of its zooxanthellae, while *S. pistillata* lost nearly all of its zooxanthellae. However, no notable shifts in lipid content were recorded for either species [62]. This contrasts with findings from another study where the soft coral *Sinularia capitalis* and the hard corals *Montipora digitata* and *Acropora intermedia* underwent bleaching at 33 °C. In this experiment, the corals lost up to 95% of their zooxanthellae. Furthermore, the total lipid content in *S. capitalis* dropped by 3.2 times, and in both *M. digitata* and *A. intermedia* it decreased by 2.7 times (Figure 4) [63].

A mathematical model was developed to determine the death probability of hard coral colonies, specifically *Acropora intermedia*, based on the lipid content in their tissues [64]. The findings suggested that the probability of colony death remains stable as long as the lipid content is above 60% of its initial level. Yet, there is a rapid increase in death probability as the lipid content drops further [65]. The speed of water temperature increase was identified as a key factor affecting lipid decline in corals [42]. At 30 °C, both slow (0.5°/day) and rapid (1.0°/day) heating led to the retention of about 60% lipids compared to control populations. However, by 33 °C, there was a pronounced difference between the two heating rates, with the rapidly heated corals exhibiting a more pronounced decrease in lipid concentrations.

In another study, a marked reduction in the total lipid content was observed in the hard coral *Acropora millepora* over 26 days, after the coral colonies were relocated to a shaded environment with filtered seawater [66]. Alongside this lipid reduction, there was a decrease in the expression of the *Dgat1* gene, which facilitates the formation of TG, and an increase in the expression of the *Tgl* gene, which performs the opposite function by releasing FA from the TG storage form. These data provide more detailed insights into the processes by which storage classes of coral lipids, such as TG, are mobilized during bleaching and stressful conditions.

#### 2.1.1. Composition of Total Lipids

Typically, the composition of total lipids is ascertained using the one-dimensional thin-layer chromatography (TLC) method, which is subsequently paired with a flame-ionization detector (“Yatroscan”) or densitometry for further analysis [59,67,68,69,70]. In the latter approach, a plate undergoes development in a chromatographic system designed for non-polar lipid separation (e.g., a mixture of hexane–diethyl ether–acetic acid in a 70:30:1 ratio). Once this step is completed, the plate is dried and then re-developed to ¼ of its length using a system intended for polar lipid separation (such as chloroform–methanol–25% ammonia in a 65:35:5 ratio) (Figure 5). By re-developing the plate, the width of the polar lipid band is expanded. This step ensures that the chromatogram does not become overly dense, making subsequent quantitative assessments more precise. However, it is important to note that achieving a complete separation of polar lipids into individual classes using this method is not feasible. As a result, all bands corresponding to polar lipids are aggregated. This grouping also includes glycolipids.

One-dimensional TLC chromatograms, developed in a single step using the solvent system CH_3_Cl-CH_3_OH-H_2_O (in a ratio of 65:25:4, *v*/*v*/*v*), showcased the total lipids from zooxanthellae. These lipids included wax esters (WEs), triacylglycerols (TGs), and various polar lipids. Chen et al. [28] presented these images in their study exploring the influence of the host gastrodermal membranes on the photosynthesis of zooxanthellae in the stony coral *Euphyllia glabrescens*.

In specific instances, a rapid lipid fractionation is executed using the low-pressure liquid chromatography method on silica gel columns. By sequentially eluting the column with chloroform, acetone, and methanol, distinct fractions of non-polar lipids, glycolipids, and polar lipids are isolated. The lipid concentration in every fraction is then quantified using the gravimetric method.

Total lipids of corals comprise a variety of compounds, including hydrocarbons (HCs), wax esters (WEs), sterol esters (SEs), monoalkyldiacylglycerols (MADAGs), triacylglycerols (TGs), free fatty acids (FFAs), sterols (STs), and polar lipids (PLs). The non-polar lipids are particularly prevalent, accounting for 59 to 83% of the cumulative lipid content [36,53,68] (as shown in Table 7, Table 8 and Appendix A).

In young propagules of Scleractinia corals, just as in mature colonies, lipids constitute the primary component, representing 34 to 85.5% of their biomass [71,72,73,74,75]. Notably, coral eggs are rich in wax esters (WEs), believed to serve as an energy reserve facilitating long-distance dispersal. A study examined the variations in lipid and fatty acid (FA) compositions in *Goniastrea retiformis*, tracing these changes from the egg stage to larvae aged 30 days [76]. The lipid composition of the eggs comprised 86.3% WEs, 9.3% polar lipids (PLs), 4.1% sterols (STs), and a mere 0.3% triacylglycerols (TGs). Over time, the WE content showed a significant decrease, while the PL, ST, and TG levels remained relatively stable. Predominant fatty acids found in *G. retiformis* eggs included 16:0, 16:1n-7, 18:1n-9, 18:2n-6, 18:3n-6, 20:4n-6, and 22:5n-3. In more recent studies, the lipid content and composition of oocytes from five species of Scleractinia and two gorgonians were examined [77,78]. The primary lipid classes identified in the oocytes were WEs, phosphatidylethanolamine (PEs), and free fatty acids (FFAs). However, the unusually high percentage of FFAs (ranging from 19 to 53% of total lipids) suggests potential artificial lipid hydrolysis, which could result in unanticipated shifts in lipid class composition.

##### Hydrocarbons

Based on the standard lipid extraction process, the total lipids fraction includes hydrocarbons if they are present in the extractable tissue. Corals have a substantial quantity of hydrocarbons, which augment the weight of the total lipids fraction. Meyers [32] conducted one of the earliest systematic studies on the hydrocarbon composition of corals. He analyzed the content and composition of both unsaturated and saturated hydrocarbons in 18 species of Caribbean Scleractinia, as well as the hydrocoral *Millepora alcicornis* (see Appendix A). The saturated hydrocarbon composition in corals differed from that in terrestrial organisms where n-alkanes C27, C29, and C31 predominated. Corals contained C16–C30 hydrocarbons, which included both even and odd numbers of carbon atoms. There was a significant presence of C17 n-alkane and pristane. It is postulated that potential sources for these hydrocarbons might be phytoplankton and copepods [32].

The content of unsaturated hydrocarbons was notably higher than that of n-alkanes, reaching up to 20 mg/g of dry tissue in *Madracis decactis* (refer to Appendix A). However, the specific composition of these unsaturated hydrocarbons was not identified. It is worth highlighting the variability in the data, particularly when looking at multiple samples from a single coral species (as shown in Appendix A). The hydrocarbon composition data provided by Meyers [32] for the hydrocoral *Millepora alcicornis* notably differed from earlier findings for this coral [43]. Specifically, *Millepora* sp. primarily contained two major saturated alkanes: C18 (55.1%) and C27 (11.6%). The combined content of other saturated alkanes ranging from C16 to C30 was 7.6%, and unsaturated alkanes, branched alkanes, phytane, and pristane were not detected.

In Joseph’s review [43], it was observed that only unbranched saturated alkanes were identified in hard corals (refer to Figure 6). The alkane distribution in species such as *Favia* sp. and *Acropora palmata* displayed a unimodal pattern, peaking at the concentration for C25 alkane. For *Acropora cervicornis* and *Agaricia agaricites*, the predominant hydrocarbons were C21 and C19 alkanes, respectively, both showing a unimodal distribution centered around long-chain waxes. Meanwhile, the hydrocarbons in the hard coral species *Porites* sp. were dominated by C20 alkane, constituting approximately 95% of its content (see Figure 6). Additionally, the presence of aromatic hydrocarbons was observed in gorgonian corals [43].

Japanese researchers did not detect hydrocarbons in the total lipids of 13 species of hard corals, the soft coral *Lobophytum crassum*, and the hydrocoral *Millepora murrayi* [36,56]. This oversight might be attributed to the TLC method used to analyze total lipids. On a TLC plate, the hydrocarbons band could be obscured by the front of the non-polar chromatographic system, or it could merge with bands of waxes and sterol ethers. Additionally, detecting saturated hydrocarbons on a TLC plate requires more stringent conditions than those needed for unsaturated lipids.

In the soft coral *Sinularia* sp., several compounds were identified: 1-Ethenyl-1-methyl-2,4-di(methylethenyl)-cyclohexane (**1**, depicted in Figure 7), 7,11-dimethyl-3-methylen-1,6,10-dodecatriene (**2**), 2-isopropenile- 4a,8-dimethyl-1,2,3,4, 4a,5,6,7-octahydronaphthalene (**3**), 1,8a-dimethyl-7-(1-methylethenyl)-1,2,3,5,6,7, 8,8a-octahydronaphthalene (**4**), 4a-methyl-1-methylene-7- (1-methylethenyl)- decahydro-naphthalene (**5**), alkane C_20_H_42_, and tetradecylpalmitate. The relative concentrations of these compounds were 5.6%, 7.0%, 41.5%, 10.2%, 8.0%, 13.6%, and 7.0% of the total hydrocarbon fraction, respectively [79].

##### Waxes

The analysis of total lipid content from over 100 coral species from the coastal waters of Vietnam (in the South China Sea) revealed that waxes were among the primary lipid classes [68] (as shown in Appendix A and Figure 8). In hard corals from this region, the percentage of waxes in total lipids varied between 26.4% (in *Porites solida*) and 66.4% (in *Favia maxima*), with an average of 48 ± 11% of the total lipids. In comparison, soft corals had a lower wax content in their total lipids. The percentage ranged from 7.0% (in *Lemnalia capnelliformis*) to 55.5% (in *Cladiella laciniosa*), averaging 35 ± 11% of the total lipids.

In 12 species of hard corals from Okinawa Island, Japan, the wax content in total lipids varied between 9.1% (in *Tubastrea* sp.) and 31.4% (in *Goniastrea aspera*) [36] (as illustrated in Table 7). The average wax content for hard corals from Okinawa Island was 20.0 ± 7.0% of the total lipids, which is significantly lower than the average wax content in hard corals from Vietnam [68]. Similarly, the proportion of waxes in the total lipids of the soft coral *Lobophytum crassum*, collected near Okinawa Island, stood at 14.6% [36], which was also considerably lower than the average wax content in soft corals from Vietnam [68].

In the cold-water soft coral *Gersemia rubiformis*, sourced from the shallow waters (6–18 m) of Avachinsky Bay in the Bering Sea, waxes made up 29.5 ± 4.9% of the total lipids [48]. In contrast, the average wax content in the total lipids of alcyonarians, gorgonians, and Antipatharia (including species such as *Anthomastus grandiflorus*, *Gersemia rubiformis*, *Capnella florida*, *Acanella arbuscula*, *Paramuricea* spp., *Primnoa resedaeformis*, *Paragorgia arborea*, and *Bathypates* spp.) from the deeper waters of 150–400 m in the North Atlantic (covering Newfoundland and Labrador) was notably lower at 12.9 ± 2.0% [53]. This indicates a disparity in wax content between corals from the shallow waters of Vietnam and Okinawa Island’s coast and those from the deeper North Atlantic regions.

Among the main components identified in the non-polar lipids of the hydrocoral *Millepora* sp., waxes were prevalent. Specifically, the primary waxes were palmitoylstearate (**6**, as depicted in Figure 9) at 64%, palmitoylpalmitate (**7**) at 34%, and palmitoylmyristate (**8**) at 2% [80]. A subsequent GC-MS analysis conducted thirty-seven years later on the chemical composition of two hydrocoral species, *Millepora dichotoma* and *M. platyphylla* from the Red Sea, revealed the presence of waxes with empirical formulas C_30_H_60_O_2_, C_32_H_64_O_2_, C_34_H_68_O_2_, and C_36_H_72_O_2_ [37].

The C_30_H_60_O_2_ wax is believed to contain a mixture of two saturated isomers: 14:0/16:0 (with 14 carbon atoms in the alkyl portion and 16 carbon atoms in the acyl portion) and 12:0/18:0. The C_32_H_64_O_2_ composition likely encompasses just one isomer, 16:0/16:0 (7). For C_34_H_68_O_2_, the expected composition includes two isomers, 16:0/18:0 (6) and 18:0/16:0, while C_36_H_72_O_2_ likely contains only the isomer 18:0/18:0. However, the exact content of each isomer was not specified. The 18:0/16:0 wax has also been identified in the soft coral *Sinularia microclavata* from the South China Sea [81].

The primary component of waxes in hard corals, palmitoylpalmitate (**7**), varied in content, with levels ranging from 100% in *Flavia* sp. to 73% in *Isophyllia* sp. These corals also contained 3–24% of palmitoylmyristate and palmitoylstearate, and trace amounts of myristylmyristate and stearoylstearate [43]. In *Goniastrea retiformis*, the total lipids consisted of about 80% palmitoylpalmitate [43]. After saponifying waxes extracted from some gorgonian corals [43], the identified aliphatic alcohols comprised only C16, C18, and C18:1 alcohols, with the C14 alcohol notably absent (Table 9).

Patton and colleagues [33] reported that the predominant fatty acid (FA) in the waxes of hermatypic corals was 16:0, accounting for an average of 58% of the total. Other significant FAs included 18:0, 18:1, and 16:1 (Appendix A). For non-symbiotic hard corals such as *Tubastraea coccinea* and *Dendrophyllia* cf. *micranthus*, the primary FAs in their waxes were 18:1 (ranging from 58.2% to 62.2%) and 16:1 (ranging from 15.8% to 17.3%). The total FAs in the waxes of hard corals such as *Acropora echinata*, *Gardineroseris planulata*, *Leptoria phyrgia*, *Pocillipora damicornus*, and *Psammocora contiqua* included 41.0, 36.6, 60.2, 28.0, and 29.2% of C22 PUFA, respectively. The FA distribution in the waxes of the azooxanthellate octocoral *Tubipora musica* mirrored that of hermatypic corals. For the hydrocoral *Stylaster* sp., about 95% of its wax’s FAs were 18:1. Meanwhile, the wax FAs of the hydrocoral *Millipora exesa* contained 22% C22 PUFA (Appendix A).

The wax content in various hard corals such as *Porites porites* and *Montastrea annularis* (from the Caribbean Sea), as well as *Pocillopora verrucosa*, *Stylophora pistillata*, and *Goniastrea retiformis* (from the Red Sea), was measured at 27.8 ± 6.5%, 42.5 ± 12.5%, 22.3 ± 5.9%, 48.6 ± 14.1%, and 42.0 ± 6.4% of total lipids, respectively [35]. The Bay of Bengal’s soft coral, *Nephthea* sp., also contained waxes [82].

A comparative study by Yamashiro and colleagues [36] involving 12 hard coral species, one soft coral species, and one hydrocoral species found that, on average, waxes constituted 19% of total lipids (Table 7). Symbiotic corals such as *Fungia fungites* and *Goniastrea aspera* had more than 30% of their total lipids as waxes. In contrast, the non-zooxanthellate *Tubastrea* sp. contained a mere 9.1% of waxes. The wax FA composition is detailed in Appendix A [36]. In nearly all species studied, palmitic acid (16:0) was the primary acid in the waxes, with the exceptions being *Oulastrea crispata*, *Tubastrea* sp., and *Millepora murrayi*. In *Oulastrea crispata* and *Tubastrea* sp., acid 16:0 was the predominant saturated FA in the waxes, while in *Millepora murrayi*, acid 18:0 took this role. The waxes had a modest average polyunsaturated fatty acid (PUFA) content at 3.4 ± 1.1%. Elevated monoenoic FA (18:1n-9) levels were found in the waxes of *P. damicornis* (16.2%), *P. verrucosa* (16.0%), *Stylophora pistillata* (17.6%), *P. lutea* (22.6%), *O. crispata* (37.3%), and *Tubastrea* sp. (41.8%). Waxes had minimal long-chain FAs.

The aliphatic alcohols in the waxes included 14:0, 16:0, 16:1, 18:0, 18:1, and 20:0. The average proportion of alcohol 16:0 was 82.2 ± 5.2% of the total alcohols, with no polyunsaturated alcohols detected [36]. The hard coral *Montipora digitata* had a wax FA composition that included 14:0 (1.5%), 16:0 (62.4%), 16:1n-7 (3.4%), 18:0 (9.1%), 18:1n-9 (5.9%), 18:2n-6 (1.0%), 22:0 (4.2%), and 22:4 (2.0%). Additionally, the alcohols in these waxes were 14:0 (2.1%), 16:0 (90.1%), 16:1 (1.6%), 18:0 (2.8%), and 18:1 (2.6%) [83,84].

The absolute content of waxes in corals and their proportion in the total lipids varies seasonally. For instance, the proportion of waxes in the total lipids of the hard coral *Goniastrea aspera* (from Okinawa Island) fluctuated throughout the year, peaking at 36% in the summer—a 1.5-fold increase compared to winter values [56]. Observations of the hard coral *Montipora digitata* revealed a decline in the percentage of wax esters (WEs) in the total lipids, decreasing from 22.3 ± 0.9% to 13.9 ± 1.1% from the base of the coral branches to their tips [83,84].

Wax esters are believed to serve as an energy reserve for corals. It is posited that increased energy consumption during unfavorable conditions might reduce the wax content. A notable decrease in total lipid content in tumors (hemispherical outgrowths) of the Scleractinia coral *Montipora informis* (from Shikoku Island, Japan) primarily resulted from a loss in waxes. In these tumor tissues, the wax content was only 2.1%, a significant drop (*p* < 0.01) compared to the 30.3% found in healthy coral tissues [61]. A marked difference in the quantity of wax esters was observed between isolated colonies of the soft coral *Sarcophyton ehrenbergi* and those that had been in contact with *Pocillopora damicornis* colonies for a year [85]. This 20% discrepancy in wax ester content is attributed to varying energy expenditures arising from competition between the soft and hard corals.

Comparisons were drawn between the lipid compositions of healthy and bleached colonies of hard corals. Healthy colonies of *Porites compressa* and *Montipora verrucosa* had wax ester contents comprising 11.6–21.9% and 4.5–9.0% of the total lipids, respectively. In contrast, bleached colonies contained no wax esters [54]. During the onset of bleaching, the proportion of wax esters in the total lipids rapidly decreased from 25–30% to 5–15% in hard corals such as *Stylophora pistillata*, *Porites cylindrica*, *Montipora aequituberculata*, *Goniastrea aspera*, *Fungia fungites*, *Montipora digitata*, and *Montipora informis* (from Okinawa Island) [59]. However, there was a negligible difference in wax ester content between healthy and bleached colonies, with the exception of *Galaxea fascicularis* [59]. The minimal variation in total lipid content between healthy and bleached colonies of *G. fascicularis* suggests the early stages of bleaching for this species.

##### Monoalkyldiacylglycerols

Coral total lipids contain a significant quantity of unique lipids known as monoalkyldiacylglycerols (MADAGs). Compared to ester bonds, the simple ether bond in a MADAG molecule is resistant to both chemical and enzymatic hydrolysis. It is hypothesized that the presence of MADAG in the cell membrane enhances its resilience against the highly reactive lipolytic enzymes of cnidarians [43]. In corals that house zooxanthellae, MADAGs are only found in the total lipids of the pure polyp tissue fraction, suggesting that MADAG could serve as a lipid marker for the host in the symbiotic relationship of the coral [69]. During soft alkaline hydrolysis of the total lipids, MADAGs (as shown in Figure 10, compound **9**) transform into alkylglycerides (compound 10), which then accumulate in the unsaponifiable lipid fraction.

Yamashiro and colleagues [36] analyzed the lipid composition across 12 species of hard corals, one soft coral species, and one hydrocoral species. They found that the concentration of MADAG in the total lipids varied between 1.0 and 9.5% (refer to Table 7). The highest concentration of MADAG was observed in the soft coral, hydrocoral, and the hard azooxanthellate coral known as *Tubastrea* sp. Among hermatypic corals, species from the *Porites* genus had the highest MADAG concentration in their total lipids, reaching up to 7.5%. On average, soft corals had a higher MADAG content compared to hard corals. An analysis of 49 hard coral species from Vietnam showed an average MADAG concentration of 4.6 ± 2.9% in their total lipids. In contrast, 59 soft coral species from Vietnam had an average MADAG concentration of 14.7 ± 8.3% [68] (refer to Figure 4). *Millepora hydrocorals* had a notable MADAG content, of approximately 18%. The gorgonian coral *Paracis* cf. *horrida* exhibited the highest MADAG concentration at 51.9% of its total lipids [68]. The boreal soft coral *Gersemia rubiformis* from the Bering Sea had a MADAG content of 9.7 ± 2.8% [48]. The hard coral *Goniastrea aspera* from Okinawa Island displayed seasonal variation in MADAG content, with values ranging from 1–2% in the winter to 5–6% during summer [84]. The hard coral *Montipora digitata* showed a declining gradient of MADAG concentration from the base of its branches to the tips, decreasing from 4.6 ± 0.6% to 2.9 ± 0.2% [83]. Analyses of the unsaponifiable lipid fraction in various gorgonian species revealed that the primary alkylglycerols derived from MADAG were chimyl alcohol (1-hexadecylglycerol) (**10**) and batyl alcohol (1-octadecylglycerol) (**11**) (refer to Table 9). Trace amounts of selachyl alcohol (1-octadeca-9-enylglycerol) (**12**) were also detected [43]. In all studied gorgonian species, except for *Pterogorgia anceps*, the amount of batyl alcohol was either higher than or equal to the amount of chimyl alcohol [43]. Alkylglycerols were also identified in the soft coral *Nephthea* sp. from the Bay of Bengal [82].

The average MADAG content in the total lipids of cnidarians such as *Anthomastus grandiflorus*, *Gersemia rubiformis*, *Capnella florida*, *Acanella arbuscula*, *Paramuricea* spp., *Primnoa resedaeformis*, *Paragorgia arborea*, and *Bathypates* spp., all sourced from the North Atlantic waters (Newfoundland and Labrador) at depths of 150–400 m, was found to be 13.7%, 15.1%, 19.7%, 11.4%, 12.2%, 12.0%, 10.6%, and 15.4% of their total lipids, respectively [53]. Additionally, sea pens from the same region had a MADAG content making up 8.6% of their total lipids [53].

In the case of the soft coral *Sinularia* sp. from Nha Trang Bay (South China Sea, Vietnam), MADAGs constituted 25.2% of its total lipids. The unbranched saturated alcohols C16, C18, and C20 accounted for 22.8%, 77.0%, and 0.2% respectively, of the combined alcohol residues in these MADAGs. Furthermore, the fatty acids (FAs) present in these MADAGs were primarily composed of 58.7% saturated acids and 6.3% arachidonic acids. The linoleic acid content was relatively low, at 0.9% (refer to Appendix A) [79].

##### Triacylglycerols

The triacylglycerol (TG) content in the total lipids of hard corals such as *Porites porites* and *Montastrea annularis* from the Caribbean Sea, as well as *Pocillopora verrucosa*, *Stylophora pistillata*, and *Goniastrea retiformis* from the Red Sea, was 18.1 ± 5.0%, 22.5 ± 6.2%, 36.6 ± 13.9%, 24.6 ± 4.5%, and 22.6 ± 5.1%, respectively [35]. An examination of over 100 coral species from the coastal waters of Vietnam revealed that the average TG proportion in the total lipids was higher in Scleractinia and hydrocorals of the *Millepora* genus (23.4 ± 8.8%) compared to soft corals (9.2 ± 4.8%) [68] (see Appendix A).

There were no noticeable differences in TG content among families and genera of both hard and soft corals. In general, coral species hosting zooxanthellae had a higher TG content than those species without zooxanthellae [68]. The TG content in the total lipids of deep-water soft corals from the North Atlantic stood at 8.0 ± 2.7% [53] (refer to Table 8). Meanwhile, the total lipids of the cold-water soft coral *Gersemia rubiformis*, found at shallow depths in the Bering Sea, contained 6.7 ± 1.9% of TG [48].

A comprehensive comparison of fatty acid (FA) composition within the TG fraction was conducted for 40 species of Australian corals (Appendix A) [33]. Acid 16:0 was predominantly found in the TG of most species, with the exceptions being *Clavarina scrabicula* and *Lobophyllia corymbosa*, where acid 18:1n-9 was more prevalent. The FA composition of the TG fraction in *Echinophyllia* sp. stood out from other symbiotic species due to its elevated levels of acid 18:2 + 18:3, 20:U, and 22:U. Three out of the eight species with the highest 18:0 acid content were devoid of zooxanthellae. Similarly, two out of the six species with the most significant 18:1 acid content and three out of the five species with the highest C22 PUFA content were non-symbiotic corals. The FA composition of the TG fraction in two non-symbiotic species from the Dendrophyllidae family, *Tubastraea coccinea* and *Dendrophyllia* c.f. *micranthus*, was relatively similar. However, they significantly differed from two other symbiotic species from the same family, *Turbinaria* c.f. *frondens* and *Turbinaria* c.f. *sinensis*.

The total lipids from 12 scleractinian species gathered around Okinawa Island ranged between 14.9% and 30.4% in triacylglycerol (TG) content (see Appendix A) [36]. In comparison, the TG content in the hydrocoral *Millepora murrayi* from the same region was somewhat lower at 13.9%, while the soft coral *Lobophytum crassum* recorded the lowest TG content of 8.9% [36]. The fatty acid (FA) composition of the TG fraction from these cnidarians predominantly consisted of saturated and monoenoic FAs, especially 16:0 and 18:1n-9 (refer to Appendix A). The proportion of polyunsaturated fatty acids (PUFAs) was relatively low, averaging 7.2 ± 1.6%. Among the hard corals, the highest content of C20–22 PUFA was noted, whereas the FA from the TG of *M. murrayi* was characterized by the highest proportion of stearic acid (32.7%) [36].

The soft coral *Sinularia* sp. from Vietnam had total lipids that comprised 17.2% of TG [79]. The FA composition of its TG fraction was dominated by saturated and monoenoic acids, notably 16:0 and 18:1n-9. Linoleic acid, accounting for 11.5%, was the principal acid among PUFAs (see Appendix A). TGs serve as one of the primary reserve lipid classes in corals. It is believed that the TG content in total lipids varies significantly based on the coral’s diet, reproduction cycle stages, and environmental factors such as lighting and water temperature. For instance, the TG percentage in the total lipids of the hard coral *Goniastrea aspera* from Okinawa Island fluctuated between 10% during winter and 16% in summer [56]. Disease in the Scleractinia *Montipora informis*, located in Shikoku Island, Okinawa, Japan, resulted in a drop in the TG level within the total lipids from 7.8% to 2.6% [61]. A study aimed at assessing the coral’s reaction to natural heat stress was conducted on *Porites* sp. from the Gilbert Islands [86]. In this species, between the two main lipid energy sources, TG is metabolized more efficiently than wax esters (WEs) and is consumed more rapidly by corals during stressful conditions.

##### Polar Lipids

The majority of coral polar lipids consist of phospholipids. Components such as DPG (**13**, see structure in Figure 11), CAEP (**14**), LPS (**15**), LPC (**16**), LPE (**17**), PI (**18**), PA (**19**), PS (**20**), PC (**21**), PE (**22**), plasmalogen PE (**23**) and cerebroside (CE) have been identified in the polar lipid composition of Vietnamese corals [87,88]. Typically, the phospholipid quantity is derived from the content of inorganic phosphorus. As a result, the content of each phospholipid class is expressed as a percentage of the total phospholipids, rather than as a percentage of the total polar lipid content. The analysis of polar lipids from Vietnamese corals was conducted using two-dimensional TLC on silica gel, as depicted in Figure 12.

The phospholipid composition of gorgonian corals from Vietnam was studied by Svetashev (as shown in Table 10) [87]. Phospholipid content varied, accounting for 14 to 33% of the total lipid content. In addition to PE, PS, and PC, gorgonians also had a significant amount of CAEP (14, structure detailed later), which is characteristic of coelenterates. Even with fresh material extraction, there were noticeable amounts of LPC and LPS in the lipid extracts. The gorgonians studied showed substantial variation in PE content. For instance, the phospholipids of *Bebryce indica* contained only 5.4% PE, with a complete absence of LPE. Alongside the typical phospholipids, unidentified highly polar phospholipids (labeled X1-X3) were also observed (refer to Table 10).

Latyshev and his team [88] analyzed the composition and seasonal variations of the phospholipid profiles in 22 alcyonarian species from Vietnam’s shallow waters. These species belong to three genera: *Sinularia*, *Lobophytum*, and *Sarcophyton*. Identified within these soft corals’ phospholipid compositions were PC, PE, PS, their lyso-derivatives, CAEP, PI, DPG, and PA (as detailed in Table 11, Table 12 and Table 13). Notably, certain phospholipids were found in plasmalogen form (**23**) with PC ranging from 15 to 28%, PE from 58 to 94%, and PS from 21 to 47%. Lipids typical for zooxanthellae, such as phosphatidylglycerol and sulfoquinovosyl-diacylglycerol, were not present.

For *Sinularia* genus corals, there was a marked decrease in PC, PE, and DPG levels during the winter months, accompanied by a rise in the content of lyso-derivatives and CAEP. Seasonal fluctuations in diacyl-form phospholipids were less pronounced in *Lobophytum* genus corals (as seen in Table 12). Notably, phospholipids in plasmalogen form were virtually absent in samples taken during the winter [88].

PE, PC, and PS were the primary phospholipids in the Caribbean gorgonian corals including species such as *Pseudopterogorgia acerosa*, *P. americana*, and others [45,89,90]. In the soft coral *Sinularia* sp. from Vietnamese waters, 13.6% of the total lipids comprised polar lipids. These included 30.7% PG, 30.6% CAEP, and others [79]. The phospholipid composition of this *Sinularia* species was notably different from previous findings for Vietnamese *Sinularia* [88]. PG, commonly found in photosynthesizing endosymbiotic microalgae (zooxanthellae), could account for up to 30% of the lipid content of an entire coral colony. Kostetsky’s research [91] on corals from the tropical Pacific Ocean revealed a variety of phospholipids, with the dominant ones being PC, PE, and PS. The proportion of CAEP was roughly equal to PE, representing 24.2% and 22.7% of total phospholipids, respectively.

For 12 hard coral species from waters near Okinawa Island, the proportion of polar lipids in their total lipid content ranged between 14.3% and 27.8% [36]. The same study found that the total lipids of Alcyonaria *Lobophytum crassum* and hydrocoral *Millepora murrayi* comprised 23.7% and 25.1% polar lipids, respectively. *Goniastrea aspera*, from the same region, exhibited seasonal variations in its polar lipid proportions. These proportions peaked in winter (reaching up to 15% of total lipids) and diminished to 8–10% during summer [56]. The boreal soft coral *Gersemia rubiformis*, sourced from the Avachinsky bay in the Bering Sea, had 31.1% of its total lipids as polar lipids [48]. Major phospholipids of *G. rubiformis* included PC, PE, PS, and CAEP (Table 14). Additionally, another phosphonolipid, ceramide-2-N-methyl-aminoethylphosphonate (CMAEP), was identified, comprising 9.5% of the total phosphorus-containing lipids.

Phosphonolipids, specifically CAEP and CMAEP, identified in corals [48,88,92], have also been found in jellyfishes and actiniae [93,94,95]. The ability to biosynthesize phosphonolipids and MADAG is a distinctive feature of cnidarian metabolism. Both CAEP and CMAEP have a P-C chemical bond in their structures, making them resistant to phospholipase, an enzyme that breaks down phospholipids. It is hypothesized that phosphonolipids help stabilize the cell membranes of cnidarians, which may be exposed to their own lipolytic enzymes [43].

### 2.2. Fatty Acids

One of the primary lipid features is the composition of their fatty acids (FAs), which are represented in a lipid molecule as acyl groups. Lipids within the same class can differ in FA composition or FA placement within the lipid molecule; these variations are termed “molecular species” of that lipid class. Each molecular species is a distinct chemical entity, but every lipid class is composed of a mixture of molecular species. The separation and quantitative analysis of lipid molecular species is a challenging endeavor, typically addressed using a blend of various analytical methods including the gas chromatography-mass spectrometry technique with a tandem mass detector, and/or HPLC. Often, the FA composition of total lipids is found to characterize coral lipids, with the FA composition of neutral, polar lipids, or individual lipid classes determined less frequently.

Most of the information we have concerns the FA composition of total lipids in entire coral colonies. Almost always, FAs are assessed as methyl esters using gas chromatography (GC) equipped with a flame ionization detector. Components are identified using standards and equivalent carbon length (ECL) values. The GC-MS method is typically used to confirm FA structures. Apart from FA methyl esters (FAMEs), pyrrolidides, DMOX, DMDS, and picolinyl esters of FAs are used for GC-MS, which helps identify the position of double bonds and substituent groups in the FA molecule.

It is worth mentioning that early studies analyzed coral FA composition via GC utilized steel-packed columns. On these columns, polyunsaturated FAs (PUFAs) were inadequately separated and sometimes degraded during analysis. Consequently, earlier data on coral FA composition should be approached with caution. The multitude of methods employed for FA analysis complicates direct comparisons between data collected by different researchers.

#### 2.2.1. The Fatty Acid Composition of the Total Lipids of Hard Corals

Meyers [32] was one of the pioneers in conducting extensive analyses of the FA composition of total lipids in corals. He studied eighteen Caribbean Scleractinia species from seven families (Appendix A). The extraction of total lipids and the production of FAME occurred under harsh conditions, which likely contributed to the significant variations observed in the PUFA content, even among individual colonies of the same species. For instance, the concentrations of 22:6n-3 within total FAs varied from 0 to 20.3% in *Madracis decactis*, 0 to 30.3% in *Porites divaricata*, and 0 to 18.3% in *Porites furcata* (Appendix A). Fatty acids 18:3, 18:4, and 20:4n-6 were absent in all coral species studied. However, the prevalence of 16:0, 18:0, and 18:1 in the total FA of hard corals was evident. Concurrently, Light [96] reported high contents of 20:4n-6 (22.6%), 20:5n-3 (14.0%), and 22:6n-3 (13.7%) among the FAs of the total lipids of the gorgonian coral *Plexaura homomalla*. In subsequent studies by Meyers [97,98,99], no PUFA percentages were specified, except for 22:6n-3. He indicated that the FA composition varied according to coral species and nutrition methods but was not influenced by habitat depth (Appendix A).

Furthermore, Meyers [32] contrasted the PUFA content and composition of eight symbiotic hard coral species. He posited that the relatively elevated levels of PUFAs, specifically 22:5 and 22:6n-3, in deep-water species (25–30 m) compared to their shallow-water counterparts might be attributed to the dietary intake of these acids from sources such as copepods. In contrast, zooxanthellae might be the primary FA source for the shallow-water species (2–5 m) (Appendix A).

The inaugural study comparing the FA composition of total lipids via GC on a fused-quartz capillary column examined 12 species of hard corals from Vietnam and Seychelles [100]. Within all species of the Acroporidae family, saturated acids 16:0 and 18:0 were predominant in the total lipids FA composition, while the prevalent PUFAs were 18:3n-6, 20:4n-6, 20:5n-3, and 22:6n-3 (Appendix A). Multiple samples of *Acropora millepora* from diverse habitats revealed significant intraspecific variations in FA composition. For instance, samples of this species from Vietnam’s fringing and oceanic reefs contained 60% and 34% PUFA of total FA, respectively. *Acropora nasuta* displayed a similar shift in n-3 series PUFA (Appendix A). Coral FA of the Pocilloporidae family was distinguished by elevated levels of 20:3n-6, which surpassed the levels of 20:4n-6 in most species. Notably, a high concentration of 18:3n-6 was observed in species from the Poritidae family.

The Dendrophyllidae family corals examined in Latyshev et al.’s study [100] lacked zooxanthellae. Among the surveyed corals, Dendrophyllidae exhibited the highest percentage of 18:1n-9, while the percentage of 22:6n-3 (a primary PUFA in hermatypic corals) remained below 1% of total FA. Both 18:3n-6 and 18:4n-3 percentages in Dendrophyllidae did not surpass 1% of total FA. However, for hard corals hosting zooxanthellae, the combined percentage of 18:3n-6 and 18:4n-3 reached 15% of total FA. The principal PUFAs of the n-6 series were 20:4n-6 and 22:4n-6, while the leading PUFAs of the n-3 series were 20:5n-3 and 22:5n-3.

The FA composition data for hard corals, specifically *Pocillopora verrucosa* and *Stylophora pistillata*, as collected by Harland [35] and Latyshev [100], showed consistency (Appendix A). However, the PUFA content in *Porites porites*, *Montastrea annularis*, and *Goniastrea retiformis*, as observed by Harland [35], was notably lower than that previously documented by Latyshev [100]. The diminished PUFA content in corals from the Caribbean and Red Seas, compared to similar species from the Indo-Pacific, might be attributable to an increased concentration of neutral lipids (WEs and TGs) in the total lipids. These neutral lipids are notably rich in saturated FAs.

The FA composition of *Galaxea fascicularis* coral’s total lipids, cultured for a month under conditions approximating the natural, revealed high contents of 18:3n-3, 18:4n-3, 20:4n-6, 20:5n-3, and 22:6n-3 (collectively constituting 47.5%). There was a relatively low presence of saturated acids 16:0 and 18:0 (collectively 24.0%), and a significant amount of the infrequent acid 22:3n-3 (10.1%) (Appendix A) [101]. Three coral species from the Red Sea, namely *Stylophora pistillata*, *Lobophyllia corymbosa*, and *Echinopora gemmacea*, predominantly contained saturated acids 16:0 and 18:0 in their total lipids. Additionally, the acid 23:0 was detected. The main unsaturated acids were 16:1, 18:1, and 19:3 [37]. It is notable that Al Lihaibi’s study [37] remains unique in reporting the presence of 23:0 and 19:3 among coral FA. Furthermore, 19:3 is labeled as “eicosatrienoic acid” in the article, aligning with the empirical formula of 20:3. It is plausible that this PUFA may, in fact, be 18:3n-6 or 18:4n-3 acid, both of which are abundantly found in all hermatypic corals. The rare acid 14:3 in the FA of the hard coral *Seriatopora hystrix* and the soft coral *Xenia umbellate* is referenced only in Al-Sofyani and Niaz’s work [102]. The same study reports exceptionally low values (3–6 μg/g) of the total lipid content in coral tissues. Extremely low values (3–6 μg/g) of the total lipid content in coral tissues are published in the same work.

The FA composition of total lipids from sixteen species of hard corals, spanning six families (Acroporidae, Pocilloporidae, Poritidae, Faviidae, Pectiniidae, and Fungiidae) collected off the coast of Vietnam during spring, is documented in Appendix A [103]. Dominant saturated acids, 16:0 and 18:0, were prevalent across all coral species. Other major FAs included 14:0, 16:1n-7, 18:1n-9, 18:3n-6, 18:4n-3, 20:3n-6, 20:4n-6, 20:4n-3, 20:5n-3, 22:4n-6, 22:5n-3, and 22:6n-3. Some common FAs such as 18:1n-7, 18:2n-6, 20:1n-7, 20:2n-6, and 22:2n-6 had a content not exceeding 2%. Excluding *Sandalolitha robusta* (Fungiidae), branched FAs and those with odd carbon numbers constituted 0.2–1.4% of the total FAs. Saturated and unsaturated very-long-chain C24 acids were identified in corals of the Poritidae family (Appendix A).

Unsaturated acids made up roughly 50% of the total FAs across all analyzed coral families. *Sandalolitha robusta* exhibited the highest levels of 18:1n-7 and 16:1n-7 (Appendix A). The content of 20:4n-6 and 20:5n-3 varied between 1.7% and 16.5% among species of the Acroporidae family. Notably, *Acropora formosa*, and *Acropora cerealis* had the highest concentrations of 20:4n-6 (14.7%) and 20:5n-3 (16.5%), respectively (Appendix A). Pocilloporidae species were marked by elevated levels of 20:3n-6, 20:4n-3, and 22:6n-3 (Appendix A). The primary PUFA patterns for Poritidae and Pocilloporidae were comparable. However, *Porites lobata* and *Seriatopora hystrix* recorded the highest concentrations of 18:1n-9 (19.0%) and 20:3n-6 (3.1%), respectively (Appendix A). *Echinophyllia orpheensis* (Pectiniidae) had the most significant concentration of 18:1n-9 (Appendix A). The average content of 18:4n-3 was consistent across all coral species studied, though this acid was absent in *Sandalolitha robusta* (Appendix A).

An extensive analysis of the FA composition was conducted using a consistent method on 51 samples of hard corals, sourced from the coastal waters of Vietnam at a depth of approximately 4 m (Appendix A) [68]. The corals under study spanned twenty genera and nine families, including Acroporidae, Agariciidae, Dendrophylliidae, Euphyllidae, Faviidae, Fungiidae, Pectinidae, Poritidae, and Oculinidae. Saturated acids 16:0 and 18:0 were predominant in every coral species. Other prominent FAs included 14:0, 16:1n-7, 18:1n-9, 18:3n-6, 18:4n-3, 20:3n-6, 20:4n-6, 20:4n-3, 20:5n-3, 22:4n-6, 22:5n-3, and 22:6n-3. Minor quantities of saturated branched FAs and those with an odd carbon count, believed to be bacterial markers, were also found. Unsaturated acids constituted around 50% of the total FAs across all examined coral families. A notable finding was the elevated level of 18:1n-7 (5.3% of total FA) in *Goniopora stokesi* (Appendix A), surpassing the level of 18:1n-9 acid. Hermatypic coral species and genera from the same family generally exhibited analogous FA compositions, with the exception of the Poritidae family genera. For instance, corals from the *Porites* genus (Poritidae) exhibited a high concentration of 18:1n-9 and a diminished presence of 20:3n-6 compared to the *Goniopora* genus, which is also part of the Poritidae family (Appendix A).

All the investigated hermatypic corals contained 18:3n-6 (up to 13%) and 18:4n-3 (2–4%), which are posited to be indicators of zooxanthellae lipids. Neither of these acids exceeded 1% in the two hard coral species devoid of zooxanthellae: *Balanophyllia* sp. and *Tubastrea aurea*. Unique to the study, *Galaxea fascicularis*, the sole species from the Oculinidae family under examination, had 1.4% of 22:4n-3 acid—a component absent in all other corals analyzed (Appendix A).

#### 2.2.2. The Fatty Acid Composition of the Total Lipids of Soft Corals

The lipid FA composition of soft corals (Octocorallia) significantly differs from that of hard corals (Hexacorallia), marked by the presence of two rare very-long-chain tetracosapolyenoic FAs (TPAs), namely 24:5n-6 and 24:6n-3. These are chemotaxonomic markers for octocorals [104,105]. Hexacorals are unable to synthesize TPA from C22 PUFA [68].

GC analysis with a packed column revealed that the retention time of TPA methyl esters is notably long, extending beyond two hours. This lengthy retention time is likely why earlier studies using packed GC-columns failed to detect the presence of TPA in soft coral lipids. For instance, Lam et al. [87] profiled the FA composition of total lipids in three gorgonian coral species from Vietnam’s coastal waters but made no mention of TPA (Appendix A). *Psammogorgia nodosa* exhibited a remarkably high percentage of saturated acids at 80.2%, while *Bebryce indica* and *Mopsella aurantia* contained 29.1% and 40.9% of 20:4n-6, respectively. Notably, the azooxanthellate gorgonians, *B. indica* and *M. aurantia*, had low levels of zooxanthellae lipid markers (18:3n-6 and 18:4n-3). While data on the presence of zooxanthellae in *P. nodosa* are absent, the lack of 18:3n-6 and 18:4n-3 suggests this gorgonian species might also lack zooxanthellae. In their study on the chemical structure and distribution of TPA in cnidarians, Vysotskii and Svetashev [104] detailed the FA composition of total lipids in the gorgonian *Paragorgia arborea*, and in the alcyonarians *Eunephthya* sp. and *Sarcophyton* sp., all sourced from depths exceeding 40 m (Appendix A). The warm-water species, *Sarcophyton* sp., showcased a high concentration of saturated acids 16:0 and 18:0. In contrast, the cold-water species, *Paragorgia arborea* and *Eunephthya* sp., predominantly contained monoenoic acids, including 18:1n-9, 20:1, 22:1n-9, 20:4n-6, and 20:5n-3. The ratios of 20:4n-6/20:5n-3 and 20:4n-6/22:6n-3 for *Paragorgia arborea* stood at 2.3 and 10.6, while the corresponding ratios for *Eunephthya* sp. were 0.4 and 0.5. This discrepancy might stem from varying dietary compositions. The highest TPA percentage, 21.6%, was found in *Paragorgia arborea*, while the alcyonarian *Sarcophyton* sp., known to contain zooxanthellae, displayed a notable concentration of 18:4n-3.

An analysis of the total lipid FA composition of soft corals was conducted for 17 species of alcyonarians and gorgonians from Nha Trang Bay (the South China Sea) using GC with a packed chromatographic column [46,106]. The primary fatty acids identified were 16:0, 20:4n-6, 24:5n-6, 18:0, 20:0, 20:5n-3, and 18:1n-9, with their collective content ranging between 57% and 85% of total FAs (Appendix A). These species also contained lesser amounts of acids such as 14:0, 16:1n-7, 16:2, 18:2n-6, 18:3, and 22:6n-3.

Gorgonians exhibited a content of saturated FA averaging 21.7% of total FA, which was half the amount found in alcyonarians, which averaged 43.8%. This disparity is largely due to elevated levels of the acids 16:0 and 20:0 in alcyonarians. The monoenoic acids 16:1 and 18:1 were roughly equivalent in both alcyonarians and gorgonians, averaging 3.0% and 3.9% of total FA, respectively. Notably, alcyonarians showed a significant presence of 20:1, with some species, such as *Sinularia capillosa* and *Sinularia* sp., registering unusually high values of 26.3% and 13.0%, respectively (Appendix A). Although alcyonarians generally displayed higher average levels of 18:2n-6 compared to gorgonians, the gorgonian coral *Junceella fragilis* recorded the maximum content of this acid at 4.4% (Appendix A).

C16 PUFA, constituting up to 8.9% of total FA, was present in all analyzed specimens. The majority of the gorgonian coral species exhibited a high proportion of 20:4n-6, with gorgonian *Plexauridae* spp. 2 leading at 50.5% of its total FA (Appendix A). On average, gorgonians had 1.5 times more 20:4n-6 than alcyonarians. Additionally, the average 20:5n-3 content in the studied gorgonian species was 5.1%, doubling the 2.2% observed in alcyonarians. Only two species, *Euplexaura erecta* and *Nicaule crucifera*, tested positive for the presence of 24:6n-3 [46]. The absence of this acid in other soft coral species from this study likely arises from analytical errors associated with the challenges in detecting chromatographic peaks with prolonged retention times and low concentrations during the GLC analysis on a packed column.

The elevated levels of 24:4n-6 in soft corals from the coastal waters of Vietnam raise skepticism [107]. This component is likely 24:5n-6, which, along with 24:6n-3, was not identified in the FA composition. It is imperative to consider the possibility of inaccurate detection of other main C16-22 PUFAs. The primary fatty acids in the composition of the total lipids of the cold-water alcyonarian *Gersemia rubiformis* include 16:0, 18:1n-9, 20:1n-7, 20:4n-6, 20:5n-3, 24:5n-6, and 24:6n-3 [48,108] (Appendix A). A notable amount of TPA (10.4% of total FAs) was detected. In samples collected in August, saturated, monoenoic, and polyunsaturated acids constituted 10.6%, 25.0%, and 61.8% of the total FAs, respectively. This coral species is believed to maintain a consistently high percentage of unsaturated FAs throughout the year, regardless of water temperature.

The FA composition of the gorgonian *Leptogorgia piccola* (both white and yellow morphs) was studied across different water temperatures [109]. Over 50 fatty acids were identified, including the uncommon 7-methyl-6-hexadecenoic acid and newly discovered fatty acids such as 10-methyl-6-hexadecenoic, 7,9-dimethyl-6-hexadecenoic, 10-methyl-6,9-heptadecadienoic, and 6,9-heptadecadienoic acids. Arachidonic acid, ranging from 13.6% to 20.5%, was predominant among PUFAs in all samples. *L. piccola* colonies from cold water exhibited twice the percentage of PUFAs compared to those from warmer waters. The TPAs 24:5n-6 and 24:6n-3 in cold water samples amounted to 15.8% and 5.3% of total FAs, respectively. However, the total TPAs in warm water samples did not exceed 4% of total FAs.

The FA composition of 11 non-symbiotic alcyonarian species from the *Dendronephthya* genus was examined (Appendix A) [68,79]. Across all species studied, 20:4n-6 emerged as the dominant acid, contributing 26.7 ± 5.9% of the total FA. Other key FAs were 24:5n-6, 16:0, 18:0, 7-Me-16:1n-10, and 24:6n-3, with average contents of 13.0%, 11.9%, 6.1%, 4.8%, and 3.8% of total FAs, respectively. PUFAs were the majority, with TPAs (24:5n-6 and 24:6n-3) averaging 16.8 ± 3.2% of total FAs. The acid 18:3n-6 was almost non-existent. In addition to the branched monounsaturated acid 7-Me-16:1n-10, Dendronephthya’s total FA also contained a notable quantity of saturated FA with an odd carbon number, primarily br-17:0, 15:0, 17:0, and 19:0. The average content of isomers 18:1n-9 and 18:1n-7 was 3.7 ± 1.0% and 2.7 ± 1.1%, respectively. Most Dendronephthya colonies were gathered from shallow waters (2–4 m), while a few species, namely *Dendronephthya* sp. 2, *Dendronephthya* sp. 3, *Dendronephthya* sp. 4, and *Dendronephthya* aff. *involuta*, were harvested using drags at depths of 80–85 m. The FA composition of the *Dendronephthya* genus displayed only slight variations based on habitat depth. The average content of 20:5n-3 in shallow-water samples (3.0 ± 0.8%) was greater than in those from deeper waters (1.6 ± 0.2%). The uncommon acid 18:2n-7, registering at 2.8 ± 1.1%, was found in deep-water colonies. However, this acid was virtually absent in Dendronephthya samples from shallow waters (Appendix A).

The FA composition of total lipids from 16 symbiotic alcyonarian species of the *Sinularia* genus was determined (Appendix A) [79,110]. The dominant saturated FA in *Sinularia* species was the acid 16:0, accounting for 26.6 ± 7.9% of total FAs. On average, saturated FA constituted 36.7 ± 8.6% of total FAs. PUFAs made up approximately half of the Sinularia FA content (50.0 ± 9.5%), with 20:4n-6 being the primary FA, averaging 17.0 ± 4.8%. The TPA content for 24:5n-6 and 24:6n-3 was 5.7 ± 1.5% and 1.6 ± 0.8% of total FAs, respectively. Other significant PUFAs such as 18:4n-3, 20:5n-3, and 22:6n-3, on average, comprised 3.6%, 2.4%, and 4.1% of the total FAs. The proportion of 22:6n-3 in *S. brassica*’s FAs reached 11.8% (Appendix A). The content of 18:4n-3 varied across *Sinularia* species, ranging between from 1.1% to 7.2% (Appendix A). All examined *Sinularia* species had minor quantities of 16:3n-4 and 16:4n-1. Three *Sinularia* species, specifically *S.* aff. *exilis*, *S. brassica*, and *S. siaesensis*, exhibited high levels of 18:3n-6, ranging between 8.4% and 14.1%, and low levels of 16:2n-7, ranging from 0.3% to 2.2% (Appendix A). In contrast, other *Sinularia* species had reduced levels of 18:3n-6 (0–2.9%) and elevated levels of 16:2n-7 (3.5–11.2%). The FA composition of four *Sinularia* species was determined utilizing a packed chromatographic column [46]; these species belong to the group exhibiting low 18:3n-6 content. Typically, an elevated concentration of 16:2n-7 was paralleled by a significant presence of the acid 18:2n-7, which constituted up to 8.4% of the total FA in *S. lochmodes* (Appendix A). The 7-Me-16:1n-10 percentage in *Sinularia* varied considerably, from 0% to 3.6%, but on average was 0.7% of the total FA.

The FA composition of total lipids from 12 symbiotic alcyonarian species of the *Sarcophyton* genus was outlined (Appendix A) [68,79]. The FA profile of this genus predominantly comprised of saturated FA, averaging 38.7 ± 8.5%, with 16:0 being the primary saturated FA, making up 30.0 ± 7.7% of the total FAs. PUFAs accounted for roughly half of the FA (46.8 ± 8.7%), with 20:4n-6 being the principal component, averaging 17.9 ± 4.6%. TPA values for 24:5n-6 and 24:6n-3 were 5.7 ± 2.0% and 0.7 ± 0.2% of total FAs, respectively. Other key PUFAs, such as 18:4n-3, 20:5n-3, and 22:6n-3, constituted 5.0%, 2.4%, and 2.9% of the total FA, respectively. The 18:4n-3 content ranged between 2.4% and 8.9% across various *Sarcophyton* species (Appendix A). Both 18:2n-6 and 18:3n-6 had an average representation of 0.3% of the total FA. The average proportion of 7-Me-16:1n-10 stood at 1.4 ± 0.5%. All examined *Sarcophyton* species were typified by a high concentration of the 16:2n-7 acid, averaging 9.2 ± 3.3%. In *S.* cf. *glaucum*, this acid constituted as much as 16.4% of the total FAs (Appendix A). The acid 18:2n-7 averaged at 1.3 ± 0.4% of total FAs. Two species, *Sarcophyton crassocaule* and *S.* aff. *glaucum*, whose FA content was determined using a packed chromatographic column [46], fall within the category characterized by a high 16:2n-7 concentration (Appendix A). On the other hand, three other *Sarcophyton* species explored in the study of Imbs et al. [46] exhibited a low concentration of 16:2n-7 (0.6–0.9%) and a high quantity of 18:3n-6 (5.5–12.0%) (Appendix A).

The FA composition of the total lipids for seven symbiotic alcyonarian species from the *Lobophytum* genus has been detailed (Appendix A) [68,79]. The FA profile for this genus predominantly consisted of saturated FAs, averaging 36.5 ± 6.9%. The primary saturated FA identified was 16:0, which accounted for 27.6 ± 6.2% of total FAs. PUFAs represented approximately half of the FAs at 44.2 ± 8.9%, with 20:4n-6 being the most prevalent, averaging 20.9 ± 5.6%. TPA values of 24:5n-6 and 24:6n-3 were 6.6 ± 4.1% and 1.0 ± 0.6% of the total FAs, respectively. The average levels of 18:4n-3 stood at 2.7 ± 1.7%, while 20:5n-3 and 22:6n-3 each constituted no more than 2% of the total FAs. Specifically, in *Lobophytum* cf. *delectum*, the 24:5n-6 content reached 14.9% of the total FAs, and this species showcased the highest concentration of 20:4n-6 (30.4%) (Appendix A). The average representations of 18:2n-6 and 18:3n-6 in the total FAs were both 1%. The 7-Me-16:1n-10 content averaged 2.2 ± 1.3%. The acids 16:2n-7 (averaging 6.7 ± 3.5%) and 18:2n-7 (averaging 2.1 ± 1.0%) were identified in all examined *Lobophytum* species except for *Lobophytum batarum*. It is plausible that the *L. batarum* colony sampled might have experienced a reduction in its zooxanthellae content.

The FA composition of the total lipids for various alcyonarian species from the genera *Cladiella*, *Lytophyton*, *Cespitularia*, *Clavularia*, *Heliopora*, *Carijoa*, *Klyxum*, *Lemnalia*, and *Nephthea* has been provided (Appendix A) [68,79]. Data concerning the presence of zooxanthellae are absent for *Heliopora* and *Carijoa* genera. The *Nephthea* genus encompasses species that both have and lack zooxanthellae. Among all soft coral species, only *Heliopora coerulea* (blue coral) and members of the *Epiphaxum* genus possess a rigid exoskeleton akin to that of hard corals. *H. coerulea* is distinguished by its extremely low concentration of 20:4n-6 (0.6%) and the near-total absence of TPA 24:5n-6 and 7-Me-16:1n-10 (Appendix A).

Symbiotic alcyonarians and *Carijoa* species exhibited notable amounts of 18:4n-3, ranging from 2.6% to 10.6%. Specifically, *Cladiella laciniosa*, *Cespitularia* sp., *Carijoa riisei*, *Klyxum molle*, and two *Lemnalia* species contained between 4.4% and 14.2% of 18:3n-6. However, 18:3n-6 was nearly absent in *Cladiella subtilis*, *C. pachyclados*, *Lytophyton* sp., and *Clavularia* sp. (Appendix A). In the lipids of the investigated alcyonarians (Appendix A), the primary C20-24 PUFAs included 20:4n-6, 20:5n-3, 22:6n-3, and 24:5n-6. The highest concentration of 20:5n-3 (27.2%) was found in *Lemnalia* cf. *peristyla* (Appendix A). Corals with zooxanthellae exhibited low levels of 16:2n-7 and 18:2n-7. The highest 18:1n-9 levels (ranging from 26.8% to 33.3%) were detected in *Lytophyton* sp. and three *Nephthea* samples (Appendix A). *Clavularia* sp. displayed the highest content and diversity of C22-24 n-3 acids (including 22:4n-3, 22:5n-3, 22:6n-3, 24:4n-3, and 24:5n-3), with 22:6n-3 being the dominant acid. Notably, 24:5n-3 (2.4%) was exclusively found in *Clavularia* sp. (Appendix A) [68]. Although the FA composition of *Clavularia* sp. contained a minimal amount of 18:3n-6 (0.5%), the content of 18:4n-3 was 8.3% of total FAs (Appendix A).

Comparative findings regarding the FA composition of the total lipids for seven gorgonian coral genera without zooxanthellae (*Acanthogorgia*, *Acabaria*, *Chironephthya*, *Echinogorgia*, *Menella*, *Ellisella*, and *Bebryce*) and two gorgonian coral genera with zooxanthellae (*Paralemnalia* and *Rumphella*) are presented in Appendix A [111]. Across all species, 20:4n-6 (approximately 40% of total FAs) and 16:0 predominated. Other principal FAs included 14:0, 16:1n-7, 7-Me-16:1n-10, 18:0, 18:1, 20:5n-3, 22:5n-6, 22:6n-3, 24:5n-6, and 24:6n-3. The average ratio of 18:1n-9 to 18:1n-7 was 17.9 for symbiotic species and 1.5 for non-symbiotic species. PUFAs from the n-6 series dominated in all examined species, with the n-6/n-3 ratio ranging from 2.7 to 17.1 and averaging 7.0. Unusual furan acids (comprising up to 9.7% of total FAs) were present in the total FA composition of gorgonians without zooxanthellae. The primary furan acids were 14,17-epoxy- 15-methyldocosa-14,16-diene and 14,17-epoxy-15,16-methyl- docosa-14,16-diene acids (Appendix A). C25-28 demospongic acids were detected in all *Bebryce studeri* species (up to 20% of total FAs) (Appendix A). In corals without zooxanthellae, there were significantly higher (*p* < 0.01) quantities of saturated branched FAs, acids with an odd number of carbon atoms, 18:1n-7, and 7-Me-16:1n-10 (up to 5.2% of total FAs) compared to corals containing zooxanthellae. The latter group was characterized by the presence of 18:3n-6, 18:4n-3, and 16:2n-7. In comparison with the typical level of 24:5n-6 (9.4%) and 22:4n-6 (0.6%) for octocorals, some analyzed coral samples exhibited an anomalously low concentration of 24:5n-6 (0.4%), accompanied by a very high level of 22:4n-6 (up to 11.9%) (Appendix A).

The FA composition of gorgonian corals was studied alongside a broader group of hard and soft corals (Appendix A) [68]. These corals encompassed twelve genera and five families: Nidaliidae, Parisididae, Plexauridae, Primnoidae, and Subergorgiidae. Predominant saturated FAs across all coral species were acids 16:0, 18:0, and 14:0. *Hicksonella princeps* exhibited the highest concentrations of 16:0 (29.8–35.8%), 20:0 (4.1–7.1%), and 22:0 (2.3–4.9%). Monoenoic branched FAs, mainly 16:1n-7 and 18:1n-9, constituted no more than 10% of the total. All gorgonian corals contained the branched monounsaturated acid 7-Me-16:1n-10, with *Siphonogorgia* cf. *harrisoni* having up to 9.5% of its total FAs. Trace amounts of acids such as 16:2n-7, 18:2n-7, 18:3n-6, and 18:4n-3 were observed in most gorgonians, with *H. princeps* having the highest concentration of 18:4n-3 at 3.3% (Appendix A).

Long-chain PUFA dominants were 20:4n-6 (peaking at 41%), 22:6n-3, and 20:5n-3. Soft corals consistently showed elevated levels of 24:5n-6 (up to 16%) and 24:6n-3 (as high as 6%). Interestingly, these percentages surpassed those in azooxanthellate species. *H. princeps*, *Viminella* cf. *petila*, and *Narella* sp. had half the average concentration of 20:4n-6 compared to other gorgonians (Appendix A). Within the *Viminella* genus, the primary TPA was 24:6n-3, whereas other alcyonarians and gorgonians primarily exhibited 24:5n-6 (Appendix A).

All shallow-water collected gorgonian coral species predominantly featured n-6 series PUFA, specifically 20:4n-6 and 24:5n-6. Only the unbranched gorgonian, *Viminella*, exhibited a reduced 24:5n-6 level but a significantly elevated 22:4n-6 level (averaging 9.2%, in contrast to 0.7% in other soft coral species). The deep-water sample of *Narella* sp. stood out, showing the highest n-3 series PUFA concentration and a near absence of 24:5n-6. Gorgonian coral species such as *Annella mollis*, *Parisis* cf. *minor*, and *Paracis* cf. *horrida* from the *Viminella* genus and Plexauridae family contained 2.1 to 5.0% of acid 22:5n-6 (Appendix A). Notably, such a significant concentration was only observed in one alcyonarian species, *Clavularia* sp. (Appendix A).

#### 2.2.3. The Fatty Acid Composition of Polar Lipids of Corals

Polar lipids (PLs) are the primary constituents of cell membranes. The fatty acid (FA) composition of PLs, which are structural lipids, tends to be more stable than that of neutral lipids, which serve as reserves. This stability in PL composition is often more reflective of the specific species of the organism. As such, some researchers use the FA profile of PLs as an indicator of the chemical makeup of the organism under study. Typically, ethanolic extracts are employed when searching for new biologically active compounds in marine organisms. Since these extracts predominantly contain PLs, only the FA composition of PLs is typically analyzed in such studies.

The FAs of PLs from the hermatypic coral *Stylophora pistillata*, collected at various depths (ranging from 3 to 35 m), showed little variation in terms of unsaturation degree and n-3/n-6 acid ratios (Appendix A) [100]. The majority of 18:3n-6, 18:4n-3, 20:4n-6, and 22:4n-6, as well as a significant portion of 20:5n-3, were found in the PLs. In contrast, neutral lipids were predominantly composed of 18:1n-9 and 22:6n-3. As the depth of the habitat increased, concentrations of 18:3n-6 and 18:4n-3 in the FAs of PLs rose, while the 20:5n-3 levels declined. These shifts are thought to correlate with an increase in the number of zooxanthellae and changes in food composition associated with greater depths.

The FA composition of the polar lipids (PLs) of 12 cnidarian species from Okinawa Island is detailed in Appendix A [36]. Across all studied species, the FAs ranged from 14 to 22 carbon atoms. The soft coral *Lobophytum crassum* did not exhibit C24 PUFA in its PL FA composition, although the presence of TPAs is typically a chemotaxonomic criterion for octocorals [68]. None of the coral species had branched FAs; however, the presence of 2-OH-16:0 was noted, accounting for up to 4% of the total FAs. The predominant FA was 16:0, ranging from 15.8% in *Porites cylindrica* to 54.8% in *Galaxea fascicularis*.

PUFA dominance in the FA composition of polar lipids was observed in species such as *Porites lutea*, *P. cylindrica*, *Tubastrea* sp., and *L. crassum*, with *P. lutea* reaching up to 54.9%. The primary PUFAs were 20:4n-6, 18:4n-3, 18:3n-6, 22:4n-3, and 22:5n-3. *Tubastrea* sp. had the highest content of 22:5n-3. This species lacks zooxanthellae, explaining the notably low concentrations of 18:4n-3 and 18:3n-6. Notably, essential long-chain PUFAs, such as 20:4n-6, 20:5n-3, and 22:6n-3, were virtually absent in the PL FAs of *Pocillopora damicornis*, *P. verrucosa*, and *Goniastrea aspera*. Although corals’ total FA typically contains substantial amounts of these key PUFAs [68], the PUFA content in PLs is generally higher than in total FAs.

One hypothesis is that the 24 h preliminary decalcification of coral colonies using formaldehyde, as employed in the study [36], might decrease PUFA levels and lead to the degradation of TPAs, potentially due to oxidation and polymerization processes.

The primary fatty acids (FA) present in the phospholipids of four species from the *Pseudopterogorgia* genus, five from the *Eunicea* genus, and two from the *Gorgonia* genus included 16:0, 18:3n-6, 18:4n-3, 20:4n-6, 22:6n-3, 24:5n-6, and 24:6n-3. Notably, the n-6 series of polyunsaturated fatty acids (PUFAs) were predominant [45,89,90] (as seen in Appendix A). The *cis*- and *trans*-isomers of 7-methyl-6-hexadecenoic acid and minor amounts (around 1%) of α-hydroxy acids (2-OH-21:0 and 2-OH-22:0) were also detected. *Gorgonia mariae* had 0.8% of (*Z*)-7-Me-16:1n-10 and 3.5% of (E)-7-Me-16:1n-10, whereas *G. ventalina* only contained the (*E*)-7-Me-16:1n-10 isomer at 1.6%. The genus *Pseudopterogorgia* exhibited up to 14% of total FAs as TPAs. The significant amounts of 18:3n-6 and 18:4n-3 (up to 28% of total FAs) in phospholipids indicate the presence of zooxanthellae in these gorgonians.

The primary FA components in the phospholipids of *Gersemia rubiformis* were 20:4n-6, 20:5n-3, 24:5n-6, and 20:1n-7 (see Appendix A) [48,108]. When compared to neutral lipids, the PLs of this cold-water *Alcyonaria* demonstrated a notably higher content of the n-6 series PUFAs (20:4n-6, 22:4n-6, and 24:5n-6) than that of the n-3 series (with an n-6/n-3 ratio of 2.6). The TPA content in *Gersemia rubiformis’s* PLs was half that in total lipids, constituting 14.7% of the total FA.

For Vietnamese *Sinularia*, the proportion of 16:2n-7 in its phospholipid FA was less than that in its neutral lipids. In contrast, the proportions of C16 PUFA, 16:3n-4, and 16:4n-1 were slightly elevated [79] (as detailed in Appendix A). The FAs in *Sinularia* sp. phospholipids contained a high percentage of 18:4n-3 (14.1%) and 24:5n-6 (9.5%). However, these acids’ proportions in total lipid FAs were only 1.7%.

Studies of the total lipid composition of the hard coral *Montipora informis* revealed that the relative proportions of phospholipids and sterols were more substantial in neoplastic tissues compared to healthy ones, primarily due to a marked decrease in wax esters [61]. The main FA in *M. informis* polar lipids was 16:0 (accounting for 45.9%). Other significant FAs included 18:0, 18:1n-9, 18:2, 18:3, 20:4, and 22:6. Notably, 22:4 (constituting 4.0 ± 1.7% of total FAs) was identified in the phospholipid FA of *M. informis* neoplastic tissues but was absent in healthy coral tissues.

#### 2.2.4. Fatty Acids and Lipids of Milleporidae Family Hydrocorals

Alongside hard and soft corals from the subclasses Hexacorallia and Octocorallia within the Anthozoa class, another prominent member of the coral reef community is the hydrocorals from the Milleporidae family, belonging to the Hydrozoa class. Their lipid composition, as discussed in earlier chapters, displays certain similarities and differences.

In terms of total lipid content, hydrocorals exhibited only slight variations compared to Scleractinia corals. For instance, the total lipids found in *Stylaster* sp. and *Millipora exesa*, sourced from the Great Barrier Reef in Australia at a depth of 4 m in July, were 2.6 and 4.5 mg/g of coral, respectively [33]. Meanwhile, *Millepora murrayi* from Okinawa Is. had a total lipid content equivalent to 29.7% of the tissue’s dry weight (DW) [36].

Delving into the primary lipid classes of hydrocorals, these include wax esters (WEs), sterol esters (SEs), MADAGs, triglycerides (TGs), free fatty acids (FFAs), sterols (STs), and phospholipids (PLs). Table 15 and Table 16 details the total lipid compositions for *Millepora murray*, *M. dichotoma*, and *M. platyphylla* [36,68]. Notably, nonpolar lipids were dominant, accounting for over 65% of the total lipids. Each of the species studied had a significant presence of MADAGs, ranging between 9 and 19% of the total, which is a lipid type characteristic of cnidarians. The notably high FFA content (12.3%) observed in *M. murray’s* lipids might be attributed to the heightened activity of lipases or phospholipases during the lipid extraction process.

Meyers [32] was a pioneer in investigating the fatty acid (FA) composition of total lipids in the hydrocoral *Millepora alcicornis*. His research revealed that the content of 22:6n-3 in the total FAs of this hydrocoral varied widely, ranging from 0 to 23.5%. Interestingly, 20:4n-6 was completely absent. In another of Meyer’s studies, the FA composition of *Millepora* sp. showed the following distribution: 14:0 (3.7%), 16:0 (46.3%), 18:0 (32.8%), 18:1 (6.1%), and 20:0 (11.1%). These figures, as highlighted in a comparative review on cnidarian lipid compositions [43], deviate from the typical polyunsaturated fatty acid (PUFA) distribution observed in this group of marine organisms.

Subsequent studies revealed that the primary saturated fatty acids (FAs) in hydrocorals from the Milleporidae family were 16:0 and 18:0. The predominant monounsaturated acid was 20:1, with a notably low presence of 16:1n-7 (Table 17) [100]. The polyunsaturated fatty acid (PUFA) profile of *Millepora* sp. was marked by a minimal amount of 20:4n-6 and 20:5n-3. The 18:3n-6, typical for hermatypic corals, was nearly non-existent. The dominant PUFAs in *Millepora* sp. were 22:6n-3 (constituting up to 61.5% of total FAs) and 22:5n-6, which was not found in Scleractinia. Data detailing the FA distribution in the total lipids of two *Millepora* species from Vietnam’s coastal waters (Appendix A) [68] aligned with previously published results [100]. In addition to the aforementioned FA composition characteristics (high levels of 22:6n-3 and 22:5n-6, and low levels of 20:4n-6, 20:5n-3, 16:1n-7, and 18:3n-6), the presence of 7-Me-16:1n-10 was noted in these hydrocorals.

Yamashiro et al. [36] found that the major FA components in the polar lipids of *Millepora murrayi* (sourced from Okinawa, Japan) were 16:0 (29.0%), 18:0 (17.4%), and 20:0 (12.8%). PUFAs 22:4n-3 and 22:6n-3 accounted for only 3.3% and 3.7%, respectively. The content of 20:4n-6 and 20:5n-3 did not exceed 0.3% of total FAs. A significant presence of 18:4n-3 (10.8%) was identified. The absence of 22:5n-6 and the reduced levels of PUFAs, which usually have higher content in polar lipids than in total lipids, make these data hard to correlate with other findings regarding hydrocoral FA composition.

#### 2.2.5. Fatty Acids and Lipids of Zooxanthellae and the Host Tissues

Most coral species house zooxanthellae, intracellular symbiotic algae belonging to the Symbiodinium group. Generally, the total lipids of non-symbiotic coral species are characterized only by the lipids present in the coral polyp tissues. In contrast, corals that host zooxanthellae have total lipid compositions derived both from the polyp tissues (the host) and the zooxanthellae (the symbionts). The lipid composition of autotrophic zooxanthellae significantly varies from the lipid makeup of heterotrophic polyps. There are limited studies examining the lipid composition of isolated zooxanthellae and pure polyp tissues. This scarcity is largely attributed to the methodological challenges associated with obtaining pure fractions of both symbionts and host tissues.

For the hard coral *Pocillopora capitata*, research by Patton and colleagues [31] demonstrated that the lipid class distribution in zooxanthellae is quite distinct from that found in the entire coral colony. Lipids from zooxanthellae constituted about 19% of the colony’s total lipids. In the entire colony, neutral lipids represented 75% of the total lipid content, while in zooxanthellae, they accounted for just 8%. Structural lipids, which include sterols, phospholipids, and glycolipids, comprised 67% of the total lipids in pure zooxanthellae and 16% in the entire *P. capitata* colony. A minor quantity of waxes was identified in the lipids of zooxanthellae, suggesting that wax biosynthesis likely occurs within the polyp tissues. The fatty acid composition of total lipids, as well as MGDG and DGDG from zooxanthellae extracted from hard corals of the Acroporidae, Faviidae, and Pocilloporidae families, can be found in Appendix A [112].

The predominant fatty acids (FAs) in the total lipids of zooxanthellae were 16:0, 18:3n-6, 18:4n-3, 18:5n-3, 20:5n-3, and 22:6n-3. Notably, there was a high concentration of the FA 18:5n-3 (reaching up to 7.6%), which is believed to be typical for free-living dinoflagellates. Glycolipids, such as MDGD and DGDG, form the foundation of thylakoid membranes in photosynthetic organisms and are almost absent in animal cells. Consequently, the FAs of glycolipids derived from the total lipids of the entire coral colony essentially reflect only those of the zooxanthellae. The primary FAs of these glycolipids were 18:3n-6, 18:4n-3, 18:5n-3, and 20:5n-3. Interestingly, the content of 18:3n-3, which is common in higher plant lipids, was significantly less than its isomer 18:3n-6, which is more typical for animals. Variations in the FA compositions of zooxanthellae isolated from different corals, foraminifera, and mollusks are believed to arise from the hosting of different species of zooxanthellae, each with unique FA profiles, across diverse taxonomic aquatic groups [112].

The FA compositions of phospholipids (PLs) and triglycerides (TGs) of three symbiotic dinoflagellate (SD) morpho-physiological types (L, B, and G) derived from hermatypic corals from Shikoku Island in Okinawa, Japan, were explored [113,114,115]. The hydrocoral *Millepora intricata* hosted L-type symbionts, the reef-building coral *Pocillopora damicornis* housed B-type symbionts, while *Seriatopora caliendrum* and *Seriatopora hystrix* carried G-type symbionts. Both B and G types were found in *Stylophora pistillata* [113]. These SD types exhibit differences in cellular dimensions, shapes, structural components, photosynthetic capabilities, initial products, pigment storage, and rates of cell division and degradation. The principal FAs in L-type SD polar lipids from *M. intricata* were 16:0, 18:4n-3 (26.2%), 18:5n-3 (8.7%), 22:5n-6 (10.3%), and 22:6n-3 (17.8%), with only trace amounts of 20:5n-3 and 20:4n-6 (as shown in Appendix A) [115]. The FA compositions of B-type SD polar lipids from *P. damicornis* and G-type SD polar lipids from *Seriatopora* were nearly identical, dominated by acids 16:0, 18:4n-3, 20:4n-6, 20:5n-3, and 22:6n-3. The concentration of 18:4n-3 in B-type SD (17.5%) surpassed that in G-type SD (10%). In contrast, the levels of 22:6n-3 were opposite, with 9.6% in B-type and 16.4% in G-type. B- and G-type SD polar lipids had a minor content of 18:5n-3 (around 1%), but they had a significant proportion of n-6 series acids, specifically 18:3n-6 and 20:4n-6, when compared to the L-type SD. FA patterns in TG of SD differed in the content of four principal acids, 16:0, 16:1n-7, 18:0, and 22:6n-3 (Appendix A) [115]. The 18:0 content was more than 20% in TG of L-type SD, while the share of this component was not more than 5% of the acids’ sum in B- and G-types’ SD. The acid 16:1n-7 was a minor component (1%) in TG of L-type SD in comparison with B- and G-types (about 6 and 12%, respectively). In TG of B- and G-types’ SD, the level of 16:0 was twice as high as in 22:6n-3.

Distinct variations in the FA composition of L-type SD led researchers to propose that L-type may represent a distinct species or even a separate genus from Symbiodinium. In contrast, B- and G-type SDs appear to be subspecies or strains of the same species [115]. The potential influence of the host organism on the FA composition of its zooxanthellae was not addressed.

One pioneering study comparing the total lipid FA composition of isolated pure zooxanthellae (symbionts) and polyp tissues (host) from the same coral colony used *Montipora digitata* as its subject [116]. While the total lipids of zooxanthellae and polyp tissues exhibited a similar array of FAs, the relative proportions of some FAs differed significantly between the two. The percentage distribution of FAs in pure zooxanthellae compared to host tissues is detailed in Table 17. The fatty acid 16:0 was the primary component in both the symbionts and the host, accounting for 32.7% and 50.2%, respectively. In zooxanthellae, predominant fatty acids included 18:3n-6, 18:1n-9, 18:0, 18:4n-3, 16:1, 22:6n-3, and 20:5n-3. In contrast, the polyp tissues were dominated by 16:1, 18:0, 18:1n-9, 18:3n-6, 22:4n-6, and 20:4n-6. Notably, only zooxanthellae fractions exhibited trace amounts of 18:3n-3 and 18:5n-3; these were absent in the coral cell tissues. Odd-numbered FAs and 18:1n-7, indicators of bacterial presence, were virtually absent in zooxanthellae but were minimally detected in host cells. Fatty acids of zooxanthellae had higher proportions of n-3 series PUFAs (such as 18:4n-3 and 20:5n-3) and 18:2n-6, but lower proportions of 16:0 compared to the polyp tissue FAs (*p* < 0.05). A study simulating coral “bleaching” conditions—through intensified light exposure combined with elevated temperatures—revealed impacts on the FA composition of both the host organism and the *M. digitata* zooxanthellae [117]. Both coral cells and zooxanthellae showed a reduction in PUFA levels under these conditions. These alterations were more pronounced in the host cells, suggesting either a heightened sensitivity of the host to environmental shifts or a protective mechanism by the host towards its symbionts.

**Table 17 marinedrugs-21-00539-t017:** Percentage ratio of FA in pure zooxanthellae to FA in the host of *Montipora digitata* [116].

Fatty Acid	Ratio	Fatty Acid	Ratio
14:0	1.48	20:5n-3	6.48
16:0	0.65	22:4n-6	0.70
16:1	0.45	22:5n-3	1.17
18:0	0.82	22:6n-3	6.59
18:1n-9	0.86	Σ SFA	0.72
18:2n-6	1.67	Σ MUFA	0.81
18:3n-6	2.04	Σ PUFA	1.92
18:4n-3	3.92	Σ bacterial FA	0.28
20:3n-6	0.38	Σ n-6 PUFA	1.31
20:4n-6	0.52	Σ n-3 PUFA	4.92

Comparison of the total lipid FA composition of *Symbiodinium* spp. cultivated dinoflagellates thylakoid membranes isolated from some hard coral’s species showed that the content ratio of 18:1n-9/18:4n-3 is rather higher in clones that are resistant to ambient temperature increase (Table 18) [118]. The authors suppose that zooxanthellae thylakoid membranes containing more polyunsaturated lipids are damaged easily by temperature increase, which leads to membrane permeability increase, failure of photosynthetic complex, and, finally, to zooxanthellae emission and coral colony death.

Subsequently, Díaz-Almeyda and colleagues [119] examined the relative thermal stability and FA composition of membranes in various phylogenetic types of *Symbiodinium* spp. cultured at 24 °C and 31 °C. Their findings did not corroborate the theory that the unsaturation of FAs in the thylakoid membrane plays a pivotal role in coral bleaching or that it could serve as a reliable indicator for predicting the thermal vulnerability of symbiotic reef corals. The primary FAs of whole cells and a fraction rich in photosynthetic membranes of *Symbiodinium* (cultured at 24 °C) included 16:0, 18:0, cis-18:1n-9, 18:3n-3, 18:4n-3, and 22:6n-3 (as shown in Table 19) [119]. While the presence of isomeric cis-18:1n-7 and trans-18:1n-9 was identified, certain vital FAs such as 18:5n-3 and 20:4n-6 were notably absent.

To identify the producer of clavulones (bioactive prostaglandin analogs) in the symbiotic relationship between the soft coral *Clavularia viridis* (from Okinawa Is.) and its symbionts, a method was developed to isolate pure zooxanthellae and host tissues from this coral species [120]. The purity of the obtained preparations of zooxanthellae and polyp tissues was verified through the analysis of their polar lipid and total FA compositions. At a qualitative level, zooxanthellae were found to have several classes of glycolipids and a minimal quantity of phospholipids. In contrast, the dominant polar lipids in the polyp tissues were phospholipids, with glycolipids being almost non-existent. The authors believed that this distribution of polar lipids affirmed the high purity of the extracted fractions. However, details regarding the chemical nature of the identified glyco- and phospholipids, along with their quantitative composition, were not disclosed. The FA composition of the total lipids from the pure fractions of *C. viridis* was explored. The FAs of polyps and zooxanthellae consisted of 38.2% and 10.4% of 20:4n-6, and 0.5% and 43.1% of 20:5n-3, respectively (full FA composition details were not provided). The remarkably low presence of 20:4n-6 in the zooxanthellae FA further solidified the purity of the isolated symbionts.

Pure zooxanthellae and polyp tissues from the reef-building coral *Turbinaria reniformis* (found in the Red Sea) were isolated to study the effects of light and nutrition levels on the FA and sterol composition of both the symbionts and the host organism of this coral species [121]. The FA profiles of zooxanthellae and the host showed distinct differences.

In zooxanthellae, C16 FAs, especially 16:0, were dominant, accounting for an average of 22% of the total FAs. Other major components in zooxanthellae included 20:5n-3 (ranging between 9 and 13%), 22:6n-3 (approximately 10%), 18:4n-3 (spanning 7–10%), and 18:3n-6 (oscillating between 6 and 8%).

In contrast, animal tissues primarily consisted of C16 FAs, C18 FAs, and 20:4n-6, which collectively represented, on average, 33%, 16%, and 11% of the total FAs, respectively. Notably, the concentrations of 18:4n-3 (1–2%), 18:3n-6 (around 3%), and 22:6n-3 (approximately 2%) in the animal tissues were markedly lower than those in zooxanthellae. Overall, the content of PUFAs, particularly those from the n-3 series, was significantly reduced in animal tissues, at 34% and 7% respectively, compared to the 51% and 30% observed in zooxanthellae.

The first insights into the lipid and FA composition of lipid classes in pure zooxanthellae and polyp tissues of soft corals were derived from *Sinularia* sp. [68]. The primary lipid class content is delineated in Table 20. Only the content of WE, MADAG, and PL showcased significant differences between zooxanthellae and polyps (*p* < 0.01). In contrast, TG, FFA, and ST content differences were not statistically significant (*p* > 0.05). Zooxanthellae’s total lipids were predominantly polar, whereas the polyp tissues were dominated by neutral lipid classes. Notably, MADAGs constituted 35% of the total lipids in polyps but were nearly absent in zooxanthellae. The chief PUFAs in zooxanthellae, listed in Appendix A, were 18:4n-3, 20:5n-3, and 22:6n-3. Remarkably, significant quantities of C16 PUFAs, 16:3n-4, and 16:4n-1 (totaling 8.9%) were identified in *Sinularia* sp.’s zooxanthellae. Preliminary evaluations indicated that zooxanthellae lipids comprised a minor fraction (1.1% of total FA) of TPAs (24:5n-6 and 24:6n-3).

Pure symbiont fractions (SFs) and host fractions (HFs) were isolated from ten symbiotic cnidarians, including one hydrocoral species, two soft coral species, and seven hard coral species, all found in Vietnam’s shallow waters (refer to Table 21). A subsequent comparative analysis was conducted on the clade designation of symbionts (Table 21) and the FA composition of total lipids in both HFs and SFs (Appendix A) [122]. No clear correlation was discerned between the clade and the FA composition of zooxanthellae. For instance, the shared clade C71a of symbionts was isolated from *Sinularia* cf. *capitalis*, *S. polydactila*, and *Acropora intermedia* (Table 21). However, the FAs in SFs from *Sinularia* included distinct acids 16:3n-4, 16:4n-1, 24:5n-6, and 24:6n-3, which were missing in SFs from *A. intermedia* (Appendix A). The content of common FAs in SFs from both *Sinularia* and *Acropora* also varied significantly. Analogously, discrepancies in FA percentages emerged when comparing clade D/D1a isolated from *M. digitata* and *P. damicornis*.

Within the lipids of all isolated symbiont fractions (SFs), the primary PUFAs were 18:4n-3, 20:4n-6, 20:5n-3, and 22:6n-3, as outlined in Appendix A. The uncommon 18:5n-3 was also identified. SFs derived from hard corals exhibited an abundance of 18:3n-6. In contrast, SFs from soft corals were distinctively marked by the presence of acids 16:3n-4 and 16:4n-1. Acids such as 18:3n-6, 18:4n-3, 18:5n-3, 16:3n-4, and 16:4n-1 predominated in SF lipids, with only trace amounts of these acids appearing in host fraction (HF) lipids, as visualized in Appendix A and Figure 6. The proportional presence of 20:5n-3 and 22:6n-3 in SF lipids exceeded that in HF lipids.

Specifically in hard corals, the HF showed a higher percentage of 22:4n-6. Interestingly, for soft corals, acids typically associated with HF lipids (24:5n-6 and 24:6n-3) were also observed in SF lipids. The hydrocoral SF stood out with its elevated percentages of 20:0, 22:5n-6, and 22:6n-3 when compared to coral SF. The unique PUFA characteristic of *Millepora*, namely 22:5n-6, was identified in both HF and SF lipids, as showcased in Appendix A and (Figure 13).

#### 2.2.6. The Chemical Structure of Uncommon Fatty Acids of Corals

Corals exhibit a diverse array of fatty acids (FAs), many of which are atypical in both marine and terrestrial animals. The α-Hydroxy-hexadecanoic acid, known as 2-OH-16:0 (24), which comprises 1.1% of the total FAs, was identified in 12 hard coral species, an alcyonarian species, and a hydrocoral species found around Okinawa Is., Japan. Its structure was deciphered using GC-MS [36]. Additionally, the 3,6,9,12,15- octadecapentaenoic acid, termed 18:5n-3 (25), was detected in various corals, and its presence was confirmed using GC and GC-MS.

Octocorals are distinct in their FA composition, particularly with the inclusion of the all-cis-6,9,12,15,18-tetracosapentaenoic (24:5n-6) (**26**) and all-*cis*-6,9,12,15,18,21-tetracosahexaenoic (24:6n-3) (**27**) acids, which hexacorals lack. At times, these acids can constitute over 20% of the FA content. Noteworthy research by Vysotskii and Svetashev [104,123,124] has purified these acids using HPLC and Ag-TLC methods. Their characteristics were then confirmed through an array of spectroscopic techniques and MS.

Some corals also exhibited the presence of the rare 7,10,13,16-nonadecatetraenoic acid, 19:4n-3 (**28**), and a novel (5*Z*,9*Z*)-14-methyl-5,9-pentadecadienoic acid (**29**), of which the latter’s structure was ascertained using several physico-chemical methods. Another identified acid, 7-methyl-(6*E*)-hexadecenoic acid (**30**), was predominantly found in non-symbiotic soft corals of the *Dendronephthya* genus, constituting up to 7.3% of the total FAs [92]. A cis-isomer of this acid was also detected in some coral species.

For the first time, hard corals displayed traces of unsaturated very-long-chain FAs with 24 carbon atoms, found in three hermatypic coral species from the *Poritidae* genus in Vietnam [103]. Their structures were authenticated using GC-MS. A selection of unique acids was found in the gorgonian *Leptogorgia piccolo* from the Atlantic Ocean, Senegal [109]. These acids, all possessing *cis*-configured double bonds, were identified using IR-spectroscopy and GC-MS of the corresponding FA pyrrolidids.

A novel unsaturated fatty acid (FA) with a distinct vicinal dimethyl-branched structure was extracted from the soft coral *Sinularia* sp. found in Okinawa Is., Japan [125]. Through comprehensive analysis using ^1^H- and ^13^C-NMR, high-resolution MS, IR-spectroscopy, and chemical modification methods, this compound’s structure was identified as (E,R)-6,7-dimethylhexadeca-7-enoic acid (**41**). The absolute configuration of its two chiral centers was determined utilizing the Ohrui–Akasaka method. Additionally, the polyunsaturated fatty acid (PUFA) ethyl ester, known as ethyl 5Z,8Z,11Z,14Z-nonadecatetraenoate (**42**), was isolated from the *Nephthea hainansis* soft coral. Spectroscopy methods were employed to ascertain the structure of this ethyl ester [126].

Unique furan acids (F-acids) were discovered in various gorgonian corals, including *Acabaria erithraea*, *Chironephthya variabilis*, *Ellisella plexauroides*, *Menella praelonga*, and *Bebryce studeri* from Vietnam (refer to Appendix A and Table 22) [111]. Among the studied gorgonians, seven distinct F-acids (**43**–**49**) were identified (see Table 22, and Figure 14).

Five of these F-acids possess a furan ring, flanked by an unbranched fatty acid residue with either 9, 11, or 13 carbon atoms on one α-position, while the opposite α-position is occupied by a C_5_H_11_ group. Depending on the specific acid, either both β-positions of the furan ring carried a methyl substituent, or just one β-position near the longer unbranched chain bore a methyl group. The remaining two F-acids have an unbranched fatty acid residue with either 11 or 13 carbon atoms on one α-position and a C_3_H_7_ group on the other α-position, with both of the furan ring’s β-positions containing methyl groups (as detailed in Table 22).

Among these F-acids, the two most concentrated were the 14,17-epoxy-15-methyldocosa-14,16-dienoic acid (Me-F22) (**48**) found in *Menella praelonga*, accounting for 2.0% of the total FAs, and the 14,17-epoxy-15,16- dimethyldocosa-14,16-dienoic acid (diMe-F22) (**49**) found in *Chironephthya variabilis*, constituting 5.1% (refer to Appendix A). The mass spectra of the methyl esters of F-acids (**48**) and (**49**) can be seen in Figure 15.

A short-chain F-acid (**50**) was previously identified in two soft coral species, *Sarcophyton glaucum* and *S. gemmatum*, comprising 0.04% of the coral’s dry weight [127]. This particular compound was not found in *S. decaryi* or the other two examined *Sarcophyton* sp. species. The structure of this F-acid was ascertained using IR and UV spectrometry, as well as ^1^H, ^13^C-NMR, and mass spectrometry.

The gorgonian coral *Bebryce studeri* exhibited a variety of very-long-chain saturated, mono-, and dienoic fatty acids, including 24:0, 24:1, 26:1, and 26:2. Additionally, certain demospongic acids, such as 25:2(5,9) (**51**), 26:2(5,9) (**52**), 26:3(5,9,19) (**53**), and 28:3(5,9,19) (**54**), were also present in its fatty acid composition (refer to Appendix A) [111]. Gas chromatography-mass spectrometry (GC-MS) confirmed the structures of these compounds.

### 2.3. Prostaglandins and Other Oxylipins

Prostanoids encompass prostaglandins and thromboxanes, which are oxidized derivatives of PUFAs containing either a 5- or 6-membered ring. Prostaglandins arise from the action of cyclooxygenase and related enzymes on C20 PUFAs, primarily 20:4n-6. They play pivotal roles in various physiological processes of terrestrial vertebrates. There are other derivatives, closely related to prostaglandins in terms of carbon skeleton structure, that are also classified as prostanoids. Furthermore, all C20 PUFA derivatives, including acyclic ones, are termed eicosanoids. This group includes prostaglandins, thromboxanes, leukotrienes, and lipoxins. Derivatives of C18 PUFAs are referred to as octadecanoids. To denote all PUFA derivatives that have undergone at least one oxidation step, whether cyclic or acyclic, the term “oxylipins” has been suggested. Information on the chemical structure, biological activity, and chemical synthesis of coral oxylipins is available in several reviews [128,129,130,131,132,133,134].

#### 2.3.1. Prostaglandins

After the chemical structure of prostaglandins, which act as hormones in higher animals, was deciphered, a significant source of prostaglandins was discovered in the Caribbean gorgonian coral, *Plexaura homomalla* [135]. This species had prostaglandin esters comprising 2–3% of its dry weight. In the foundational study by Weinheimer and Spraggins, *P. homomalla* from the Florida coast revealed the main substances as 15R-PGA2 (**55**) and its acetate methyl ester 15R-PGA2 (**56**, see Figure 16). Notably, the configuration at C-15 was opposite to that found in mammalian prostaglandins. Subsequently, Light and Samuelsson [136] identified a mix of 15R-PGE2 (**58**), its methyl ester 15R-PGE2 (**59**), and methyl ester 15R-PGA2 (**57**).

*P. homomalla* from other Caribbean locations, such as the Cayman and Bahama Islands, was later found to have the same prostaglandins but with a 15*S* configuration (**60**, **61**)—a commonality among higher animals. Both isomeric forms were seldom found together in coral samples [136,137]. Subsequent analysis of *P. homomalla* var. extracts uncovered an additional compound, identified as (15*S*)-15-hydroxy-9-oxo-5-trans-PGA2 (**62**) [138]. Apart from 15S-PGA2 (**60**), several other compounds such as monoacetate of methyl ester PGF2α (**63**), methyl ester of PGF2α (**64**), PGF2α in its acid form (**65**), and minor quantities of PGE2 (**61**) and 5,6-trans-PGA2 (**62**) were also detected in *P. homomalla* from the Cayman Islands [139]. Traces of 13,14-*cis*-PGA2 (**66**) acetate, methyl ester acetate of 13,14-dihydro-PGA2 (**67**), and the Michael adduct of 13,14-dihydro-PGA2 (**68**) were noted for samples from the Cayman Islands [140].

The R-type *P. homomalla* from Florida produced the methyl ester of 15-acetate-15R-prostaglandin E2 (**69**), constituting 2–10% of the primary component, the methyl ester of 15-acetate prostaglandin A2. The isolated prostaglandin’s structure was discerned through ^1^D and ^2^D ^1^H-NMR, HPLC, and HPLC-MS analyses. Equivalent quantities of the 15S-isomer were found in samples of the S-type *Plexaura homomalla* [141]. Additionally, prostaglandin PGF2α (**70**) was extracted from the gorgonian *Euplexaura erecta* sourced from Japan (Shimoda, Sagami Bay). Its chromatographic mobility on TLC and the mass spectrum of its ester matched that of the standard [142].

The soft coral *Lobophyton depressum*, found in the Red Sea, was identified to have four derivatives of prostaglandin F: the methyl ester of 11-acetate-15(S)-PGF2α (**71**), its 18-acetoxy derivative (**72**), and the corresponding carboxylic acids (**73**) and (**74**) [143]. Two other derivatives, prostaglandin PGB2 (**75**) and its methyl ester (**76**), were discovered in two soft coral species, *Sarcophyton trocheliophorus* and *Lithophyton arboretum*, also from the Red Sea. They were present in amounts of 3.6 and 4.0 μg/g of raw tissue, respectively. Their identification was based on spectral data matching literature values for these substances [144].

In the Vietnamese coral *Lobophytum cornatum*, methyl and ethyl esters of prostaglandins PGA2 were found, along with their degradation products, prostaglandins PGB2 (**75**) [145]. These compounds were identified by comparing them to the retention times of standard samples on HPLC and GC chromatograms, as well as by examining their mass spectra. The presence of ethyl esters is speculated to be a result of coral preservation in ethanol. Additionally, a substance matching the PGA2 methyl ester in UV spectrum and TLC analysis was identified in the cold-water soft coral *Gersemia rubiformis*, located in the Bering Sea near Kamchatka [108].

A rare prostaglandin, the methyl ester of (5Z)-9,15-dioxoprosta-5,8(12)-dien-1-oic acid (**77**), was isolated from the soft coral *Sarcophyton crassocaule* in the Indian Ocean. Spectral analysis, encompassing techniques such as ^1^H- and ^13^C-NMR, DEPT, H-H COSY, C-H COSY, HRMS, and HMBC, confirmed its structure [146].

#### 2.3.2. Cyclic Oxylipins

In 1982, two Japanese research groups [147,148] isolated a new prostanoid series from the soft coral *Clavularia viridis* found in Okinawa Island. Initially named clavulones [147] and claviridenones [148], the term “clavulones” is now predominantly used in literature. Discovered in the 1980s, nine clavulones were categorized into three groups based on their chemical composition. All clavulones possess a cyclopentenone fragment, similar to the one found in prostaglandin A. This fragment contains structures with seven carbon atoms (associated with the carboxyl group) and eight carbon atoms (related to the ω-chain), along with a 12S-acetoxy group. The first set of clavulones, namely clavulone I, II, III, and IV (**78**–**81**, as illustrated in Figure 17), incorporate compounds with a 4R-acetoxy group. These compounds are differentiated by the geometry of their Δ5 and Δ7 double bonds [147,148,149,150].

The second group of clavulones comprises three 20-acetoxy derivatives (**82**–**84**) [151], while the third group features two 4-deacetyle derivatives (85, 86). The structure and absolute configuration of clavulones were determined using a combination of IR, UV, 1H, and ^13^C-NMR analysis, supplemented by circular dichroism spectroscopy. Additionally, the method of fractional chemical modification through the ozonolysis of the Δ5 double bond, and the subsequent analysis of the reaction products, contributed to their structural determination.

Notably, unique halogenated clavulone analogues, termed chlorovulones I–IV (**87**–**90**) [152,153,154], bromovulones (**91**), iodovulones (**92**) [155], and the 10,11-epoxide of chlorovulone I (**93**) [156] were also isolated from the soft coral *C. viridis*. These halogenated and epoxy-prostanoids were identified to possess a 12*R*-configuration, signifying the opposite stereoconfiguration of the anomeric center at the 12th carbon atom in clavulones (Figure 18). Given the structural similarity of these compounds and their shared origin, the biosynthesis mechanism behind these various isomers remains an area of intrigue.

Interestingly, *C. viridis* stands out as the sole species among the *Clavularia* corals from the same region to have detectable prostanoids. Phylogenetic investigations into the host organisms of *Clavularia* soft corals (located in Okinawa Island) revealed that *C. viridis* is genetically distinct from other *Clavularia* species. This distinction is further emphasized by the absence of prostanoid biosynthesis in these other species, as indicated in the molecular genetic dendrogram [157].

Subsequent studies into lesser-known prostanoids from *C. viridis* led to the isolation of five novel 7-acetoxy derivatives of chlorovulones (**94**–**98**), as reported by Watanabe et al. in 2001. The structures of these new compounds were primarily determined using NMR data. From 220 g of dehydrated cool coral tissue, the yields were as follows: (**94**) yielded 29.8 mg, (**95**) produced 2.6 mg, (**96**) resulted in 1.1 mg, (**97**) amounted to 0.3 mg, and (**98**) gave a scant 0.1 mg. Moreover, 15 new halogenated prostanoids were discovered in *C. viridis* as minor constituents. These included three fresh iodovulones, seven 12-O-acetyl derivatives encompassing iodo-, bromo-, and chlorovulones, along with five 10,11-epoxy derivatives of the aforementioned iodo-, bromo-, and chlorovulones [158,159]. Spectral methods and chemical transformations were employed to authenticate the structures of these compounds.

Prostanoids that are believed to be potential biosynthetic precursors of clavulones have been identified in *C. viridis* [159]. Notably, the methyl ester of preclavulon A (**99**), which is a crucial intermediary in clavulones synthesis, was isolated from a natural source for the first time, alongside its stereoisomer (**100**) and another presumed biosynthetic intermediary (**101**) situated between preclavulon A and clavulones. The structural makeup of these three compounds was ascertained through spectroscopic data. From 220 g of dehydrated cool coral tissue, 2.8 mg of preclavulon A, 0.6 mg of its stereoisomer, and 1.2 mg of the intermediate were extracted.

Shen and colleagues [160] identified seven new clavulones in *Clavularia viridis* sourced from Taiwan. These include 4-deacetoxyl-12-O-deacetylclavulon I (**102**), 4-deacetoxyl-12-O-deacetylclavulon II (**103**), bromovulon II (**104**), iodovulon II (**105**), 4-deacetoxyl-12-O-deacetylclavulon III (**106**), bromovulon III (**107**), and iodovulon III (**108**), with amounts of 1.1, 2.6, 3.1, 9.1, 2.2, 25, and 4.4 mg respectively.

From the soft coral *C. viridis*, two γ-lactones derived from clavulones II and III, named clavulactones II (**109**) and III (**110**), were isolated. These compounds possess a lactone ring in the α-chain [161]. Subsequently, clavulacton I (**111**) and 17,18-dehydroclavulacton I (**112**) were characterized by Iwashima and his team [162]. Lactones of preclavulones I (**113**) and II (**114**) were also identified. These lactones stand out as the first natural prostanoids whose side chains’ absolute configuration differs from the configuration seen in vertebrate prostaglandins [163]. From 3.3 kg of raw *C. viridis* coral tissue, 2.1 mg of preclavulone I lactones and 0.6 mg of preclavulone II lactones were extracted.

Most 4-acetoxy derivatives of clavulones had an R configuration. However, a minor quantity of 4(*S*) epimers of clavulones, specifically 4-epiclavulone II (**115**) and 4-epiclavulone III (**116**), were also isolated. In this context, the content of these compounds was 4.9 mg and 0.9 mg for every 7 kg of wet tissue, respectively. In the same study, the yields of clavulone II (**79**) and clavulone III (**80**) were found to be 1090 mg and 4242 mg, respectively [164].

Beyond the typical prostanoids with 20 carbon atoms, two C15 aldehydes named clavirins were identified in *C. viridis* [165]. These are likely the products of oxidative degradation of clavulones I and III (or II and IV) at the Δ5 double bond in their main carbon chain. Clavirins I (**117**) and II (**118**) were obtained in quantities of 2.2 and 2.5 mg from 6.9 kg of coral, respectively. Further study of minor components from *C. viridis* led to the discovery of five clavulones as free acids, termed claviridic acids A–E (**119**–**123**, Figure 18) [166]. The respective yields of claviridic acids A through E were 8, 8, 12, 15, and 11 mg from 6.5 kg of the soft coral.

New oxylipins, namely tricycloclavulone (**124**) and clavubicyclone (**125**), were isolated from *C. viridis* [167]. Tricycloclavulone’s structure features a tricyclo [5.3.0.0(1,4)] decane ring system, while clavubicyclone has a bicyclo [3.2.1] octane ring system.

In 1985, Baker and colleagues [168] isolated a new series of 10-chloro-substituted prostanoids from the soft azooxanthellae coral *Telesto riisei*, naming them punoglandins (**126**–**132**). This marked the first discovery of natural halogenated prostanoids. The synthesis and specific stereochemistry of the substituent groups at the 5, 6, and 12 carbon atoms of a collection of punoglandins were later detailed [153,154,169,170]. Notably, punoglandins 2 (**127**), 4 (**129**), and 6 (**132**) (arachidonic acid derivatives) and their 17,18-dehydro counterparts, punoglandins 1 (**126**), 3 (**128**), and 5 (**131**) (eicosapentaenoic acid derivatives), along with the *Z*-isomer of punoglandin 4 (**130**), were the main components among the 19 punoglandins identified [171,172].

Carijenone (**133**), extracted from the octocoral *Carijoa multiflora* (= *Telesto multiflora*) [173], introduces a new class of bicyclic prostanoids. Its carbon structure mirrors that of punoglandin acetate, but with an added oxane ring. IR, UV, 1H-NMR, 13C-NMR, and mass spectrometry determined the carijenone structure, which is analogous to a possible condensation product of punoglandin 2 found in another species from the same genus, the soft coral *Telesto riisei* [171].

Lastly, eight novel brominated oxylipins, inclusive of two glycosides, were extracted from the Red Sea corals *Dendronephthya* sp. (in both red and yellow morphs) (**134**–**137**), *Tubipora musica* (**138**,**139**), and *Dendrophyllia* sp. (**140**,**141**) [174].

#### 2.3.3. Acyclic Oxilipins

Two novel 10-hydroxy-docosapolyenoic acids, namely (10R,7Z,11E,13E,16Z,19Z)-10-hydroxydocosa-7,11,13,16,19-pentaenoic acid (**142**) and (10R,4Z,7Z,11E,13Z,16Z,19Z)-10-hydroxydocosa-4,7,11,13,16,19-hexaenoic acid (**143**, as depicted in Figure 19), were extracted in the form of methyl esters. This was achieved following treatment with ethanolic extract diazomethane from the coral Scleractinia *Madrepora oculata*. This specimen was gathered near Saint-Paul Island in the southern part of the Indian Ocean at a depth of 290 m [175]. From the same deep-sea coral species, collected in the Norwegian Sea, acid (142) and the methyl ester of the new compound (10R,7Z,11E,13Z,16Z)-10-hydroxydocosa-7,11,13,16-tetraenoic acid (**144**) were also isolated. The structural composition of these newfound compounds was discerned using a combination of spectrometric methods. Additionally, (10R,7Z,11E,13Z,16Z,19Z)-10-hydroxydocosa-7,11,13,16,19-pentaenoic acid (**145**) and other compounds, namely (5Z,9E,11Z,14Z,17Z)-8-hydroxyeicosa-5,9,11,14,17-pentaenoic acid (8-HEPE) (**146**) and (5Z,9E,11Z,14Z)-8-hydroxyeicosa-5,9,11,14-tetraenoic acid (8-HETE) (**147**), were extracted from the deep-sea coral Scleractinia *M. oculata*. Notably, these acids were previously identified in the antipatharia (black coral) species *Leiopathes glaberrima* [176].

The cold-water soft coral *Gersemia rubiformis*, sourced from Kamchatka in the Bering Sea, demonstrated the presence of 8-hydroxy-eicosatetraenoic acid (8-HETE) (**147**) and traces of 8-hydroxy-eicosapentaemoic acid (8-HEPE) (**146**). The latter may act as an intermediate in the prostaglandin biosynthesis via the lipoxygenase pathway [177]. A new oxylipin, 15-hydroxytetracosa-6,9,12,16,18-pentaenoic acid (15-HTPE) (**148**), as well as known oxylipins such as 11-hydroxyeicosa-5,8,12,14-tetraenoic acid (11-HETE) (**149**) and 9-hydroxyoctadeca-6,10,12-trienoic acid (9-HOTE) (**150**), were extracted from the soft coral *Sinularia numerosa*, which was collected in southern Japan [178].

In the Caribbean gorgonian coral *Plexaurella dichotoma*, the R-isomer of 11-HETE (11-R-hydroxy-5Z,8Z,12E,14Z-eicosatetraenoic acid) (**149**) was identified [9,10,11]. Notably, prostaglandins identified in *Plexaura homomalla* from the Caribbean differed from those in *Plexaurella dichotoma*. This leads to the conclusion that 11-HETE biosynthesis in Caribbean gorgonians is not always followed by prostaglandin formation. A γ-lactone (**151**), derived from docosahexaenoic acid 22:6n-3, was discovered in the zooxanthellae JCUSG-1 culture (clade A2). This culture was isolated from soft coral and the compound was subsequently termed zooxanthellactone [179]. The concentration of this oxylipin ranged from 0.28 to 0.39 mg/g of the raw cell weight, depending on environmental conditions.

## 3. Biosynthesis of FA in Corals

One primary pathway for PUFA biosynthesis involves the synergistic activity of two enzyme types: desaturases and elongases [180,181]. Desaturases convert saturated fatty acids (FAs) into monounsaturated FAs, which are then transformed into PUFAs. Each desaturase introduces a single double bond into the FA carbon chain, with the carbon atom number indicating where this bond forms included in the desaturase’s name. In contrast, elongases extend the FA carbon chain by two carbon atoms.

Typically, the synthesis of unsaturated FAs in both plants and animals commences with the introduction of the first double bond at the Δ9 position of saturated acids 16:0 and 18:0. Interestingly, while animals dehydrogenate an intermediate derivative of the FA itself, plants dehydrogenate the FA while it is part of a phosphatidylcholine molecule. In plants, the presence of Δ12 and Δ15 desaturases is essential for the biosynthesis of acids 18:2n-6 and 18:3n-3. These acids serve as precursors for the most prevalent PUFAs in the n-6 and n-3 series [182] (see Figure 20):

Animals, with a few exceptions, lack desaturases capable of introducing additional double bonds into the FA carbon chain beyond the Δ9 position. To continue PUFA biosynthesis, animals derive 18:2n-6 and 18:3n-3 (or 18:3n-6 and 18:4n-3) from plant sources [183]. Their capacity to further elongate and desaturate PUFA varies significantly [184,185]. Higher animals, including humans, often obtain key PUFAs from their diet due to an absence or limited capacity for their biosynthesis. These PUFAs are termed essential fatty acids.

A significant number of hard coral species and a substantial portion of soft coral species, especially alcyonarians rather than gorgonians, harbor intracellular symbiotic microalgae known as zooxanthellae, which belong to the Simbiodinium group. The host (polyp) is an animal, while its endosymbionts are plant-like. FA transformation pathways differ between plants and animals, leading to distinct PUFA profiles. Estimates suggest that zooxanthellae-derived FAs can constitute up to 30% of the total FAs in a coral colony. The overall FAs extracted from a symbiotic coral colony represent a blend of both plant and animal origins. All corals are heterotrophic and ingest both plant- and animal-derived (zooplankton) FAs. Strong evidence indicates a transfer of FAs from zooxanthellae to their host. These FAs can be further metabolized by the host, yet these pathways remain largely unexplored in corals. The impact of zooxanthellae on the lipid composition of the coral is so profound that the FA profiles of coral genera with zooxanthellae from various orders more closely resemble each other than those of genera within the same order but with differing symbiont levels.

When examining FA biosynthesis pathways in corals, it is essential to account for several factors:Taxonomic classification (i.e., hexacorals, octocorals, hydrocorals, etc.)The presence of zooxanthellae and the coexistence of plant and animal biosynthetic mechanisms within the same colony.The potential for FA transfer between zooxanthellae and the host organism.Dependence on heterotrophic nutrition.The host organism’s capacity to influence PUFA biosynthesis in its symbionts.The presence of associated organisms.

In-depth studies on PUFA biosynthesis in corals have yet to be conducted. Currently, there is no information on the composition and characteristics of enzymes responsible for PUFA synthesis in corals. However, to understand the unique fatty acid composition of corals, one can draw from established knowledge on PUFA biosynthesis in other animals and plants. A comprehensive database detailing the fatty acid composition of various corals, alongside comparative research on the fatty acid compositions of hard versus soft corals, corals with and without zooxanthellae, pure tissue samples from the host organisms, and their symbionts, would provide invaluable insights into the processes of PUFA biosynthesis in corals.

### Features of PUFA Biosynthesis in Corals

The distribution of PUFAs in whole colonies of both reef-building and soft corals, whether or not they contain zooxanthellae, is summarized as follows based on various studies [68,79,92,103,105,111]:In total FAs of hard corals with zooxanthellae, there is a broad spectrum of typical unsaturated C16-22 FAs. This includes monounsaturated fats such as 16:1n-7, 18:1n-9, as well as various isomers of 20:1 and 22:1. PUFAs of the n-6 series (ranging from 18:2n-6 to 22:4n-6) and the n-3 series (from 18:4n-3 to 22:6n-3) are also present. These coral species showed a minimal presence of 20:4n-3 and an almost negligible amount of 22:5n-6.Hard corals that lack zooxanthellae differ from their symbiotic counterparts. They exhibit high levels of 22:5n-3 and reduced amounts of 22:6n-3, 18:3n-6, and 18:4n-3.Soft corals with zooxanthellae showcase additional FAs in their composition. This includes C16 PUFAs such as 16:2n-7, 16:3n-4, and 16:4n-1, as well as 18:2n-7 and very-long-chain PUFAs such as 24:5n-6 and 24:6n-3. These corals also contain small amounts of both 22:5n-6 and 22:5n-3. An inverse correlation between the levels of 16:2n-7 and 18:3n-6 has been observed.Soft corals without zooxanthellae, when compared to their symbiotic counterparts, are characterized by higher concentrations of 22:5n-3, 22:5n-6, and 24:5n-6. In contrast, they display significantly lower levels of 16:2n-7, 18:3n-6, and 18:4n-3.

The composition of total lipids in an entire coral colony, their symbionts, and the polyp tissues displayed similar PUFA varieties. However, the quantities of these acids varied significantly, as shown in Figure 21 [70].

*In hard corals*: The primary FAs in zooxanthellae were 18:3n-6, 18:4n-3, 20:5n-3, and 22:6n-3. The dominant FAs in the polyp tissues were 20:4n-6, 22:4n-6, and 22:5n-3.

*In soft corals*: The primary FAs of zooxanthellae consisted of 18:4n-3, 20:5n-3, 22:6n-3, and a variety of C16 PUFAs. The main FAs in the polyp tissues were 20:4n-6, 16:2n-7, 18:2n-7, and 24:5n-6. The exception is 16:2n-7, the high level of which in one partner of the symbiont–host association corresponded to the low level of this component in the other partner. The entire coral colony held a median position, with PUFA levels that fell between those of the plant and animal components of the symbiotic organism, as depicted in Figure 21.

Consequently, a range of C18-22 PUFAs from the n-6 series was identified in corals. The biosynthesis of these acids can be depicted by a typical sequence of associated desaturases and elongases. Specifically, this involves the Δ4, Δ5, Δ6, Δ12, and Δ15 desaturases, as well as the C2-elongase (E). This sequence aligns with the contemporary understanding as referenced in [186] (Figure 22).

The C18-22 PUFAs of the n-3 series follow a similar sequence (Figure 23):

Hard corals lack the ability to synthesize fatty acids (FA) with more than 22 carbon atoms in their chain. However, soft corals can produce very-long-chain tetracosapolyenoic acids (TPAs) from both the n-6 and n-3 series, specifically 24:5n-6 and 24:6n-3. To describe the synthesis of these TPAs in certain cells of higher animals, the following model has been proposed [187,188] (Figure 24):

Most elongases are highly specific to the substrate FA carbon chain length. Only a select few can extend the FA carbon chain regardless of its original length [185,188]. It is clear that hard corals lack the elongases required to extend C22 FAs to C24 FAs. However, soft corals present an intriguing source of these specific elongases.

The existence of TPA serves as a chemotaxonomic marker for soft corals [111]. This highlights a fundamental distinction in PUFA biosynthesis between octocorals and hexacorals. TPA has been found in every soft coral species, irrespective of the presence of zooxanthellae [92]. This suggests that the biosynthesis of TPA from C22 PUFA occurs within the host (in the animal tissues) and operates independently from the symbiotic microalgae, zooxanthellae.

C18-22 PUFAs from the n-3 series are predominantly found in zooxanthellae, while C20-22 PUFAs from the n-6 series are more abundant in polyp tissues (refer to Figure 8). For instance, the soft coral *Clavularia viridis* has been shown to contain 20:5n-3, which is found exclusively in the zooxanthellae, and 20:4n-6, which is found only in the polyp tissues [120]. Based on this, it is hypothesized that the biosynthesis of C18-22 PUFAs from the n-3 series mainly occurs in zooxanthellae, while the synthesis of C20-22 PUFAs from the n-6 series primarily takes place in the coral’s polyp tissues. This supposition aligns with contemporary understanding of distinct PUFA biosynthesis pathways in animals and plants [186,189]. It is further supported by data on FA composition found in hard coral zooxanthellae [70,112,115,116,121,134].

Combining the fundamental principles of PUFA biosynthesis with data on FA distribution in corals, we can suggest the following pathway for n-6 series PUFA synthesis in hard coral polyps (Figure 25):

Food lipids can serve as the source of the precursor acid 18:2n-6 for hard coral polyps that lack zooxanthellae. For polyps of hard corals with zooxanthellae, these symbiotic algae, transferring 18:2n-6 and 18:3n-6 to the polyps, can provide the initial acids for n-6 series PUFA synthesis. Given the virtual absence of 22:5n-6 in the FA of hard corals, irrespective of zooxanthellae presence, it suggests that these coral polyps lack Δ4 desaturase, which would convert 22:4n-6 into 22:5n-6. Although zooxanthellae possess Δ4 desaturase, as evidenced by their ability to synthesize 22:6n-3 from 22:5n-3, the sites of FA synthesis between zooxanthellae and polyps seem spatially distinct. This indicates that either 22:5n-6 is not transferred from polyps to zooxanthellae, or it is not utilized as a substrate by the enzymes of the zooxanthellae.

For non-symbiotic hard coral species, which lack zooxanthellae capable of producing 22:6n-3, there is a marked reduction in the level of 22:6n-3 compared to reef-building corals. Conversely, these corals have a considerably higher 20:5n-3 level, which could be attributed to a greater intake of phytoplankton in their diet [44,183,190]. These corals also exhibit elevated levels of 22:5n-3, suggesting that an increased concentration of 20:5n-3 might promote the production of 22:5n-3 through the elongation of 20:5n-3.

The soft coral species *Clavularia* sp. stands out as an anomaly; it is one of the few studied species in which not only 22:4n-3 is detected but also the rare TPAs of the n-3 series: 24:4n-3 and 24:5n-3 [79]. This suggests a unique biosynthesis pathway for n-3 series C22-C24 acids in *Clavularia* sp. (Figure 26):

Japanese researchers have identified a wide array of oxidized PUFA derivatives in one species of the *Clavularia genus*, known as clavulons. These compounds are analogous to prostaglandins found in terrestrial animals [120]. It is conceivable that the presence of these rare TPAs and oxylipins in alcyonarians of this genus may be interconnected.

The primary lipid markers of zooxanthellae are PUFAs of the n-3 series. Drawing from data on the fatty acid composition of zooxanthellae and established knowledge on fatty acid biosynthesis in plants, we can hypothesize the following pathway for n-3 series PUFA synthesis in the zooxanthellae of hard corals, involving the Δ4, Δ5, Δ6, Δ12, and Δ15 desaturases (Figure 27):

The n-6 series PUFAs are also found in the zooxanthellae of hard corals. The Δ6 desaturase in these zooxanthellae can participate in the synthesis of 18:3n-6, one of the lipid markers for these symbionts. The 18:3n-6 acid acts as a precursor for the n-6 series C20–22 PUFAs. Thus, there are two potential sources of n-6 series PUFAs in hard coral zooxanthellae: their own synthesis and a transfer from the host organism. The absence of 22:5n-6 in zooxanthellae is intriguing, as this acid could be derived from 22:4n-6 through the Δ4 desaturase, analogous to how 22:6n-3 is produced from 22:5n-3.

The total fatty acids of symbiotic coral species and hydrocorals include the uncommon acid 18:5n-3, which is produced by zooxanthellae [112]. It is worth noting that 18:5n-3 acid undergoes isomerization under alkaline conditions [191,192], leading to its non-identification by researchers who employed alkaline agents in microalgae FA analysis. Historically, 18:5n-3 was identified in free-living dinoflagellates and proposed as a marker for this group of algae [43,193]. Joseph [43] postulated that 18:5n-3 might be produced from 18:4n-3 through the action of Δ3-desaturase or via the β-oxidation of 20:5n-3. The biosynthesis route of 18:5n-3 was studied in *Prymnesium parvum* (Haptophyceae), which was nourished with labeled fatty acids [194]. The addition of labeled 18:2n-6 and 18:3n-3 resulted in the creation of labeled 18:4n-3, 18:5n-3, and 20:5n-3. In contrast, the introduction of labeled 20:5n-3 led to the production of labeled 22:5n-3 and 22:6n-3. This suggests that 18:5n-3 might be synthesized through a Δ3-desaturase, which incorporates a Δ3 double bond into 18:4n-3.

The fatty acid (FA) synthesis in soft coral polyps extends beyond the production of polyunsaturated fatty acids (PUFAs) with 22 carbon atoms, which is the limit in hard coral polyps. Soft coral polyps have the capability to synthesize tetracosapolyenoic acids (TPAs) with 24 carbon atoms in their carbon chain. In addition to this unique synthesis capability, soft corals possess another distinct feature: the FAs of non-symbiotic soft corals, encompassing both alcyonarians and gorgonians, include both 22:5n-6 and 22:5n-3. While these C22 PUFAs are also present in soft corals with zooxanthellae, their concentrations are typically lower.

To describe TPA biosynthesis in soft corals, drawing parallels with Sprecher’s model for higher animals [187,188], it can be inferred that 24:5n-6 and 24:6n-3 are derived from 22:4n-6 and 22:5n-3, respectively. This transformation is achieved through the sequential actions of elongases and the Δ6 desaturase (Figure 28):

Nonetheless, the aforementioned scheme presumes the existence of two intermediate compounds—24:4n-6 and 24:5n-3. Interestingly, these were not identified within the fatty acid profiles of soft corals. Given that a majority of the studied soft coral species contained the fatty acids 22:5n-6 and 22:5n-3, an alternative biosynthetic pathway for TPA in soft corals can be proposed. This pathway would involve the actions of Δ4 desaturase and elongase during the final two stages (Figure 29):

This proposed scheme supports the typical sequential action of desaturases and elongases, clarifying the presence of 22:5n-6 and 22:5n-3, as well as the absence of 24:4n-6 and 24:5n-3, in the fatty acids of soft corals. A notable observation was made in certain colonies of various soft coral species: a significant increase (by an order of magnitude) in the concentration of 22:4n-6, which was paralleled by a corresponding decrease in 24:5n-6 levels [111]. Yet, no significant variations in concentrations of analogous acids of the n-3 series, specifically 22:5n-3 and 24:6n-3, were observed. Given that 22:4n-6 serves as a biosynthetic precursor to 24:5n-6, this observation was attributed to the inhibition of the elongase responsible for the transition from 22:4n-6 to 24:4n-6, as per Sprecher’s scheme [187]. This rationale inherently implies the existence of two distinct C22 → C24 elongases in soft corals, each specific to the PUFA of either the n-6 or n-3 series. According to this assumption, the elongase for the n-6 series is inhibited, while the other remains unaffected, which seems improbable.

Presently, our hypothesis is that the aforementioned observation can be attributed to the inhibition of the Δ4 desaturase. In this scenario, both transitions—22:4n-6 to 22:5n-6 and 22:5n-3 to 22:6n-3—would be simultaneously impeded. However, unlike 22:5n-6, corals can readily obtain 22:6n-3 from dietary sources. This acid, 22:6n-3, serves as a substrate for the elongase, ensuring consistent levels of 24:6n-3. Concurrently, the inhibition of Δ4 desaturase results in an accumulation of 22:4n-6, which is not further converted to 24:5n-6, leading to a marked reduction in 24:5n-6 levels (Figure 30).

A significant presence of 22:5n-6 and an elevated level (relative to corals) of 22:6n-3 in the fatty acids (FAs) of the hydrocoral *Millepora* can likely be attributed to the active functioning of the Δ4 desaturase, facilitating transitions from 22:4n-6 to 22:5n-6 and from 22:5n-3 to 22:6n-3. Consequently, chemotaxonomic differences in PUFA composition among hard corals, hydrocorals, and soft corals can be ascribed to the absence of Δ4 desaturase in reef-building corals, its presence in *Millepora hydrocorals*, and the existence of two enzymes (Δ4 desaturase and C22 → C24 elongase) in soft corals. Further research is necessary to validate this hypothesis.

Distinct variations in the composition of C16 PUFAs were observed between soft and hard corals (refer to Figure 8). A noticeable presence of PUFAs with 16 carbon atoms, exhibiting varying degrees of unsaturation (such as 16:2n-7, 16:3n-4, and 16:4n-1), was identified in the zooxanthellae of soft corals. It is hypothesized that the synthesis of C16 PUFAs occurs within the zooxanthellae of soft corals. This synthesis can be envisioned as a result of the sequential actions of Δ6, Δ12, and Δ15 desaturases acting on the acid 16:1n-7 (Figure 31):

Undoubtedly, the potential pathway exists in hard coral zooxanthellae, as evidenced by the traces of 16:2n-7 and 16:3n-4 found in *Acropora zooxanthellae*. Marine green macroalgae contain a substantial quantity of C16 PUFAs. Still, their biosynthesis follows the scheme 16:1n-9 → 16:2n-6 → 16:3n-3 → 16:4n-3 [195]. Given the discovery of the acid 16:2n-7 in both symbionts and the host (as noted in Appendix A), it is plausible that the synthesis of 16:2n-7 in soft corals with zooxanthellae occurs with the participation of polyp tissues. Moreover, the concentration of 18:2n-7 in the polyp tissues of soft corals is significantly higher than that in zooxanthellae (see Figure 8). Hence, the elongation of 16:2n-7, leading to the formation of 18:2n-7, likely takes place in the polyp tissues of soft corals.

Significant quantities of 18:3n-6 and 16:2n-7 have been found in symbiotic coral species. Zooxanthellae are the primary source of 18:3n-6 and C16 PUFAs in corals. Hard coral zooxanthellae exhibit high levels of 18:3n-6 and low levels of C16 PUFAs. Similarly, certain soft coral species have low C16 PUFA levels and high 18:3n-6 levels. However, some soft coral species have a notable quantity of C16 PUFAs but low 18:3n-6 levels. A reverse correlation between the contents of 18:3n-6 and 16:2n-7 was observed in most symbiotic soft coral species [103].

Crucially, 18:3n-6 and 16:2n-7 are derived from 18:2n-6 and 16:1n-7, respectively, through the action of the Δ6 desaturase. When one enzyme acts on two distinct substrates, the enzyme’s product composition hinges on the availability of each substrate [103] (Figure 32).

Different coral species are believed to provide varying substrates (18:2n-6 or 16:1n-7) for Δ6 desaturase, suggesting that the host organism influences, or even governs, PUFA biosynthesis in zooxanthellae. This influence by the host organism might account for the diverse ratios observed between 18:3n-6 and C16 PUFA in zooxanthellae across different taxonomic groups of corals [122].

Hydrocorals, particularly those of the genus *Millepora*, are common inhabitants of coral reef communities. While they belong to a distinct cnidarian class separate from corals, their fatty acid (FA) composition is notably different from that of both hard and soft corals [79]. Lipids from intact *Millepora* colonies and their host tissues have an exceedingly low concentration of 18:2n-6 and 16:1n-7, which could serve as substrates for Δ6 desaturase. Consequently, the lipids of zooxanthellae from *Millepora* virtually lack 18:3n-6 and C16 PUFAs. We theorize that *Millepora influences* the PUFA composition of zooxanthellae through the transfer of PUFAs from the host organism, especially 22:5n-6, which is a distinctive marker for hydrocorals.

The various FA synthesis processes discussed above can be consolidated into a comprehensive diagram illustrating the anticipated PUFA biosynthesis pathways in corals. This overview should consider certain PUFAs obtained from food sources, as well as the potential transfer of key PUFAs from symbionts to the host organism (refer to Figure 33).

It can be inferred that the distinct patterns of PUFAs, which led to the observed chemotaxonomic differentiation between specific genera and coral families, are a result of genetic differences in their PUFA biosynthesis enzyme activity. For instance, the distinction between the *Sinularia* genus and the *Sarcophyton* genus corals was largely attributed to n-6 series PUFAs. This might be a consequence of metabolic differences in this crucial fatty acid group [105]. Concurrently, variations in PUFA content among different taxa are often linked to the presence or lack of symbionts. As an illustration, a high concentration of TPAs (up to 16% 24:5n-6; up to 7% 24:6n-3) was a primary factor differentiating azooxanthellate corals of the *Dendronephthya* genus from soft corals with zooxanthellae [79]. It is probable that the elevated TPA levels in soft corals without zooxanthellae are not due to increased TPA biosynthesis, but rather the absence of zooxanthellae PUFAs. This absence reduces the relative percentage of TPAs in the total fatty acids of symbiotic soft coral species.

## 4. Chemotaxonomic Significance, Markers of Symbionts, and Food Sources in Corals

The fatty acid (FA) composition is a primary characteristic of lipids. FAs have been effectively utilized for the chemosystematics of various intricate taxonomic groups, including bacteria, vibrios, fungi, and micro- and macroalgae [183,195,196,197,198]. The FA composition is determined by a species’ genetically encoded ability to biosynthesize a particular set of FAs. However, some FAs are introduced to the organism through diet, meaning that the FA composition can also depend on dietary intake and environmental factors. Consequently, significant variations in FA composition can occur even among individual animals of the same species.

Corals are polytrophic in nature and typically house symbiotic microalgae, known as zooxanthellae. These symbionts significantly influence the overall FA composition of the coral. The versatility of corals in selecting food sources, combined with the influence of environmental factors such as habitat depth, seasonal water temperature shifts, and light conditions, can obscure the differences in FA composition across different taxonomic groups of these organisms (Figure 34).

Prior to 2005, the available information on coral FA composition was markedly sparse, sourced from various analytical methods that rendered cross-comparisons virtually impossible. Consequently, it was believed that FAs should not be employed as markers in coral chemotaxonomy [199,200,201]. Instead, sterols were suggested as potential chemotaxonomic indicators for corals [36]. However, detailed findings emerged about the relationship between a coral’s taxonomic classification and its FA composition. Notable FA differences among individual soft coral species and genera were recorded [45,104,202]. Tetracosapolyenoic FAs, specifically 24:5n-6 and 24:6n-3, were identified in the total FA composition of members from all four octocoral subclasses: Alcyonaria, Gorgonaria, Helioporida, and Pennatularia (Anthozoa: Octacorallia). In contrast, these FAs were absent in hexacorals (Anthozoa: Hexacorallia) [203].

An analysis comparing 14 hard coral species from the Indo-Pacific region, which fall under the Anthozoa and Hydrozoa classes, highlighted that certain coral families had a significant presence of specific FAs: Pocilloporidae contained 20:3n-6; Acroporidae had 18:3n-6, 18:4n-3, and 22:4n-6; and Poritidae had 18:3n-6 [100]. Subsequent research refined these findings, identifying distinct FA sets for several families [204] (Figure 35). For instance, the hydrocoral family Milleporidae (Hydrozoa) had a unique FA composition that significantly differed from that of genuine corals.

Statistical multivariate analysis, particularly principal component analysis (PCA), was effectively applied to differentiate soft corals based on their FA composition [105]. This analysis incorporated data from cnidarians spanning 56 species, 16 genera, and 6 families. The results showed distinct clusters, with certain genera such as *Sinularia*, *Lobophytum*, and *Sarcophyton* sharing a central region (Figure 35). This central grouping predominantly included corals that hosted zooxanthellae, suggesting a correlation between their FA composition and the presence of these symbiotic microorganisms. Notably, certain coral genera were segregated based on their zooxanthellae status.

Antipatirians, or black corals, which belong to the subclass Ceriantipatharia (Anthozoa) and possess a hard skeleton, were distinctly separated from soft corals of other subclasses when using total FAs as variables for PCA (see Figure 35).

Analysis revealed that the greatest contributors to the differentiation of soft coral taxa were C18-22 PUFAs, TPAs (24:6n-3 and 24:5n-6), specific dienoic FAs, notably 16:2n-7 and 18:2n-7, and a variety of branched unsaturated and monoenoic FAs with an odd number of carbon atoms [105]. It is understood that the composition of total lipids in animals varies less than the fatty acid composition of these lipids. Indeed, the total lipid compositions of animal species within the same genus are remarkably similar. The correlation between the overall lipid composition and the taxonomic placement of corals has been recently discerned.

For PCA analysis, data on the content of six primary lipid classes from 51 hard coral samples spanning 21 genera (10 families) and 58 soft coral samples (including 39 alcyonarians and 19 gorgonians) across 21 genera (12 families) were used [68]. Among the studied species, two hard coral genera (*Balanophyllia* and *Tubastrea*), one alcyonarian genus (*Dendronephthya*), six gorgonian genera (*Annella*, *Melithaea*, *Parisis*, *Menella*, *Paracis*, and *Viminella*), and *Siphonogorgia* were devoid of zooxanthellae. In contrast, other coral genera had zooxanthellae.

The average lipid class content for the six coral groups (namely hard corals, alcyonarians, and gorgonians, further categorized by the presence or absence of zooxanthellae) is illustrated in Figure 12 [68]. In hard corals, the levels of polar lipids (PLs), sterols (STs), and monoalkyldiacylglycerols (MADAGs) were consistently higher, whereas the content of triglycerols and waxes was lower compared to that in soft corals. On average, species with zooxanthellae exhibited significantly fewer structural lipids (PLs + STs) and a higher content of MADAGs compared to those without zooxanthellae (Figure 36).

PCA revealed that the lipid composition of hard corals (hexacorals) and soft corals (octocorals) was distinctly influenced by their taxonomic classification (as depicted in Figure 37, line A–A). Conversely, within these two taxonomic groups, the lipid composition was predominantly influenced by the presence or absence of zooxanthellae (as seen in Figure 37, line B–B) [68]. Hard corals and zooxanthellate alcyonarians had a higher concentration of reserve lipids, notably waxes, compared to their counterparts without zooxanthellae. These findings suggest that the presence of zooxanthellae promotes the accumulation of this specific lipid class in corals.

The presence or absence of zooxanthellae primarily determines the characteristics of the overall lipid composition in corals at the genus and family levels. Only at the class level does the taxonomic attribute of the coral become the predominant factor influencing the proportion between lipid classes, regardless of the presence or absence of zooxanthellae [68].

### 4.1. Fatty Acids in Chemotaxonomy of Hard Corals

Consistent data collection on the FA composition of hard corals, using standardized modern analytical methods, enabled us to assess the feasibility of using FAs as chemotaxonomic markers for hard corals through statistical approaches. The most insightful outcomes were derived not from the full FA matrix but from information on the content of five primary PUFAs from the n-6 series (18:3n-6, 20:3n-6, 20:4n-6, 22:4n-6, 22:5n-6) and five from the n-3 series (18:4n-3, 20:4n-3, 20:5n-3, 22:5n-3, 22:6n-3) [103].

These data, collected from 25 samples of 26 hard and hydrocoral species spanning 8 families from Vietnam and Seychelles, underwent statistical analysis using multidimensional scale analysis (MSA). Three distinct families—Milleporidae, Dendrophylliidae, and Faviidae—each occupied separate areas in the two-dimensional space outlined by axes 1 and 2, as depicted in Figure 38 [103]. Notably, the family Dendrophylliidae was represented solely by Tubastrea corals, which are devoid of zooxanthellae. The central region comprised three interwoven areas, demarcating three other coral families: Acroporidae, Poritidae, and Pocilloporidae. Lone species from the Pectiniidae and Fungiidae families also occupied the central region. However, their specific distribution remained ambiguous due to the limited sample size.

Along axis 1, defined primarily by the acids 22:4n-6, 20:3n-6, 20:5n-3, and 20:4n-3, distinct separations occurred between families and even within certain genera. Axis 2, influenced mainly by the acids 22:5n-6 and 22:6n-3, delineated families and some genera within them.

In conclusion, the PUFA composition can serve as a viable tool for the chemotaxonomy of hard corals at the family level. Researchers propose that each coral family hosts a specific taxonomic group of zooxanthellae, each possessing a unique set of PUFAs. This distinct PUFA composition subsequently impacts the overall PUFA composition of the entire coral colony [103].

The relationship between hard coral FA composition and their taxonomic classification was explored using 45 samples from the Acroporidae, Faviidae, Fungiidae, and Poritidae families. To minimize the environmental influence on FA composition, all samples were collected simultaneously, from the same region, and at identical depths [68]. Past research utilized data on the contents of 10 primary PUFAs for statistical analysis [103]. The ANOVA test revealed insignificant differences (*p* > 0.05) in three PUFAs (20:4n-6, 18:3n-6, and 18:4n-3) but observed significant variations in four C18–20 monoenoic FAs. Additionally, 22:5n-6 was absent in corals from these families. Consequently, nine FAs (18:1n-9, 18:1n-7, 20:1n-9, 20:1n-7, 20:4n-3, 20:5n-3, 22:4n-6, 22:5n-3, and 22:6n-3) were chosen as variables for PCA. The analysis, depicted in Figure 39 [68], showed that all four families occupied distinct spaces delineated by axes 1 and 2. Additionally, the Poritidae family split into two distinct areas, representing the *Porites* and *Goniopora* genera. Along axis 1, driven primarily by acids 20:1n-9, 18:1n-7, 20:4n-3, and C22 PUFAs, the Faviidae and Goniopora were differentiated from *Porites*, *Acropora*, and *Fungiidae*. The distinction of Porites, Acropora, and Fungiidae along axis 2 mainly arose from variations in 18:1n-9 and 20:1n-9 content, with 20:4n-3, 20:5n-3, and 22:6n-3 also playing roles.

Data from FA composition samples of five symbiotic hard coral families (Acroporidae, Faviidae, Fungiidae, Poritidae, and Pocilloporidae) were compiled to preliminarily evaluate the potential of FAs as chemotaxonomic indicators [68,103]. We integrated and contrasted FA composition data from these five family representatives using PCA. Variables chosen were acids 18:1n-9, 18:1n-7, 20:1n-9, 20:3n-6, 20:4n-6, 20:4n-3, 20:5n-3, 22:4n-6, 22:5n-3, and 22:6n-3. Figure 40 illustrates that representatives from each of the five analyzed families occupied their own distinct domains.

PCA has illuminated the correlation between unsaturated FA composition and the taxonomic classification of hard corals. The PUFA pattern can potentially serve as a chemotaxonomic identifier for specific families and, in certain instances, distinct hard coral genera. This has been validated for five hard coral families (Acroporidae, Faviidae, Fungiidae, Pocilloporidae, and Poritidae) from Vietnam and Seychelles [68,103]. To ascertain FA distribution uniformity in other families and genera, a broader coral sample set is necessary. The shallow waters of Vietnam host a larger number of symbiotic hard corals compared to non-zooxanthellate hard coral species. Notably, the FA compositions of symbiotic and non-symbiotic coral species exhibit significant disparities. As a result, for more accurate outcomes, it is recommended that statistical analysis of FA composition in symbiotic coral species be conducted independently from that in non-symbiotic species. It is hypothesized that the observed differences in PUFA composition across the examined hard coral families arise from unique biosynthesis attributes and varying accumulation levels of specific PUFA [68].

### 4.2. Fatty Acids in Chemotaxonomy of Soft Corals

The initial effort to use total FA for the chemotaxonomy of soft corals was undertaken in 2005 on 32 species from Vietnam [106]. Through PCA, the authors managed to segregate gorgonians, alcyonarians, and antipatharians based on their total FA composition, demonstrating its effectiveness at the subclass level. Regrettably, several errors in FA identification within this study undermine the reliability of its conclusions and complicate subsequent comparisons.

The FA composition of soft corals is species specific but can fluctuate significantly due to environmental factors, food source availability, and symbiont composition [108,145]. If the species-specific fluctuations in individual FA content surpass interspecies variations, then employing these FAs as taxonomic markers becomes challenging. For instance, in a comparative analysis of the tropical alcyonarian *Sarcophyton* sp., the gorgonian *Euplexaura erecta*, and the boreal alcyonarian *Gersemia rubiformis*, distinct differences in individual FA contents among these species, hailing from diverse taxonomic groups and habitats, were observed [204]. Three acid groups—14:0 + 16:0 + 18:3n-6, 16:2 + 20:4n-6 + 24:5n-6, and 18:1n-7 + 20:1n-7 + 20:5n-3 + 24:6n-3—were proposed to typify *Sarcophyton* sp., *E. erecta*, and *G. rubiformis*, respectively. It was noted that no significant disparities existed (*p* > 0.05) among the examined soft coral species in terms of three primary FAs (18:1n-9, 18:2n-6, and 22:6n-3), meaning intraspecies variations in these FA contents outstripped interspecies differences. The relative content of about half of all individual FAs in *Sarcophyton* sp., *E. erecta*, and *G. rubiformis* did not significantly differ pairwise (*p* > 0.05). Due to intraspecies variability, the standard deviation (SD) of individual FA content in soft corals might approach ±30%. This variation should be considered when selecting individual FAs as markers for soft coral chemotaxonomy. Consequently, the authors advised focusing on a subset of the FA composition matrix, prioritizing PUFAs such as 18:3n-6, 20:4n-6, 20:5n-3, 22:5n-6, 22:5n-3, and 24:5n-6, and select monoenoic FAs to more distinctly delineate the systematic relationships of individual soft coral taxa [204].

MSA was employed to compare the total lipid FA composition of two soft coral groups based on the presence or absence of zooxanthellae [92]. The first group comprised 10 alcyonarian species from the genus *Dendronephthya*, which lacked zooxanthellae. The second group included 12 alcyonarian species from the genera *Sarcophyton*, *Lobophytum*, *Cladiella*, *Lytophyton*, *Cespitularia*, and *Clavularia*, all of which contained zooxanthellae. Out of 46 identified FAs, only 12 showed significant differences between the two soft coral groups. Noteworthy FAs, which indicated the presence or absence of zooxanthellae in soft corals, were 7-Me-16:1n-10, 17:0, 18:4n-3, 18:1n-7, 20:4n-6, 22:5n-6, 24:5n-6, and 24:6n-3. Figure 41 presents the PCA results for the content of nine principal FAs across twenty-four soft coral samples. In the two-dimensional space formed by axes 1 and 2, coral groups distinguished by their zooxanthellae content are entirely separated, with axis 2 acting as the discriminant. The primary variables along axis 2 were acids 22:3n-6, 20:4n-6, 24:5n-6, and 18:1n-7.

In conclusion, the total FA profile can serve as a marker at the family level, while the PUFA profile can function as a marker at the genus level in soft coral chemotaxonomy. Importantly, environmental factors seem to neutralize interspecies differences in FA composition [105]. Furthermore, as exemplified by the alcyonarian genus *Dendronephthya*, the absence of zooxanthellae significantly influences the total FA composition of soft corals [92].

A substantial number of alcyonarian and gorgonian species from Vietnam underwent PCA analysis to validate these assertions and determine the role of zooxanthellae in the chemotaxonomy of soft corals based on their FA patterns [68]. The comprehensive matrix of the total FA content, encompassing 45 variables, was utilized for this purpose. As depicted in Figure 19, the analysis distinctly separates non-symbiotic and symbiotic soft coral species. Furthermore, zooxanthellate soft corals are subdivided into two additional groups.

One subgroup of symbiotic corals comprises alcyonarians from the genera *Carijoa*, *Alcyonium*, *Cladiella*, *Lemnalia*, *Nephthea*, gorgonians from *Hicksonella*, and the blue coral *Heliopora coerulea*. This group is characterized by a low concentration of 16:2n-7. Conversely, the second subgroup consists of alcyonarians from the *Lobophytum* and *Sarcophyton* genera, both of which exhibit a high concentration of 16:2n-7. Notably, alcyonarians from the *Sinularia* genus were found dispersed between these two subgroups, contingent on their 16:2n-7 content. A pronounced concentration of 16:2n-7 (reaching up to 16.4% of total FAs) correlated inversely with the level of 18:3n-6, except in the case of the *Nephthea* genus.

The fatty acid (FA) composition data have proven invaluable in the chemotaxonomy of alcyonarians, specifically for the genera *Sinularia* and *Sarcophyton* (Cnidaria: Alcyonaria) [105]. For the analysis, 11 polyunsaturated fatty acids (PUFAs)—18:3n-6, 18:4n-3, 20:3n-6, 20:4n-6, 20:4n-3, 20:5n-3, 22:4n-6, 22:5n-6, 22:5n-3, 24:5n-6, and 24:6n-3—were used as variables. In the visualization illustrated by Figure 42, *Sinularia* (with 18 samples) and *Sarcophyton* (also with 18 samples) distinctly formed two separate clusters on the plane defined by axes 1 and 2. Additionally, three samples from the *Lobophytum* genus were incorporated into the analysis. While the sample size for this genus was limited, the *Lobophytum* cluster was noticeably distinct from those of *Sinularia* and *Sarcophyton*.

In Figure 19, the primary distinction between symbiotic and non-symbiotic species along axis 1 was influenced by 12 of the 45 variables: 16:0, 16:2n-7, 7-Me-16:1n-10, 18:1n-7, 18:3n-6, 18:4n-3, 20:4n-6, 22:4n-6, 22:5n-6, 22:5n-3, 24:5n-6, and 24:6n-3. Further, Figure 43 showcases the bifurcation of symbiotic soft coral species into two subgroups along axis 2. The variables 16:2n-7, 16:1n-9, and 18:2n-7 contributed most positively to this separation, while 18:3n-6, 18:2n-6, 20:5n-3, and 22:6n-3 had the most negative impact. The distinction between symbiotic and non-symbiotic species was characterized by high levels of four FAs: 16:0, 16:2n-7, 18:3n-6, and 18:4n-3. This underscores the potential of the total FA profile as a chemotaxonomic indicator for the presence or absence of zooxanthellae.

Figure 43 demonstrates that prominent alcyonarian genera, including *Lobophytum*, *Sarcophyton*, and *Sinularia*, were not distinctly separated based on their total FA composition. To differentiate these genera, 11 PUFAs, previously proposed for octocoral chemotaxonomy, were employed as variables [105]. Consequently, while *Sinularia* and *Sarcophyton* were clearly distinguished, *Lobophytum* was not differentiated from its close relative, *Sarcophyton*, as visualized in Figure 44.

The primary factors driving this result were the n-3 series PUFAs 18:4n-3, 20:4n-3, and 24:6n-3. This might indicate variances in the conversion processes of these alcyonarian n-3 series PUFAs, specifically in the pathway 18:4n-3 → 20:4n-3 → 20:5n-3 → 22:5n-3 → 24:5n-3 → 24:6n-3. The principal contributors to the n-3 series PUFA are zooxanthellae and dietary microalgae. It is plausible that these closely related alcyonarian species assimilate and utilize these FA sources distinctively.

Consequently, the overall FA profile can serve as a chemotaxonomic marker for soft corals at the subclass level. However, within this subclass, the FA profile of lower-level taxa is impacted by the presence or absence of zooxanthellae. As such, chemotaxonomic analysis based on FA distribution should be separately conducted for zooxanthellate and azooxanthellate soft corals. Both MSA and PCA have yielded promising results. The PUFA pattern acts as a chemotaxonomic indicator for soft corals at the family level and for specific genera. However, distinct lipid markers at the species level have not been identified.

### 4.3. Marker Lipids of Corals, Their Food Sources, Symbionts, and Associated Organisms

The total FA composition in corals stems from a blend of the host organism’s FAs, FAs from intercellular symbiotic microalgae (when present), and associated organisms such as bacteria, filamentous algae, sponges, and marine fungi. This FA composition is shaped by several factors: the specific attributes of the host organism’s FA biosynthesis, the quantity of zooxanthellae, the composition and volume of associated organisms, the FA composition of the food source, environmental influences, and the coral’s developmental stage. Many members of the symbiotic coral association, as well as some coral food sources, possess unique FAs that can serve as markers for these organisms.

Such marker lipids can be instrumental for various purposes in coral research: they can facilitate coral chemotaxonomy, identify the presence and purity of symbionts and associated organisms, shed light on lipid exchanges between symbionts and the host organism, determine coral food sources, and assess the health of coral colonies. Combining FA markers with stable isotope markers has proven effective in studying the trophic ecology of corals [204,205,206,207].

#### 4.3.1. Marker Lipids and FAs of Zooxanthellae and the Host in Corals

The composition of total lipids and FAs has been determined for numerous coral species [68,208,209]. However, there are limited data on the composition of lipids and FAs of zooxanthellae extracted directly from corals [101,112,115]. Only a few studies, such as those by Papina and colleagues [116], Treignier and team [121], and Imbs and associates [134], have concurrently identified the FA composition of both components (algae and animal) from the same hard coral colony. The initial data regarding the FA composition of pure zooxanthellae from soft corals have only recently been gathered [69,134].

Although the FA compositions of pure zooxanthellae vary depending on their source (see Appendix A), a comparative analysis reveals certain FAs typical for all symbiotic dinoflagellates. These can be viewed as markers for zooxanthellae.

A universally recognized marker for zooxanthellae is 18:4n-3 (octadeca-6,9,12,15-tetraenoic acid). Found in significant amounts in zooxanthellae lipids from hydrocorals and hard and soft corals, this PUFA is predominantly present in the glycolipids of zooxanthellae. These glycolipids are essential for the functioning of the thylakoid membranes in the photosynthetic machinery of zooxanthellae [118]. In hard coral zooxanthellae, the content of 18:4n-3 in total lipids and glycolipids can reach up to 15% and 40% of total FAs, respectively. For hydrocorals from the genus *Millepora* and soft corals from the *Sinularia* genus, the percentages are around 17% and 13%, respectively. Among ten strains of zooxanthellae cultured in labs, the 18:4n-3 content varied between 6 and 30%, averaging around 15%. In the case of the hard coral *Stylophora subseriata* and the surrounding zooplankton, 18:4n-3 made up 12.7%, 6.6%, and 1.4% of their total FAs, respectively [38], suggesting that zooxanthellae are the primary source of 18:4n-3 in *S. subseriata*.

Almost all of the 18:4n-3 is localized within the zooxanthellae lipids. The proportion of 18:4n-3 in polyp tissues is significantly reduced, being 15–20 times lower than in zooxanthellae. Hence, by determining the percentage of 18:4n-3 in the lipids of entire coral colonies, we can estimate a crucial aspect: the proportion of zooxanthellae lipids in the total lipids of the colony. This factor represents the extent of the influence of zooxanthellae on the coral’s lipid composition. This is crucial for discerning coral chemotaxonomy via lipid markers and also vital when investigating coral bleaching and nutrition. However, it is worth noting that the 18:4n-3 concentration in the zooxanthellae is variable, making the assessment of the zooxanthellae lipid portion only an approximation.

The 18:4n-3 concentration in whole colonies of both hard corals and the soft corals from the *Sinularia* genus typically ranges between 2 and 4% of total FAs. The mean concentration of 18:4n-3 in the zooxanthellae FAs of these corals stands at around 15%. A rough calculation indicates that the zooxanthellae lipids account for roughly 20% of the coral’s total lipids. Interestingly, the 18:4n-3 content in entire colonies of soft coral species is roughly double that found in whole hard coral colonies. Nevertheless, the absence of data on the 18:4n-3 concentration in pure zooxanthellae from these soft coral species prevents us from precisely determining the zooxanthellae lipid portion.

Azooxanthellate corals’ total FA contains up to 0.5% of 18:4n-3, suggesting that zooxanthellae are not the sole source of 18:4n-3. This fatty acid might also be sourced from free-living dinoflagellates or algae that are associated with the coral colonies.

The acid octadeca-6,9,12-trienoic (18:3n-6) can be recognized as a marker for hard coral zooxanthellae [68]. In pure zooxanthellae from hard corals, its content in total lipids varied between 5 and 23%, averaging about 10% of total FA. For the FA of zooxanthellae glycolipids, it averaged 13%. Cultivated zooxanthellae strains had just 2–4% of 18:3n-6 [118]. The 18:3n-6 content in hard coral polyp tissues was considerably higher (2–4% of total FA) than that of 18:4n-3. This indicates a possible transfer of this PUFA from the zooxanthellae to the host or an uptake of 18:3n-6 by the coral from food, or its own biosynthesis from 18:2n-6. Consequently, accurately determining zooxanthellae lipid shares using 18:3n-6 content, similar to the approach with 18:4n-3, is not feasible.

The fatty acid 18:3n-6 was virtually absent in both entire colonies and pure zooxanthellae fractions and polyp tissues of the *Millepora* genus hydrocorals.

For zooxanthellate hard corals, the combined amounts of the marker PUFAs 18:4n-3 and 18:3n-6 fluctuated based on the coral’s taxonomic classification and environmental conditions. Even partial bleaching (loss of zooxanthellae) led to a notable reduction in the concentration of these markers in the coral’s total FAs. In the hermatypic coral *Pavona frondifera*, a reduction in zooxanthellae led to a drop in the combined content of 18:3n-6 and 18:4n-3 from 12.2% to 1.8% [210]. In comparison, healthy symbiotic hard corals had these acids ranging from 2 to 15% of their total FA [68,100,103]. The levels of 18:3n-6 and 18:4n-3 were minimal in non-symbiotic alcyonarians [79], gorgonians [111], and certain hard corals [68,100].

In summary, if the sum of 18:3n-6 and 18:4n-3 exceeds 2% of a coral colony’s total FAs, it can indicate the presence of zooxanthellae within that coral. The 18:3n-6 content in zooxanthellae extracted from various *Sinularia* species was minimal. Notably, an inverse correlation between 18:3n-6 and 16:2n-7 contents was identified in soft corals, particularly within the *Sinularia* genus [68]. Some *Sinularia* species exhibited a low concentration of 18:3n-6 (0.5%) and a high concentration of 16:2n-7 (8.1%). In contrast, other *Sinularia* species showed a low concentration of 16:2n-7 (0.4%) and a high concentration of 18:3n-6 (6.4%). Those specific *Sinularia* colonies from which pure zooxanthellae were extracted had a very minimal amount of 18:3n-6. Consequently, their zooxanthellae naturally lacked this marker acid. Nonetheless, it is plausible that zooxanthellae extracted from soft corals with a high concentration of 18:3n-6 might also possess a similar level of 18:3n-6 to that found in zooxanthellae from hard corals. Thus, the acid 18:3n-6 can be regarded as a zooxanthellae marker for alcyonarians from the genera *Carijoa*, *Alcyonium*, *Cladiella*, *Lemnalia*, *Nephthea*, specific *Sinularia* species, gorgonians from the *Hicksonella* genus, and the blue coral *Heliopora coerulea*.

Although all species within the *Lobophytum* and *Sarcophyton* genera contain zooxanthellae, the 18:3n-6 content in these soft corals was low. As a result, 18:3n-6 should not be considered a definitive lipid marker for zooxanthellae in soft corals from these two genera. The two C16 PUFAs, 16:3n-4 (hexadeca-6,9,12-trienoic acid) and 16:4n-1 (hexadeca-6,9,12,15-tetraenoic acid), are indicative of soft coral zooxanthellae [69,134]. The biosynthesis of C16 PUFAs (16:2n-7 → 16:3n-4 → 16:4n-1) is restricted in hard coral zooxanthellae and represents a primary biochemical trait of soft coral zooxanthellae. On average, the total content of 16:3n-4 and 16:4n-1 in pure zooxanthellae lipids from *Sinularia* genus soft corals constituted 4.3% and 3.2% of total FA, respectively. The ratio of these markers between zooxanthellae and polyp tissues was between 7 and 10. Unlike plants, animals lack Δ12 and Δ15 desaturases, meaning they cannot produce 16:3n-4 and 16:4n-1 independently. Primary sources of these PUFAs in soft coral polyp tissues, aside from zooxanthellae, are likely plant-based foods such as phytoplankton.

Certain dinoflagellate FAs, such as 20:5n-3 and 22:6n-3, were notably more abundant in pure zooxanthellae lipids compared to the lipids of an entire coral colony. While these PUFAs are characteristic of zooxanthellae, their use as definitive lipid markers presents challenges. For instance, the content of 22:6n-3 in zooxanthellae lipids, whole *Stylophora subseriata* hard coral colonies, and zooplankton collected from the same region showed little variation [38]. In addition to obtaining 20:5n-3 and 22:6n-3 from zooxanthellae, corals also acquire these FAs from phyto- and zooplankton lipids, which form a significant portion of their diet. Currently, methods to differentiate between autotrophic and heterotrophic lipid sources are still in development [211]. Likely, a combined approach using both lipid and isotope marker techniques will be necessary to ascertain the amount of 20:5n-3 and 22:6n-3 introduced into corals via zooxanthellae.

Initially, in total lipids of hard corals of pure zooxanthellae, the 18:5n-3 acid was found in quantities ranging from 2 to 8%, and up to 24% in zooxanthellae glycolipid FA [112]. Subsequent studies confirmed the presence of this acid in zooxanthellae FA composition [115,116,117]. *Millepora* polar lipid FA had an 18:5n-3 content of 8.7% [115]. As 18:5n-3 is distinctively indicative of dinoflagellates and is nearly absent in other sources, it offers potential as a unique zooxanthellae lipid marker. However, many studies on dinoflagellate FA composition used alkaline conditions to extract FA from total lipids, resulting in the degradation of the fragile 18:5n-3 acid and leading to inaccurate measurements of its content [192].

The processes of PUFA biosynthesis in animals and plants differ. Typically, animal desaturases cannot insert a double bond beyond the C-9 atom of the FA molecule’s carbon chain, meaning animals cannot produce 18:2n-6 and 18:3n-3 [210]. This is a primary reason for the distinction between the PUFA compositions of endosymbiotic dinoflagellates (plants) and their hosts (animals). All previously mentioned zooxanthellae FAs as lipid markers are either n-3 series PUFAs or products of specific Δ12 and Δ15 plant desaturases. Animals typically exhibit a dominance of n-6 series PUFAs. Some of these PUFAs, such as 20:4n-6 (eicosa-5,8,11,14-tetraenoic acid) and 22:4n-6 (docosa-7,10,13,16-tetraenoic acid), which are concentrated in polyp tissues, have been suggested as marker indicators for the host organism in hard corals [70]. Many studies have found that the percentages of 20:4n-6 and 22:4n-6 in the hard coral host organism are 2–3 times greater than those in zooxanthellae. In some hard coral species, the 22:4n-6 content constituted over 20% of the total FAs. Hashimoto and co-workers [120] observed that pure zooxanthellae from the soft coral *Clavularia viridis* lacked arachidonic acid (20:4n-6) entirely. Yet, for soft corals of the *Sinularia* genus [69,70], no notable difference was found in 20:4n-6 content between polyps and zooxanthellae, and the 22:4n-6 content was less than 1% of the total FAs.

Very-long-chain fatty acids, specifically 24:5n-6 (tetracosa-6,9,12,15,18-pentaenoic acid) and 24:6n-3 (tetracosa-6,9,12,15,18,21-hexaenoic acid), serve as distinctive markers for the hosts of soft corals. These PUFAs act as chemotaxonomic indicators for octocorals. Tetracosapolyenoic acids (TPAs) have been detected in both zooxanthellate and azooxanthellate soft corals. However, the concentrations of these PUFAs are notably higher in non-symbiotic species compared to their zooxanthellate counterparts. This suggests that the biosynthesis of 24:5n-6 and 24:6n-3 occurs within the polyp tissues, and the zooxanthellae do not contribute to this process [92].

The fatty acid eicosa-4,7,10,13,16-pentaenoic acid (22:5n-6) has been suggested as a host marker for hydrocorals of the genus *Millepora*. This acid is found in minimal amounts in both hard and soft corals. The complexity arises from the similar percentages of 22:4n-6 in both zooxanthellae and hydrocoral polyp tissues. It is theorized that the biosynthesis of n-6 series PUFAs occurs within the host tissues of *Millepora*, following the sequence 18:3n-6 → 20:3n-6 → 20:4n-6 → 22:4n-6 → 22:5n-6. The low concentrations of intermediates (18:3n-6, 20:3n-6, and 20:4n-6) suggest a rapid biosynthesis pathway in *Millepora*. There is a possibility that 22:5n-6 gets transferred from the host to the zooxanthellae. More research is essential to understand the biosynthesis and transfer processes of 22:5n-6 in *Millepora*.

Galactolipids, specifically MGDG and DGDG, which are typical of plant photosynthetic apparatus biomembranes, should be recognized as distinct lipid markers of zooxanthellae in the symbiotic coral community. The PUFAs of these galactolipids primarily mark zooxanthellae [112]. The absence of galactolipids in the polar lipids of polyp tissue fractions from the soft coral *Clavularia viridis* has been cited as proof of this fraction’s high purity [120]. It is likely that Forssman antigen-like glycosphingolipids (gangliosides) of zooxanthellae have a significant role in the symbiosis between corals and Symbiodinium. The lectin SLL-2, extracted from the octocoral *Sinularia lochmodes*, attaches to the glycosphingolipids on zooxanthellae’s surface. Consequently, Symbiodinium cells transition from a flagellated swimming form to a non-flagellated coccoid form, which resembles the symbiotic stage observed in corals [212].

Another unique marker, MADAG, was found in soft coral host lipids. This marker was not present in zooxanthellae lipids but was identified in about 35% of the polyp tissue total lipids [69]. This uncommon lipid class is typical for cnidarians [43]. MADAG content is lower in hard corals compared to soft corals, with zooxanthellate coral species characterized by a higher MADAG content.

#### 4.3.2. Marker FAs of Associated Organisms in Corals

In addition to symbiotic microalgae (zooxanthellae), coral colonies harbor a diverse group of associated organisms, including bacteria, filamentous algae, sea fungi, sponges, and various biofouling entities that often settle on the colony’s exterior. While these associated organisms do not establish true symbiotic relationships with the host coral polyps, their biological activities significantly influence the coral’s nutrition and the exchange of inorganic and organic substances between the corals and their environment. For instance, certain filamentous algae have been identified to form genuine symbiotic bonds with hard corals. The composition, both quantitative and qualitative, of these associated organisms tends to vary and is only somewhat influenced by the coral species. Much of the research on marine associates is qualitative, focusing primarily on the species composition of the identified flora and fauna. Chemical markers, particularly lipid markers, can be employed to determine the presence or absence of specific associates and to provide a rough estimate of their total abundance within coral colonies. Regrettably, research in this direction is in its infancy, and there is a dearth of data on the composition and utilization of lipid markers in coral-associated organisms.

A consistent bacterial community is found in most marine symbiotic animals, and corals are no exception. A comparative study of microbes in healthy and disease-afflicted coral colonies (specifically those affected by white and yellow spot diseases) of 11 coral species from the reefs of Nha Trang Bay, Vietnam, was conducted [213]. This study identified 59 strains of heterotrophic bacteria in the mucus covering the coral colonies. The bacterial composition was not species specific relative to the examined corals and typically included 2–4 primary heterotrophic bacterial species. The fatty acid composition of these bacteria’s lipids was also characterized (refer to Appendix A).

Among the microbial composition of healthy corals, γ–Proteobacteria, notably halomonads, were predominant. Additionally, Gram-negative bacteria such as *Pseudomonas* and *Vibrio* spp., which are part of the Cytophaga–Flavobacterium–Bacteroides (CFB) phylogenetic cluster, were identified [213]. The Gram-positive bacteria detected included species of *Bacillus*, *Staphylococcus*, *Halococcus*, *Micrococcus*, and coryneformic bacteria. The fatty acids (FAs) of these bacteria mainly consisted of saturated and monoenoic FAs, both with unbranched and branched carbon structures, ranging from 10 to 19 carbon atoms (see Appendix A). Hydroxy- and cyclopropane acids were also present. Primary FAs in the polar lipids of *Pseudomonas* and *Halomonas* were 16:0, 16:1n-7, and 18:1n-7. Pseudomonas specifically had cy-17:0. *Vibrio* sp. was characterized by a content of over 40% of 16:1n-7. For the genera *Bacillus*, *Planococcus*, and *Micrococcus*, the predominant FAs were ai-15:0 and ai-17:0.

Numerous studies on coral–microbe interactions have revealed a diverse microbial community residing on coral surfaces and potentially within coral tissues [44,190]. However, it remains uncertain whether these microbes play a specific role in coral biology or if the observed association merely represents an opportunistic collaboration between corals and marine water bacteria. Bacteria stand out from eukaryotes in their ability to synthesize a broad range of branched saturated and unsaturated FAs, as well as acids with an odd number of carbon atoms [183]. They also produce 18:1n-7. Acids with an odd number of carbon atoms, branched acids, and 18:1n-7 are considered bacterial markers. These markers have been utilized to identify bacterial presence [68].

In comparison to symbiotic coral species, the content of “bacterial” FAs was significantly higher (*p* < 0.05) in non-symbiotic species, indicating a more robust bacterial community [68]. A notable increase (*p* < 0.01) in bacterial FA levels within the total FAs of non-symbiotic soft corals from the *Dendronephthya* genus was observed [92]. The bacterial FA content showed a significant difference (*p* < 0.05) between non-symbiotic and symbiotic gorgonian corals (refer to Figure 45). This indicates a more prominent bacterial community in azooxanthellate gorgonians [111].

Another intriguing compound, the 7-methyl-6-hexadecenoic acid (7-Me-16:1n-10), was found in all examined soft corals. The concentration of this acid was significantly higher in azooxanthellate alcyonarians from the *Dendronephthya* genus compared to other zooxanthellate alcyonarians (*p* < 0.01) [92]. In *Dendronephthya*, the 7-Me-16:1n-10 content reached up to 7.3% of the total FAs. In contrast, the average content of 16:1n-10 was only 1.1 ± 0.7% across 13 zooxanthellate alcyonarian species. Moreover, the level of 7-Me-16:1n-10 was significantly higher (*p* < 0.01) in non-symbiotic gorgonians at 2.4 ± 1.2%, compared to their symbiotic counterparts [111]. On average, this FA constituted 2.9% of the polar lipid FAs in *Gorgonia* corals [45]. An increase in the 7-Me-16:1n-10 content was also observed in hard corals lacking zooxanthellae. The concentration of this FA was 2.3% of the total FAs in azooxanthellate species such as *Balanophyllia* sp. and *Tubastrea aurea* [68], while it made up no more than 0.6% of the total FAs in zooxanthellate hard corals.

The fatty acid 7-Me-16:1n-10 has been identified in marine invertebrates such as sponges, actiniae, and gorgonians. Carballeira and colleagues [45] suggested that either bacteria or zooxanthellae might produce this FA. However, a comparison between the fatty acids of zooxanthellate and azooxanthellate soft corals revealed that zooxanthellae likely are not the primary producers of 7-Me-16:1n-10. Notably, marine sponges, which harbor a significant bacterial population and exhibit high levels of 7-Me-16:1n-10 [214,215], do not host zooxanthellae. Consequently, bacteria appear to be the most likely source of 7-Me-16:1n-10 in soft corals.

Significant fluctuations in the levels of 7-Me-16:1n-10, and the presence of 2.7% of this acid in the total fatty acids of the symbiotic gorgonian *R. aggregate* (referenced in Figure 46), challenge the idea of a direct correlation between elevated 7-Me-16:1n-10 levels and the absence of zooxanthellae in corals [111]. It is possible that 7-Me-16:1n-10 marks the presence not of the entire bacterial community, but of specific bacterial groups associated with corals [92].

Zooxanthellae serve as the primary source of organic carbon for their host organisms. The rise in bacterial populations within azooxanthellate corals might represent an adaptive strategy to compensate for the absence of zooxanthellae, enabling the corals to obtain organic carbon from their associated bacteria [92].

A unique group of furan acids (F-acids), which have potential as biomarkers, was identified in gorgonian corals [111]. These acids appear to be widespread across various life forms, having been found in sea sponges, algae, crabs, other invertebrates, and even fish [216]. Both higher plants and algae have been demonstrated to form the carbon skeleton of F-acids from linoleic acid (18:2n-6). Later stages see the elongation of F-acids with acetate residues to achieve their higher homologs. Several studies have reported the presence and biosynthesis of F-acids in phototrophic bacteria [217,218], which are common to many marine creatures. As such, the F-acids discovered in these animals are believed to originate either from marine bacteria or algae [217]. F-acids with chain lengths of 18–22 carbon atoms (43–49) were identified in six azooxanthellate gorgonian coral species, with the exception of *A. isoxia* [111]. Their contribution to the total FA ranged between 0 and 9.7%. The same compounds were previously noted in the aforementioned biological entities [216,217,218].

The source of F-acids in octocorals [111] could be microalgae consumed as food or the microorganisms that live in association with corals, such as bacteria or other microalgae. We are inclined to support the idea that azooxanthellate soft corals form a temporary, non-specific association with phototrophic bacteria or certain microalgae. In this context, the quantity of these associated microorganisms within the coral colony might be estimated based on the levels of their markers, the F-acids.

Furthermore, the presence of C25–28 Δ5,9 unsaturated very-long-chain fatty acids (51–54) or demospongic acids was initially identified in a *Bebrice studeri* sample from near Den Is. (Vietnam) and subsequently in other samples of the same species from Nha Trang Bay (Vietnam) [111]. These acids are characteristic of sponge FAs [219]. Excluding the aforementioned acids, demospongic acids were absent in all other coral species studied. Although the collected *Bebrice* colonies were free from fouling, the existence of an endosymbiotic sponge from this genus had been previously documented [220]. Consequently, the discovery of demospongic acids in the total FAs of *B. studeri* gorgonians attests to the presence of a symbiotic sponge. These acids can serve as specific markers for the presence of symbiotic sponges in other coral species. Fatty acids comprise around 25% and 6% of the dry weight of corals and sponges, respectively. In *B. studeri*, demospongic acids constituted 20.3 ± 4.5% of the total FAs. A rough calculation indicates that the symbiotic sponge could make up as much as 50% of the overall dry weight of the *B. studeri* coral [111,214,215].

#### 4.3.3. Lipid and FA Transfer between Host Organism and Zooxanthellae in Corals

Corals are marine organisms, and many species obtain organic carbon derived from photosynthesis through their symbiotic relationship with intracellular dinoflagellates known as zooxanthellae [221]. For zooxanthellate corals, phototrophic nutrition is a primary carbon source. In some of these coral species, over 90% of photosynthetic production can be relayed from zooxanthellae to the host cells [222,223]. In return, the host provides zooxanthellae with carbon dioxide and water for photosynthetic processes, as well as the necessary substrates for the synthesis of their cellular components, including the photosynthetic apparatus. The zooxanthellae produce low-molecular-weight compounds, such as glycerol, organic acids, glucose, and amino acids, which are then integrated into the coral tissues [224]. A portion of the organic substances transferred from zooxanthellae to the host consists of lipids. It was initially believed that this transfer took place in the form of “lipid droplets”, observable under microscopic examination of zooxanthellate corals. However, later research indicated that these droplets contain minimal lipids [222].

A study on *Pocillopora damicornis* in laboratory tanks with seawater enriched with dissolved inorganic nitrogen (N) evaluated the rate at which zooxanthellae were lost [225]. For the first two weeks, the lipid content in the N-enriched corals was comparable to that of the controls. However, after this period, it decreased to about two-thirds of the control levels, suggesting that N enrichment leads to a decreased rate of photosynthate transfer to the host.

The mechanism of lipid transfer between the members of the symbiotic community in corals remains an area of investigation. Some studies have indicated that carbon synthesized by zooxanthellae may be passed to the host not just as glycerol, but also in the form of acyl lipids. An analysis comparing the metabolism of labeled ^14^C-bicarbonate and 1-^14^C-acetate in two hard coral species proposed that carbon, once photosynthetically fixed by zooxanthellae, is promptly used for lipid acyl residue synthesis before being transferred to the host organism [33].

Short-term incubations of the zooxanthellate hard coral, *Stylophora pistillata*, with ^14^C-bicarbonate were conducted [226]. Zooxanthellae primarily incorporated ^14^C into a chloroform-methanol fraction rather than water-soluble compounds (88–94% vs. 4–7%). In contrast, the host displayed a greater preference for incorporating ^14^C into water-soluble compounds over the chloroform-methanol fraction (44–67% vs. 33–56%). In vivo, the average translocation rate, defined as the amount of ^14^C found in the host fraction as a percentage of the total ^14^C fixed, was 60.4%. This study affirms that the host takes up a portion of the photosynthetic products from its symbionts. However, the specifics of the compounds transferred, as well as the composition of lipids within the mixture of low-polar compounds in the chloroform-methanol fractions, remain elusive.

For the hard coral *Montipora digitata*, there was documentation of lipid biosynthesis from ^14^C-glucose, which the zooxanthellae transferred to the host [83]. Treignier and co-workers [211] hypothesized that certain lipids are directly transferred from zooxanthellae to the host in the scleractinia *Turbinaria reniformis*.

It was noted that the fatty acid (FA) profile of zooxanthellae is distinct from that of coral polyp tissues. When analyzing the overall FA composition of an entire coral colony, several marker FAs are detected in both the zooxanthellae and the host. However, the origins of these acids and the potential for their transfer between the zooxanthellae and the host are ambiguous. Meyers [97,98,99] theorized that corals obtain saturated FAs from zooxanthellae and unsaturated FAs from external sources, such as zooplankton. Yet, the current literature lacks direct evidence showcasing the transfer of individual FAs from zooxanthellae to the host. Some studies [116,121] alluded to experiments that support the transport of both saturated and unsaturated FAs from zooxanthellae to the host in hard corals. However, these references do not offer experimental data directly evidencing the transfer of FAs from symbionts to the host within corals.

The marker FA method was employed to validate the hypothesis of PUFA transfer from zooxanthellae to the host. The aim was to identify the presence of distinct zooxanthellae marker PUFAs within host tissues. Several studies [116,121] have examined the FA composition of isolated zooxanthellae and pure polyp tissue from identical hard coral colonies. Notably, a modest quantity of PUFAs, specific to zooxanthellae, were identified within the FA profile of the host’s total lipids. PUFAs from the n-3 series, such as 18:4n-3, 22:5n-3, and 22:6n-3, constituted a minor fraction of the total FAs in polyp tissues from the hard coral *Montipora digitata* [116]. Of these, 18:4n-3 is especially significant as a primary zooxanthellae marker, given that corals could acquire 22:5n-3 and 22:6n-3 from dietary sources or through intrinsic biosynthesis. The ratio of 18:4n-3 percentages between pure zooxanthellae and *M. digitata* polyp tissues was 3.92 (as per Papina and co-workers [116]). PUFAs from the n-3 series, namely 18:4n-3, 20:5n-3, and 22:6n-3, were also identified in pure polyp tissues of the hard coral *Turbinaria reniformis* [121]. The percentage compositions of 18:4n-3 in the total FAs of pure zooxanthellae and polyp tissues from *T. reniformis* were 7.9% and 1.4%, respectively. Given that these zooxanthellae marker PUFAs were detected in unadulterated polyp tissues, it is inferred that these PUFAs were sourced from symbionts.

A recent study [122] simultaneously determined the FA compositions of total lipids from zooxanthellae and polyp tissues extracted from various hard corals: *Montipora faliosa*, *M. digitata*, *Pocillopora damicornis*, *Acropora intermedia*, *A. muricata*, *Porites cylindrica*, and *Pavona decussata*, as well as the hydrocoral *Millepora platyphylla* (as outlined in Appendix A). There were marked differences in the FA patterns between zooxanthellae and their hosts. Zooxanthellae predominantly contained n-3 series PUFAs, while polyp tissues were enriched in n-6 series PUFAs. Notably, the symbionts from all examined species demonstrated an elevated presence of 18:4n-3 (up to 29.5% in *P. decussata*). Meanwhile, this marker acid was detectable in the polyp tissues of all analyzed cnidarian species, albeit at a considerably reduced proportion (not exceeding 1.3% of total FAs). Conversely, the proportion of 18:3n-6 in polyp tissues was slightly elevated (2–3% of total FAs) compared to the 18:4n-3 levels (as seen in Table 23). Thus, the partition coefficient (the ratio of acid percentages in zooxanthellae to polyp tissues) for 18:4n-3 was substantially higher (ranging between 11.5 and 22.7) than that for 18:3n-6 (between 1.1 and 9.8). The heightened partition coefficient value for 18:4n-3 underscores the high purity of the zooxanthellae fraction.

The modest presence of zooxanthellae marker FAs in polyp tissues supports the hypothesis that these PUFAs are transferred from the symbionts to their hard coral hosts. It is worth highlighting that the concentration of 18:3n-6 in the zooxanthellae of *M. platyphylla* was notably low, and this acid was also absent in the polyp tissues of this hydrocoral.

When examining the distribution of zooxanthellae marker FAs between the symbionts and the host in soft corals from the genus *Sinularia*, it was observed that 18:4n-3 was predominantly found in zooxanthellae, similar to its concentration in hard corals (as shown in Table 24) [69,134]. Only a slight presence of 18:4n-3 (comprising 1.2–1.3% of total FAs) was detected in the soft coral host tissue. The partition coefficient for 18:4n-3 in Sinularia varied between 7.0 and 10.2. Meanwhile, the partition coefficients for two other marker acids, 16:3n-4 and 16:4n-1—both characteristic of soft coral zooxanthellae—were comparatively lower, ranging from 3.8 to 11.0. The 16:2n-7 concentration, which acts as a precursor to C16 PUFA in zooxanthellae, was relatively consistent in both the zooxanthellae and soft coral polyp tissues, with a partition coefficient between 1.2 and 1.5. Interestingly, the concentration of 18:2n-7, which derives from 16:2n-7, was found to be greater in the polyp tissues (as detailed in Table 24).

The minor presence of zooxanthellae marker FAs in polyp tissues aligns with the hypothesis suggesting the transfer of PUFAs from symbionts to the host in soft corals. However, the proportion of these marker PUFAs in the total FAs of the host was relatively low for both soft and hard corals. It is plausible to assume that the presence of zooxanthellae PUFAs in the polyp tissues could be attributed to contamination by zooxanthellae lipids during polyp tissue extraction. Additionally, the host could intake C16 and C18 PUFAs from their food sources.

A detailed study assessed the dynamics of zooxanthellae and their marker PUFAs in whole colonies of both hard (specifically *Montipora* sp. and *Acropora* sp.) and soft (*Sinularia* sp.) corals during coral bleaching [134]. In this study, the marker PUFAs’ content was gauged in unfractionated, intact coral colonies to prevent potential contamination by zooxanthellae lipids. For the hard corals studied, 18:3n-6 and 18:4n-3 were identified as the zooxanthellae marker PUFAs, while for the soft corals, C16 PUFAs and 18:4n-3 were the markers. A 50% reduction in zooxanthellae led to only a 1.1-fold decrease in their marker PUFA levels in *Montipora* sp. (as shown in Figure 46A) and a 1.3-fold decrease in *Sinularia* sp. (Figure 46C). Surprisingly, even with a substantial 65% reduction in initial zooxanthellae levels in *Acropora* sp., there was not a significant dip in marker PUFA levels (Figure 46B). By the experiment’s conclusion, when coral bleaching was nearly complete (with a 90–95% loss of zooxanthellae), the levels of 18:3n-6 were 28% and 49% of their original values in *Montipora* sp. and *Acropora* sp., respectively (as shown in Figure 46A,B). As for the 18:4n-3 levels, they were 46%, 38%, and 36% of their starting levels in *Montipora* sp., *Acropora* sp., and *Sinularia* sp., respectively, by the study’s end. Furthermore, the C16 PUFA level was at 45% of its initial value in *Sinularia* sp. (illustrated in Figure 46C).

The comparative analysis of zooxanthellae dynamics and their marker PUFAs revealed that a high concentration of zooxanthellae marker PUFAs remained in corals, even when approximately 95% of zooxanthellae had been lost from these colonies. This indicates that the marker PUFAs were retained within the host, rather than being lost alongside the symbionts. Logically, the concentration of symbiont marker PUFAs in the host would decrease when their primary source, the zooxanthellae, is diminished. This decline, however, seems to be a gradual process, likely spanning several days [134]. For instance, when half of the zooxanthellae were depleted, the concentration of symbiont marker PUFAs in the overall FAs of the intact coral colony experienced only a minor reduction. As a specific example, experimental temperature stress led to a 1.5-fold decrease in the zooxanthellae density within hard coral *Turbinaria reniformis* colonies, yet the levels of zooxanthellae marker FAs, such as 18:3n-6 and 18:4n-3, remained relatively stable [227]. The stable carbon isotopic values (δ13C) of 18:3n-6, 20:4n-6, and 22:4n-6 from *T. reniformis* colonies did not show significant variance when external food sources were excluded. This observation aligns with the notion that 18:3n-6 may be transferred from zooxanthellae to the host, serving as a precursor for the synthesis of 20:4n-6 and 22:4n-6 in polyp tissues.

Coral bleaching experiments have indirectly shown that certain symbiotic marker PUFAs are transferred to the host in both hard and soft corals [134,227]. The exploration of the transfer of C20-22 PUFAs, such as 20:5n-3, 22:5n-3, and 22:6n-3, from symbionts to the host is challenging using the marker acid method. This is because these PUFAs, while not unique to zooxanthellae, are scarcely found in the coral’s diet and are predominantly synthesized in the coral tissues themselves.

Beyond the marker FA method, the stable carbon isotope (δ^13^C) composition of individual FAs, determined using gas chromatography-isotope ratio mass spectrometry (GC-IRMS), has proven effective in examining coral FA acquisition through translocation from zooxanthellae or via heterotrophic plankton consumption. The δ^13^C values of FAs in corals would align with those of the same compound in its dietary source, as the isotope composition reflects the FA’s synthetic pathway and source. The δ^13^C values for FAs produced by zooxanthellae and zooplankton differ significantly, by over 5‰ [211].

Comparing the δ^13^C values of individual FAs in zooxanthellae and coral hosts from two hard coral species, *Montastraea faveolata* and *Porites astreoides*, revealed that most energy-rich saturated FAs in coral colonies come from their zooxanthellae [228]. The study also suggested that individual coral colonies source their essential PUFAs from both autotrophic and heterotrophic inputs, with the feeding mechanism specific to each colony. For instance, in *P. astreoides*, the δ^13^C values of 18:4n-3 in the host ranged between −12.2‰ and −15.8‰, which closely resembled those in zooxanthellae (from −11.5‰ to −13.8‰). However, these values differed significantly from those in zooplankton (from −22.0‰ to −25.0‰), indicating that zooxanthellae are the most likely source of 18:4n-3 in the host [228].

In another study using ^13^C and ^15^N isotope tracers, Tanaka and co-workers [229] delved into the carbon (C) and nitrogen (N) metabolism within the coral–zooxanthellae symbiotic system of *Acropora pulchra*. They posited that the symbiotic system selectively consumed organic compounds with high C:N ratios, such as lipids and carbohydrates, faster than those with low C:N ratios such as proteins and nucleic acids.

The study explored the use of carbon in the coral tissue and zooxanthellae of *Stylophora pistillata* and *Favia favus* from the Red Sea, extending down a depth gradient of up to 60 m. Additionally, the isotopic carbon composition of the lipid fraction, taken from both the coral tissue and algal symbionts, was measured [230]. The findings revealed that for both species, the δ^13^C values in algal lipids were higher than those in tissue lipids. As depth increased, δ^13^C values diminished by 7–8‰ in lipid fractions. Moreover, algal lipids had δ^13^C values that were, on average, ~2‰ lower than those of the entire zooxanthellae at all depths. This suggests a highly efficient carbon recycling process between the coral host and its algal partners.

In another study, the stable isotope (^13^C) uptake from dissolved inorganic carbon (NaH^13^CO_3_) in the coral *Acropora millepora* and a clade B dinoflagellate culture was documented [231]. Using the symbiotic anemone *Aiptasia pulchella* (Anthozoa, Cnidaria) as a primary model system, the research delved into free FA (FFA) synthesis in both the dinoflagellate symbionts and cnidarian host and the lipid translocation from zooxanthellae to the host. Notably, no discernible evidence was found of the symbiont-derived enriched isotope FA or the direct utilization of catabolized ^13^C derivatives in host long-chain FA lipogenesis. Such results do not concur with the widely accepted translocation model concerning the employment of translocated symbiont photo-assimilates in host long-chain FA lipogenesis. It is crucial to understand that the findings of Dunn and colleagues [231] were exclusively based on free FA, which only constituted a small percentage of the total lipids. Such lipids can stem from both non-specifically hydrolyzed lipids and unidentified associated microorganisms. Therefore, the ^13^C values recorded for FFA might not accurately portray the inorganic carbon integration in the total lipids and their FA. They also do not shed light on the lipid translocation process from zooxanthellae to the host in *A. pulchella*.

In summary, a potential mechanism for the PUFA transfer from symbionts to the host in corals can be postulated. Acids such as 18:4n-3, 16:3n-4, and 16:4n-1 are produced in zooxanthellae and are either transferred in limited quantities to the host or are quickly metabolized by the host. Meanwhile, 18:3n-6 is transferred in larger volumes, especially in hard and certain soft coral species, and may serve as a precursor for n-6 series PUFAs in animal tissues. An interchange of 16:2n-7 likely occurs between the zooxanthellae and the host. The elongation of 16:2n-7 to 18:2n-7 predominantly happens within animal tissues. The methods and chemical forms of PUFA transfer from symbionts to hosts remain subjects of ongoing research and discussion.

A review of studies on zooxanthellae lipids reveals that zooxanthellae derived from different cnidarian species exhibit diverse FA compositions. For instance, tetracosapolyenoic FA is found in zooxanthellae lipids from soft corals, while 22:5n-6 is identified in hydrocoral lipids. Many researchers believe that corals are colonized by distinct zooxanthellae species, each with its unique, species-specific FA composition. However, there has not been direct taxonomic identification of these zooxanthellae species in certain studies [112,115]. FAs that are not typical of microalgae are suspected to have originated from contamination by animal lipids (such as polyp tissues) during the extraction of symbiont fractions [69].

The taxonomy of coral zooxanthellae remains underdeveloped. All potential coral zooxanthellae species are generally categorized under the Simbiodinium group [232]. The prevalent classification of coral zooxanthellae relies on genetic analysis of 16S ribosomal DNA. Currently, eight distinct zooxanthellae clades have been identified (A–H) [233], and of these, five (A–D, F) are symbiotic with corals. The lipid composition of individual coral zooxanthellae clades has not been extensively studied. Yet, it is puzzling that if all coral zooxanthellae fall under the same genus, their FA compositions would vary so significantly. This variation in zooxanthellae FA composition can be more comprehensibly explained if one assumes that lipid transfer is bidirectional: not only from zooxanthellae to polyps, but also in the reverse, from the host organism back to the symbionts in corals.

A principal component analysis (PCA) of the PUFA content in whole colony lipids, zooxanthellae, and polyp tissues from 10 cnidarian species, which included both soft and hard corals, as well as *Millepora platyphylla* (referenced in Table 21, Figure 47 and Figure 48), was conducted to validate the aforementioned hypothesis [122]. Eight PUFAs (namely 16:3n-4, 16:4n-1, 18:3n-6, 18:4n-3, 20:4n-6, 20:5n-3, 22:5n-6, and 22:6n-3) were selected as variables for the PCA.

The intact colonies of hard corals, soft corals, and hydrocorals, along with their polyp tissues, were distinctly segregated into three groups within a two-dimensional space defined by axes 1 and 2, as illustrated in Figure 47. Likewise, zooxanthellae were categorized into three groups that aligned with the taxonomic classifications of their hosts. Notably, zooxanthellae from hard corals were distinctly separated from those of soft corals and hydrocorals (as seen in Figure 47). Clearly, if all zooxanthellae shared identical FA compositions, they would not cluster into these distinct, tight groupings.

Interestingly, no correlation was observed between the FA composition and the phylogenetic classification of the investigated zooxanthellae (as referenced in Table 21 and Appendix A). This suggests that the variations in FA composition among zooxanthellae from different cnidarian groups are not influenced by the phylogenetic classifications of these zooxanthellae [122].

Zooxanthellae derived from soft corals (Octocorallia) were distinctly set apart from those of hard corals and hydrocorals (Hexacorallia) based on their PUFA composition, as shown in Figure 47. To clarify the differences in the FA composition of zooxanthellae across different coral types, the distribution of polyp marker PUFAs between the symbionts and their hosts was analyzed (refer to Appendix A and Figure 6). The host’s signature markers are n-6 series PUFAs, such as 20:4n-6 and 22:4n-6 [69,121]. Tetracosapolyenoic FAs, specifically 24:5n-6 and 24:6n-3, are indicative of octocoral polyp tissues, whereas 22:5n-6 is characteristic of Milleporidae polyp tissues [68].

A considerable quantity of both 20:4n-6 and 22:4n-6 was present in all zooxanthellae (referenced in Appendix A and Figure 38). Additionally, the concentration of 20:4n-6 in both polyps and their respective zooxanthellae was notably similar in certain coral species, particularly in *P. damicornis* and *P. cylindrica*. No significant disparities were found in the levels of 22:4n-6 and 22:5n-6 between the zooxanthellae and polyps of *M. platyphylla*. FAs 24:5n-6 and 24:6n-3 were identified in the zooxanthellae of soft corals; these fatty acids serve as markers for the host organism (as illustrated in Appendix A and Figure 38). The detection of distinct animal-specific PUFA markers in zooxanthellae may be attributable to the transfer of these PUFAs from the host to their symbiotic partners. Such a transfer could alter the inherent FA pattern of zooxanthellae and might account for the observed variations in the FA compositions of zooxanthellae across different cnidarian classifications.

In the 10 cnidarian species studied, the zooxanthellae of hard and soft corals predominantly contained 18:3n-6 and C16 PUFAs, respectively. Neither 18:3n-6 nor C16 PUFAs were present in hydrocorals. Clearly, PUFA biosynthesis in these zooxanthellae groups follows distinct pathways [70]. Crucially, both 18:3n-6 and 16:2n-7 are synthesized from 18:2n-6 and 16:1n-7, respectively, employing the same enzyme, Δ6 desaturase. It is plausible that the host organism exerts some level of regulation or modulation over PUFA biosynthesis within zooxanthellae, possibly by supplying the Δ6 desaturase with varied substrates (either 18:2n-6 or 16:1n-7). This regulatory influence exerted by the host on PUFA synthesis within the zooxanthellae might also explain the dependency of zooxanthellae FA composition on the host’s taxonomic classification.

Consequently, the impact of the host’s nature on the FA composition of their symbionts has been demonstrated. The transfer of host PUFAs to zooxanthellae and/or the modulation of PUFA biosynthesis in zooxanthellae by the host might be contributing factors to this effect. In their interaction, zooxanthellae occupy almost all the internal space of the host cell, reducing its cytoplasm and cell membrane to a thin external layer. Such spatial organization might facilitate the diffusion and transport of lipids and FAs between the host and the zooxanthellae. However, the exact mechanism of this transfer requires further study.

## 5. Conclusions

Reef-building corals, pivotal to marine ecosystems, boast an intricate and varied assortment of lipids with substantial ecological and potential medicinal value. This wide array of lipids, spanning from prevalent to unusual types, offers avenues for diverse scientific breakthroughs. Beyond underlining the biochemical diversity of these lipids, the complex symbiosis between corals and their resident dinoflagellates underscores the intricate nature of substances derived from corals. As such, the potential for delving deeper into the study and application of these compounds in both research and therapeutic arenas is vast.

## Figures and Tables

**Figure 1 marinedrugs-21-00539-f001:**
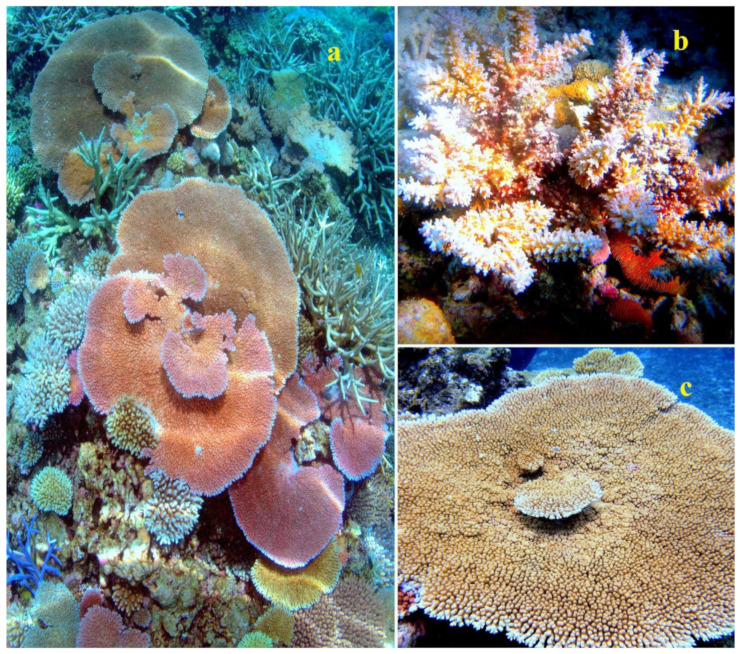
Corals belonging to the family Acroporidae from the phylum Cnidaria: (**a**) *Acropora hyacinthus*, (**b**) *Acropora echinata*, and (**c**) *Acropora hyacinthus*. Acropora species belong to small polypedic stony corals, a family of cnidarians in the order Acroporidae. The name of the family comes from the Greek “akron”, meaning “top”, and refers to the presence of corallite at the tip of each coral branch. This group is known as staghorn corals which form the backbone of the reef. The color of the coral is entirely due to symbionts—zooxanthellae—and they are usually brown colonies, dirty greenish brown, and less often crimson, and the ends of the branches are often blue and sometimes can be yellow. Acropora is cosmopolitan and tends to dominate the Indo-Pacific reefs. Representatives of this family are producers of unusual fatty acids and lipids. See text for details. The pictures are taken from websites that give credit for non-commercial use.

**Figure 2 marinedrugs-21-00539-f002:**
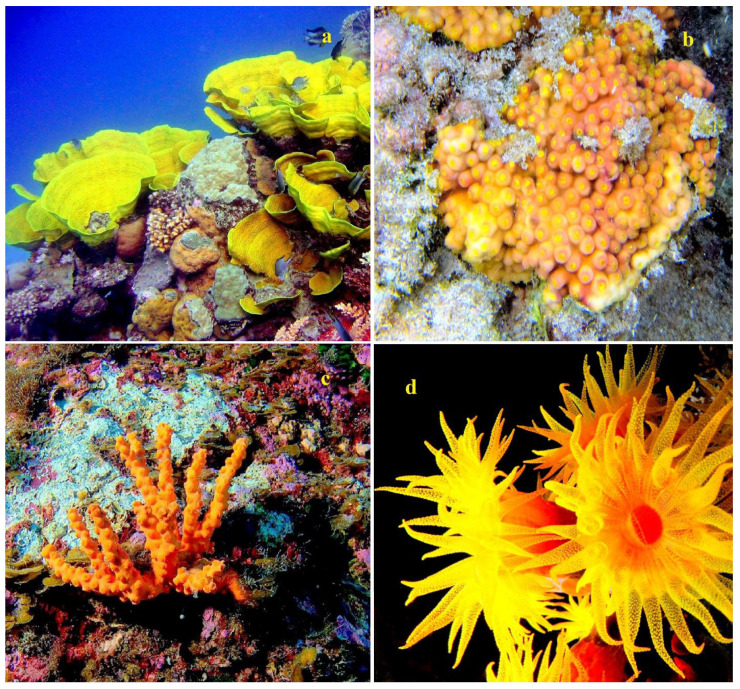
The family of stony corals Dendrophyllidae includes (**a**), *Turbinaria mesenterina*, (**b**), *Turbinaria* cf. *frondens*, (**c**), *Dendrophyllia* cf. *micranthus*, and (**d**), *Tubastraea coccinea*. Most species in this family are azooxanthellates and must capture food with their tentacles rather than relying on photosynthesis to produce their food. Members of this family are producers of interesting lipids and fatty acids and other metabolites.

**Figure 3 marinedrugs-21-00539-f003:**
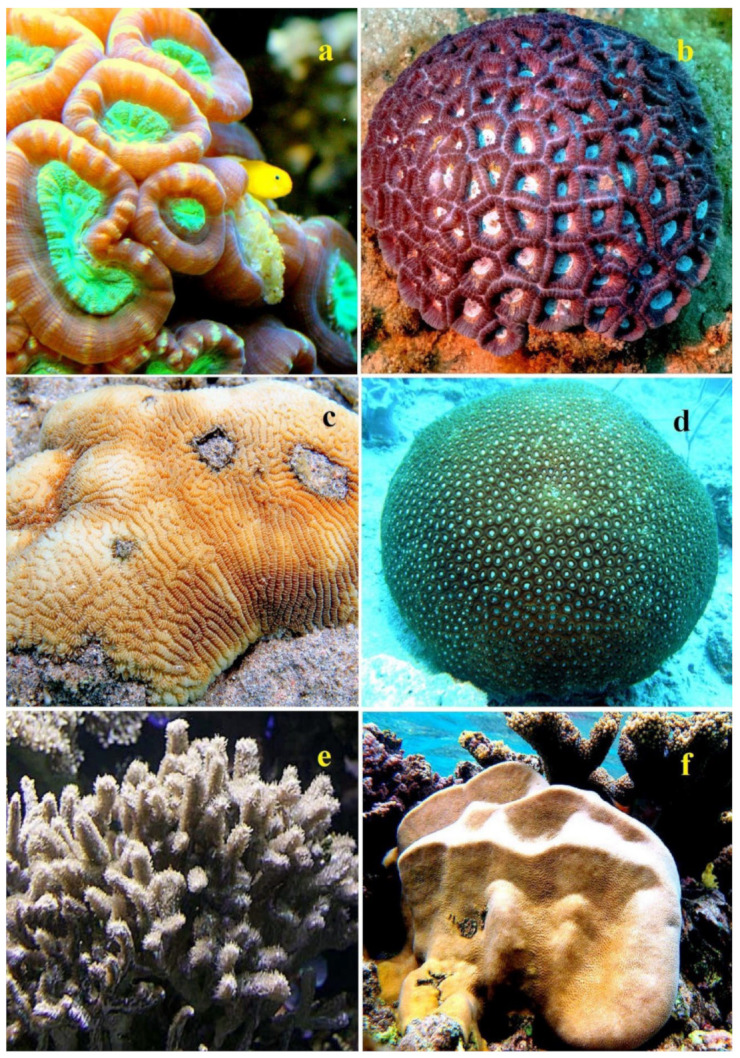
Stony corals (Faviidae family) are in the order Scleractinia and the subclass Hexacorallia (or also known as Zoantharia): (**a**), *Caulastrea furcata*, (**b**), *Favia favus*, (**c**), *Leptoria phyrgia*, (**d**), *Favia* sp., (**e**), *Hydnophora rigida*, (**f**), *Goniastrea stelligera* (Merulinidae family) are widely distributed in the South Pacific.

**Figure 4 marinedrugs-21-00539-f004:**
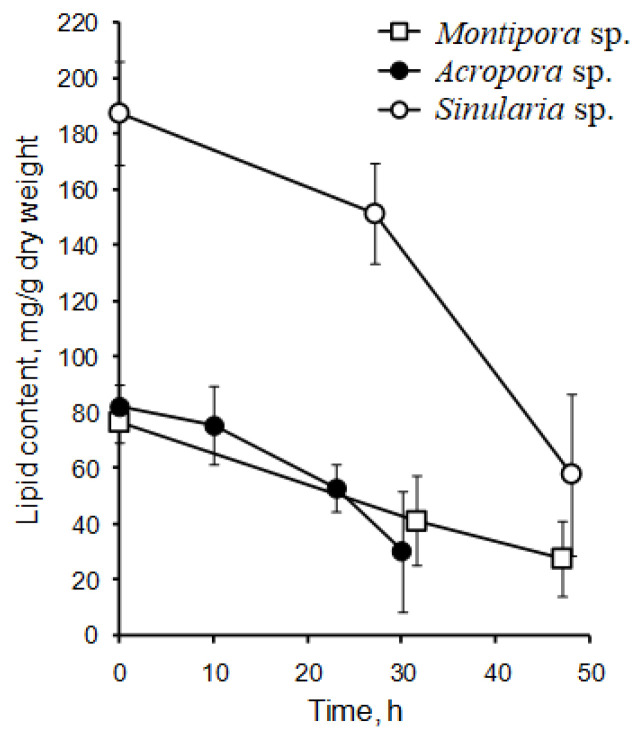
The lipid content change (mg/g DW an average ± SD, *n* = 3) during the bleaching (95% loss of the initial zooxanthellae content) of the soft coral *Sinularia capitalis* and hard coral *Montipora digitata* and *Acropora intermedia* colonies under artificial heat stress (33 °C).

**Figure 5 marinedrugs-21-00539-f005:**
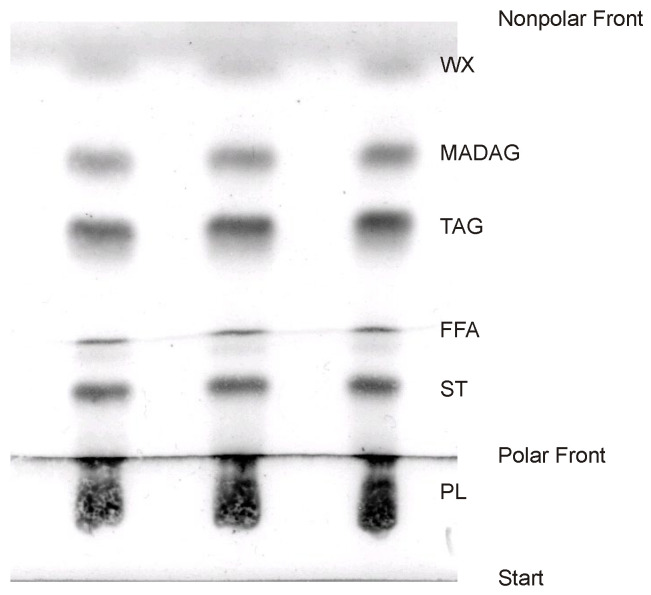
Thin-layer chromatography is often used to digest neutral lipids isolated from corals. WE, wax esters; SE, sterol esters; MADAG, monoalkyldiacylglycerols; TG, triacylglycerols; FFA, free fatty acids; ST, sterols; PL, polar lipids.

**Figure 6 marinedrugs-21-00539-f006:**
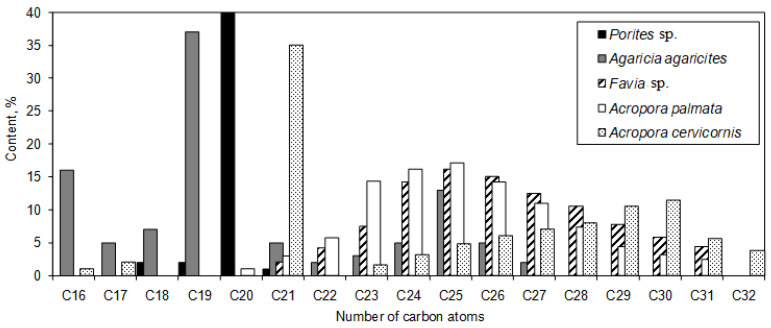
Unbranched saturated hydrocarbons composition (% of the sum) in five species of the hard corals [43].

**Figure 7 marinedrugs-21-00539-f007:**
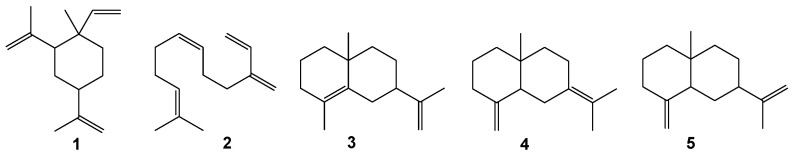
Volatile hydrocarbons derived from extracts of the soft coral *Sinularia* sp.

**Figure 8 marinedrugs-21-00539-f008:**
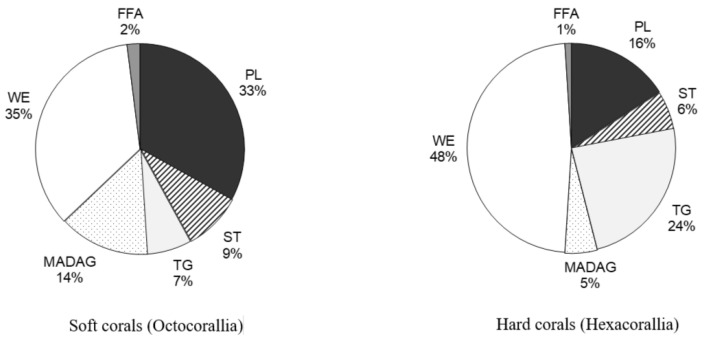
Comparative lipid composition of hard and soft corals derived from South China Sea [68].

**Figure 9 marinedrugs-21-00539-f009:**
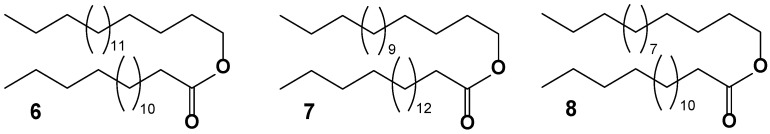
Major components detected in non-polar lipids of the hydrocoral *Millepora* sp.

**Figure 10 marinedrugs-21-00539-f010:**
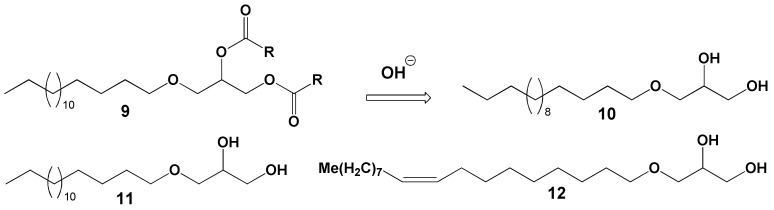
Neutral lipids derived from gorgonian *Pterogorgia anceps*, and the soft coral *Nephthea* sp.

**Figure 11 marinedrugs-21-00539-f011:**
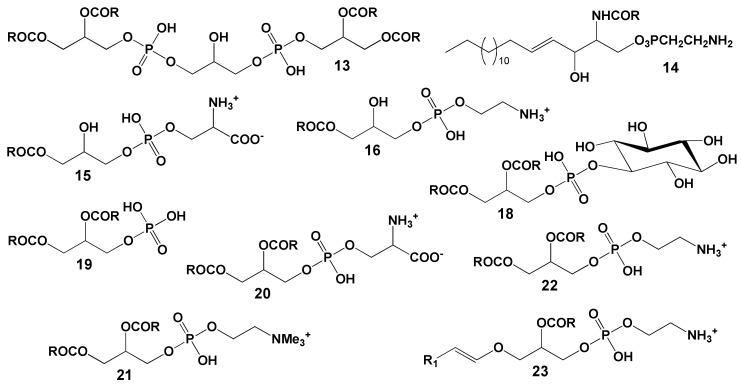
Phospholipids derived from the soft corals.

**Figure 12 marinedrugs-21-00539-f012:**
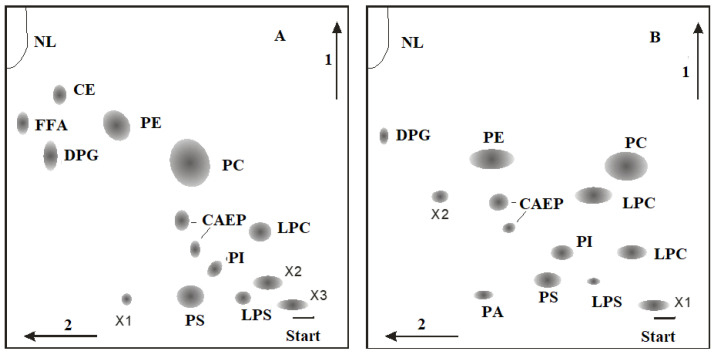
(**A**) Two-dimensional thin-layer chromatogram of the polar lipids of gorgonian *Mopsella aurantia* [87], and (**B**) TLC of the phospholipids of alcyonaria *Sarcophyton digitatum* [88]. Solvent systems: 1, direction chloroform-methanol-25% ammonia, 65:35:5. 2, direction (**A**) chloroform-acetone-methanol-acetic acid-water, 50:20:10:10:5, (**B**) chloroform-methanol- benzyl-acetic acid-acetone-water, 70:30:10:5:4:1. Polar lipid classes: DPG, CAEP, LPS, LPC, LPE, PI, PA, PS, PC, PE, CE, cerebroside; X1, X2, X3, unknown phospholipids.

**Figure 13 marinedrugs-21-00539-f013:**
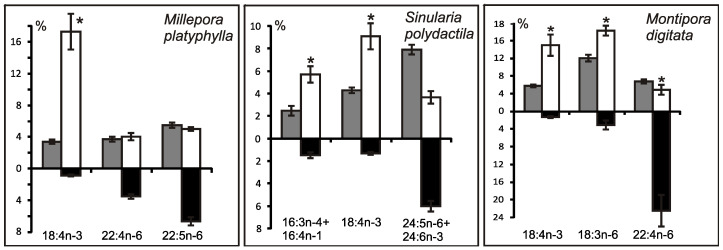
The distribution of marker fatty acids (% of total FAs, mean ± SD, *n* = 3) between symbiotic fractions (white bars) and the host fractions (black bars) isolated from hydrocoral *Millepora platyphylla*, soft coral *Sinularia polydactila*, and hard coral *Montipora digitata* with the comparison to their holobionts (gray bars). *—significant difference (*p* < 0.01) between SF and HF [122]. Statistical analyses were performed using STATISTICA 5.1 (StatSoft, Inc., Tulsa, OK, USA).

**Figure 14 marinedrugs-21-00539-f014:**
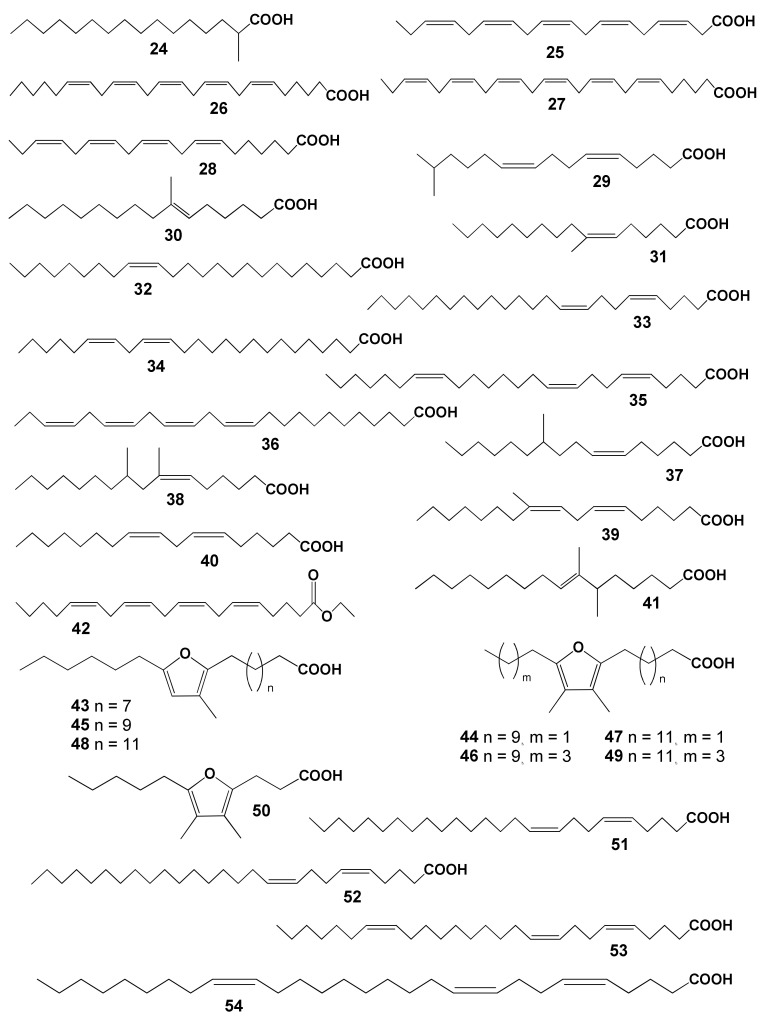
Rare and uncommon fatty acids derived from coral tissue.

**Figure 15 marinedrugs-21-00539-f015:**
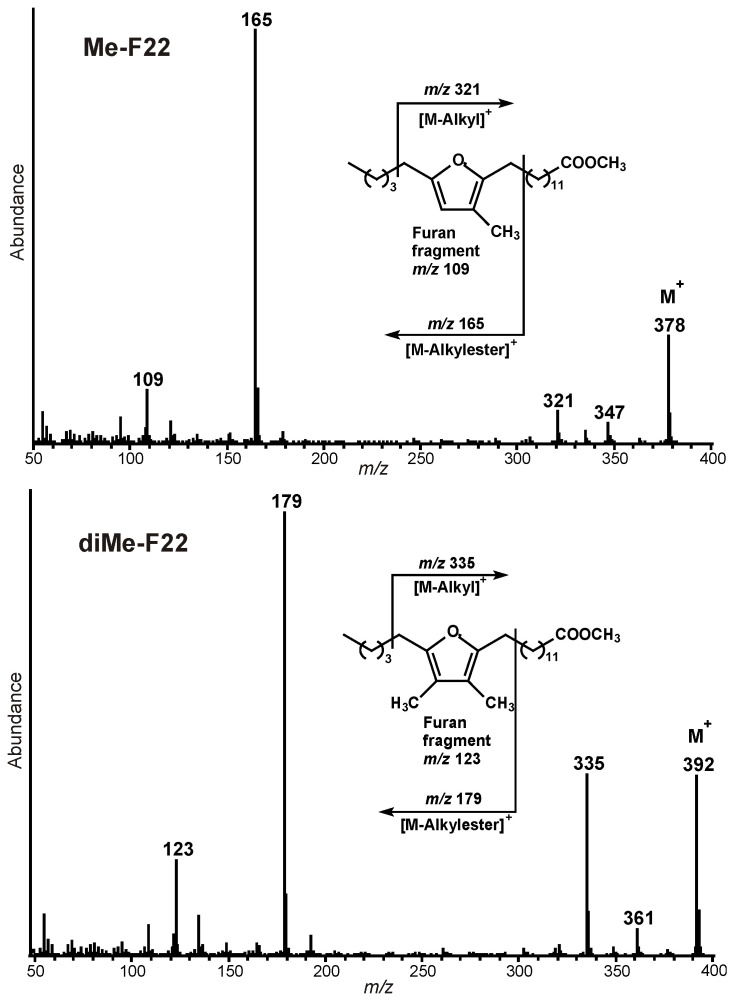
Mass-spectra (electron impact ionization) of methyl esters of 14,17-epoxy- 15-methyldocosa-14,16- dienoic acid (**48**) and 14,17-epoxy-15,16-dimethyldocosa-14,16-dienoic acid (**49**).

**Figure 16 marinedrugs-21-00539-f016:**
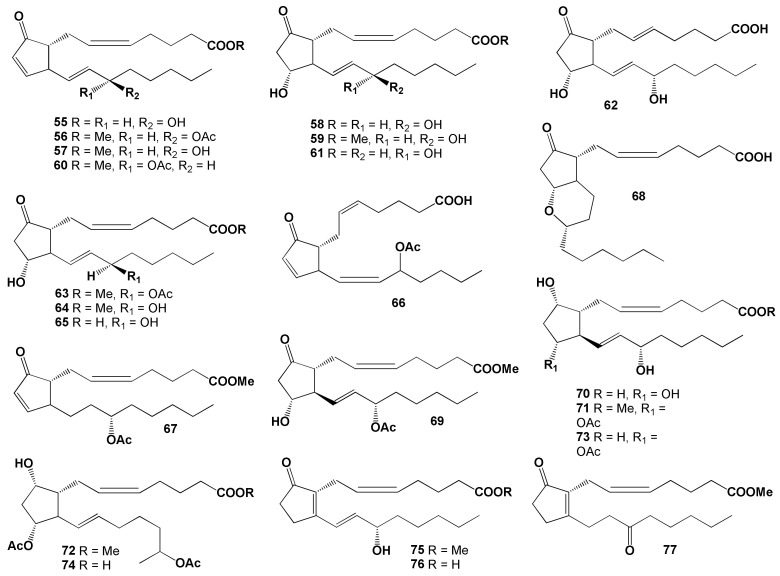
Several prostaglandins derived from lipid extracts of Caribbean gorgonian coral *Plexaura homomalla*.

**Figure 17 marinedrugs-21-00539-f017:**
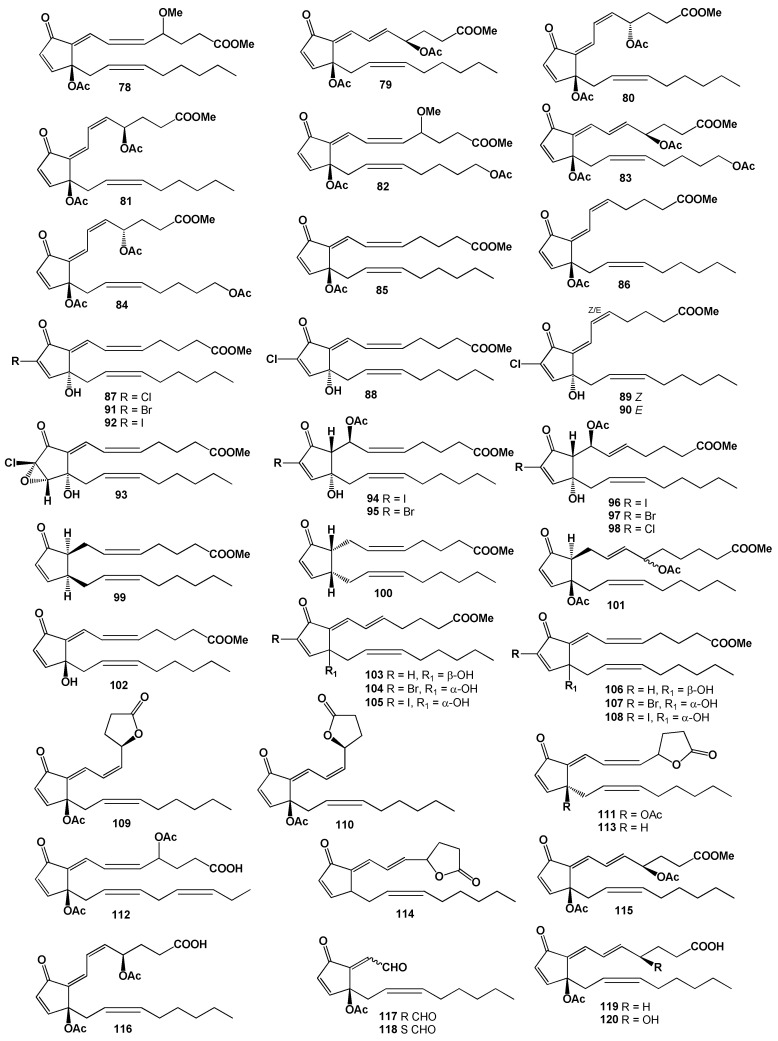
Cyclic oxylipins derived from coral species.

**Figure 18 marinedrugs-21-00539-f018:**
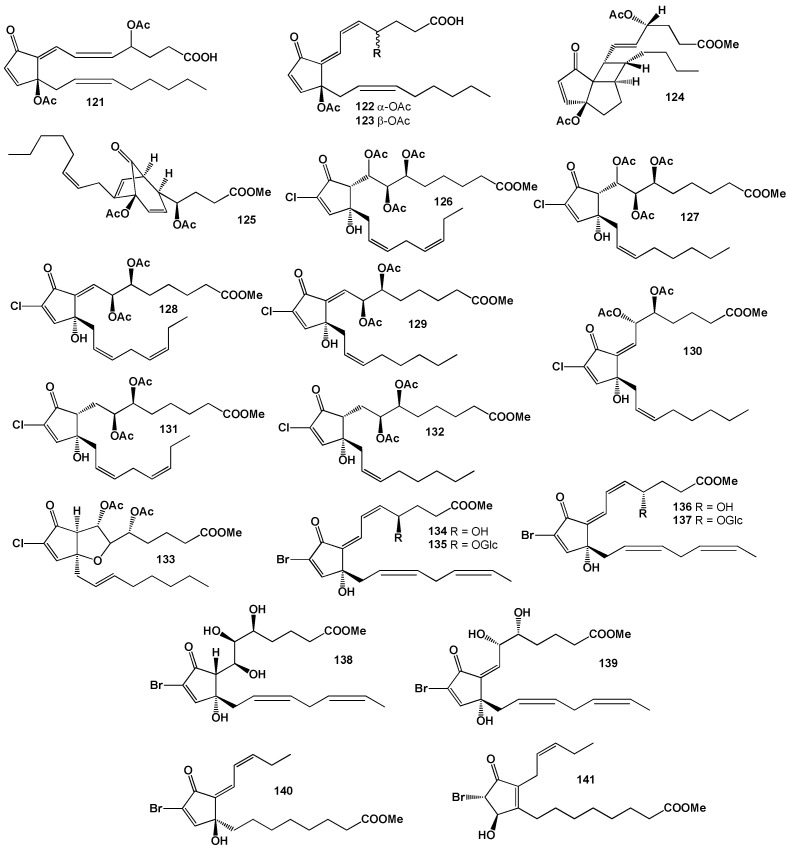
Cyclic and halogenated oxylipins derived from coral species.

**Figure 19 marinedrugs-21-00539-f019:**
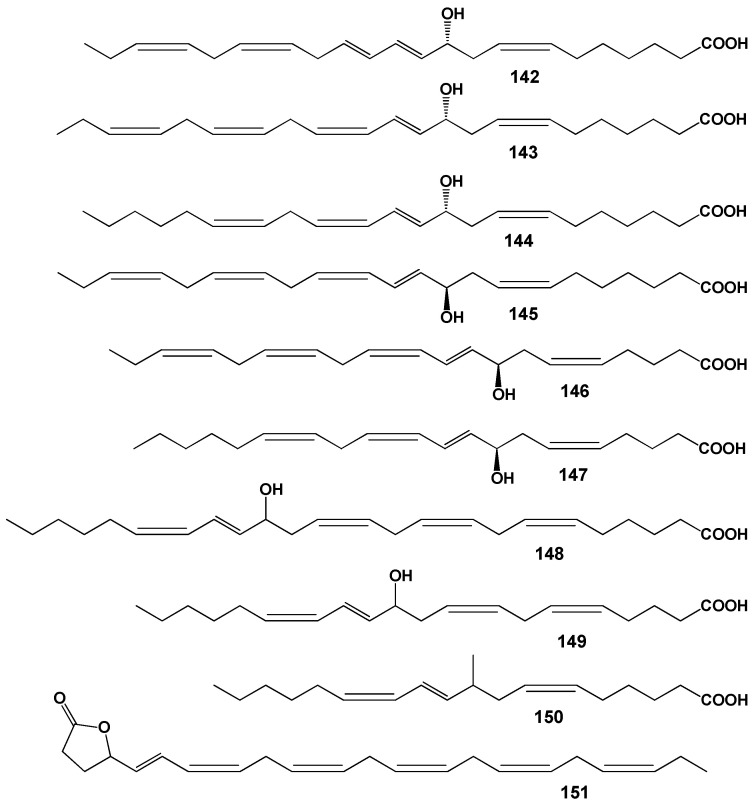
Acyclic oxilipins produced by sort corals.

**Figure 20 marinedrugs-21-00539-f020:**
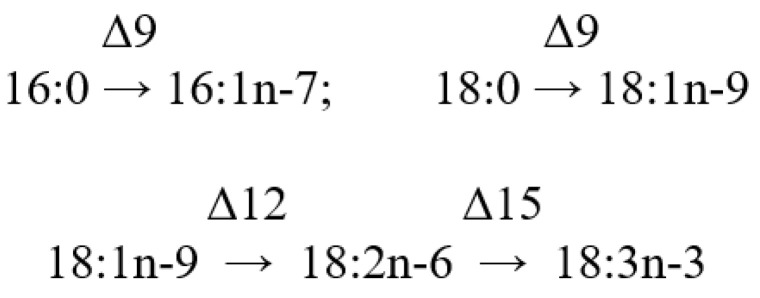
Biosynthesis of unsaturated FA using Δ9-, Δ12- and Δ15-desaturases.

**Figure 21 marinedrugs-21-00539-f021:**
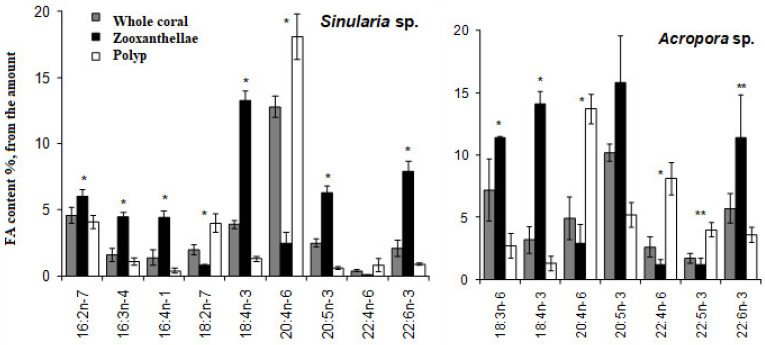
Content of principal PUFAs in zooxanthellae, polyp tissues, and the intact colony of the soft coral *Sinularia* sp. and the hard coral *Acropora* sp. (the South China Sea, Vietnam). Statistically significant differences in PUFA content (* *p* < 0.01; ** *p* < 0.05) between zooxanthellae and polyps are presented [70]. Statistical analyses were performed using STATISTICA 5.1 (StatSoft, Inc., Tulsa, OK, USA).

**Figure 22 marinedrugs-21-00539-f022:**
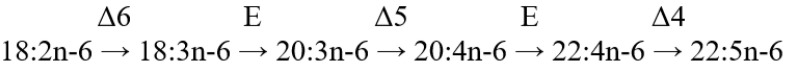
Biosynthesis of C18-22 PUFAs from the n-6 series.

**Figure 23 marinedrugs-21-00539-f023:**
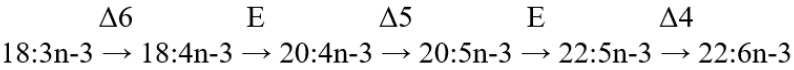
Biosynthesis of C18-22 PUFAs from the n-3 series.

**Figure 24 marinedrugs-21-00539-f024:**
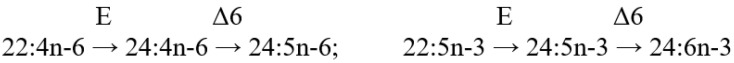
Soft corals synthesize FA from the n-6 series and the n-3 series, in particular 24:5n-6 and 24:6n-3.

**Figure 25 marinedrugs-21-00539-f025:**
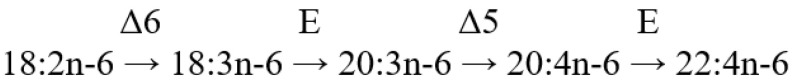
Pathway for n-6 series PUFA synthesis in hard corals.

**Figure 26 marinedrugs-21-00539-f026:**
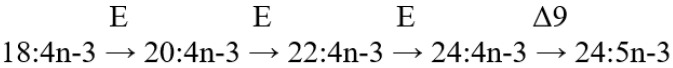
Biosynthesis of 24:4n-3 and 24:5n-3 fatty acids in *Clavularia* sp.

**Figure 27 marinedrugs-21-00539-f027:**
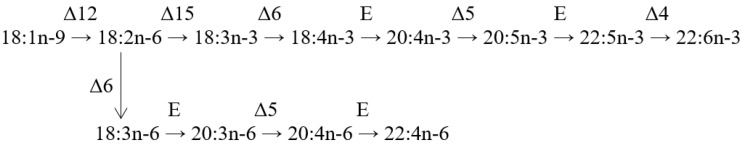
Pathway for n-3 series PUFA synthesis in the zooxanthellae of hard corals,.involving the Δ4, Δ5, Δ6, Δ12, and Δ15 desaturases.

**Figure 28 marinedrugs-21-00539-f028:**
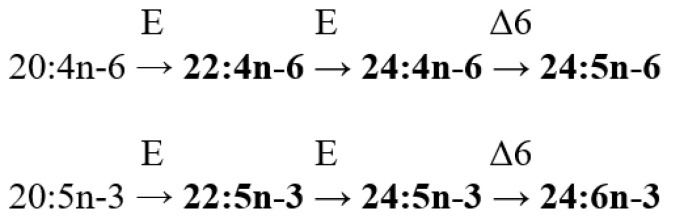
The acids 24:5n-6 and 24:6n-3 are derived from 22:4n-6 and 22:5n-3, respectively, through the sequential action of elongases and Δ6 desaturase.

**Figure 29 marinedrugs-21-00539-f029:**
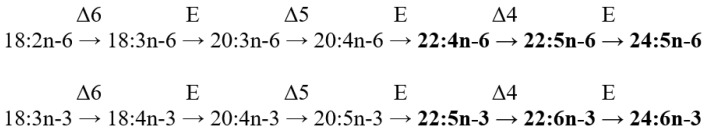
Biosynthesis of 24:4n-6 and 24:5n-3 from 22:5n-6 and 22:5n-3 in soft corals by the action of Δ4-desaturase and elongase in the last two stages of biosynthesis.

**Figure 30 marinedrugs-21-00539-f030:**
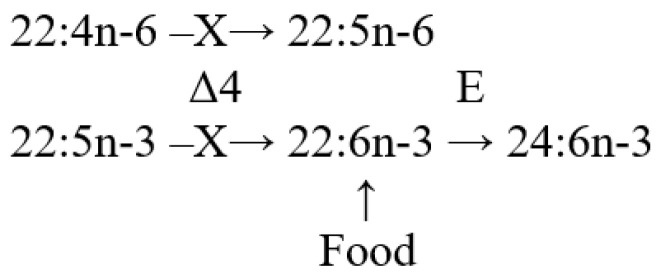
Corals obtain 22:6n-3 from dietary sources, and this acid serves as a substrate for elongase, providing a constant level of 24:6n-3. Simultaneously, inhibition of Δ4-desaturase leads to the accumulation of 22:4n-6, which is not subsequently. converted to 24:5n-6.

**Figure 31 marinedrugs-21-00539-f031:**
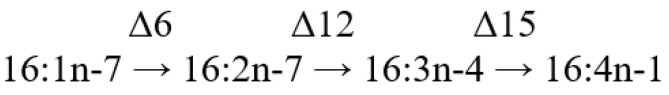
The synthesis of unsaturated acids 16:2n-7, 16:3n-4 and 16:4n-1 occurs in the zooxanthellae of soft corals, by sequential action of desaturases Δ6, Δ12 and Δ15 acting on the acid 16:1n-7.

**Figure 32 marinedrugs-21-00539-f032:**
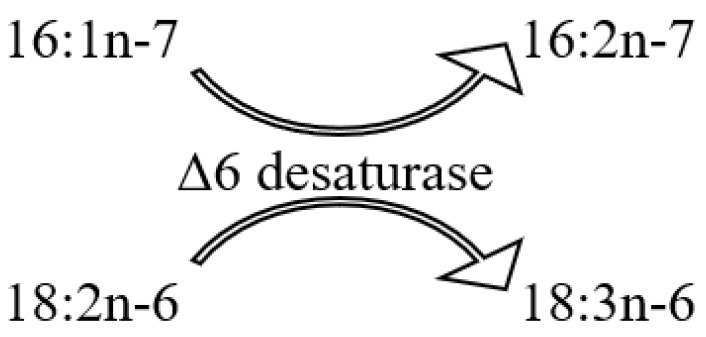
Acids 18:3n-6 and 16:2n-7 are formed from 18:2n-6 and 16:1n-7, respectively, by the action of Δ6 desaturase, when one enzyme acts on two different substrates.

**Figure 33 marinedrugs-21-00539-f033:**
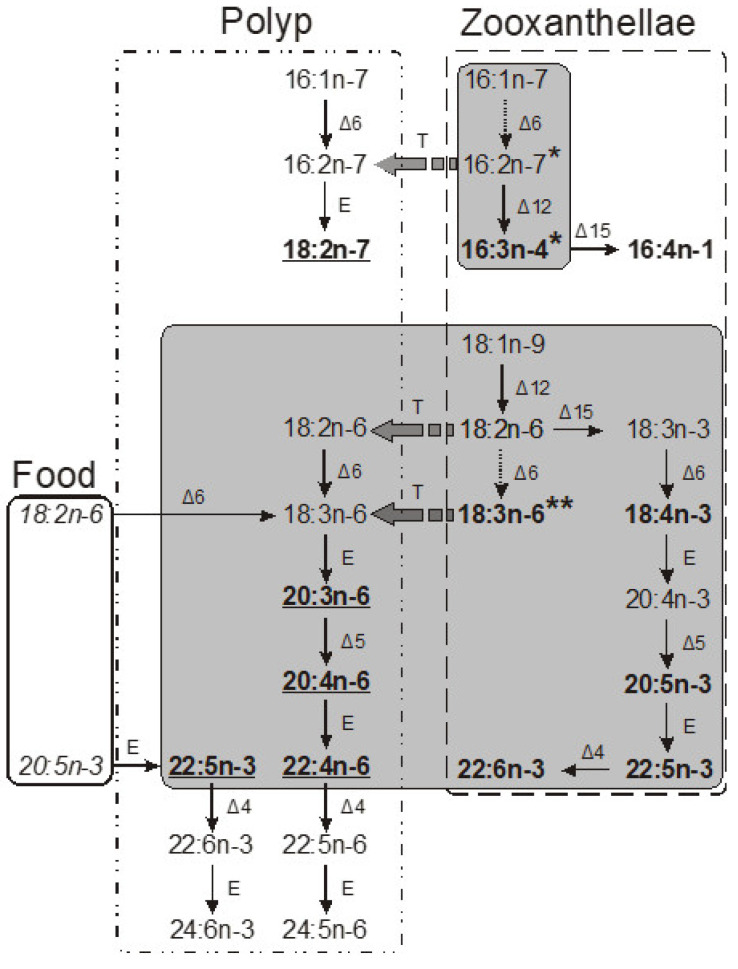
PUFA biosynthesis scheme in reef-building and soft corals, as well as possible transport of these acids between polyp and its intracellular symbiotic dinoflagellates (zooxanthellae). Acid names, which prevailed in polyps and zooxanthellae, are highlighted in bold and underlined bold, respectively. Components of reef-building coral are greyed out. Δ4, Δ5, Δ6, Δ12 and Δ15—corresponding desaturases; E—elongase; T—transport; * acids 16:2n-7 and 16:3n-4 are detected in zooxanthellae of hard corals as traces; ** acid 18:3n-6 is absence in some soft coral species [70].

**Figure 34 marinedrugs-21-00539-f034:**
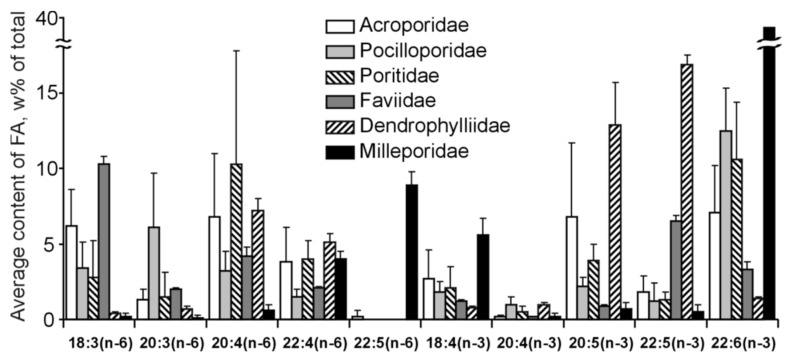
Average content of the main FAs (% of total) of hard coral total lipids, belonging to the families Acroporidae, Pocilloporidae, Poritidae, Faviidae, Dendrophylliidae, and hydrocoral total lipids of the family Milleporidae (the South China Sea, Vietnam) [100,104].

**Figure 35 marinedrugs-21-00539-f035:**
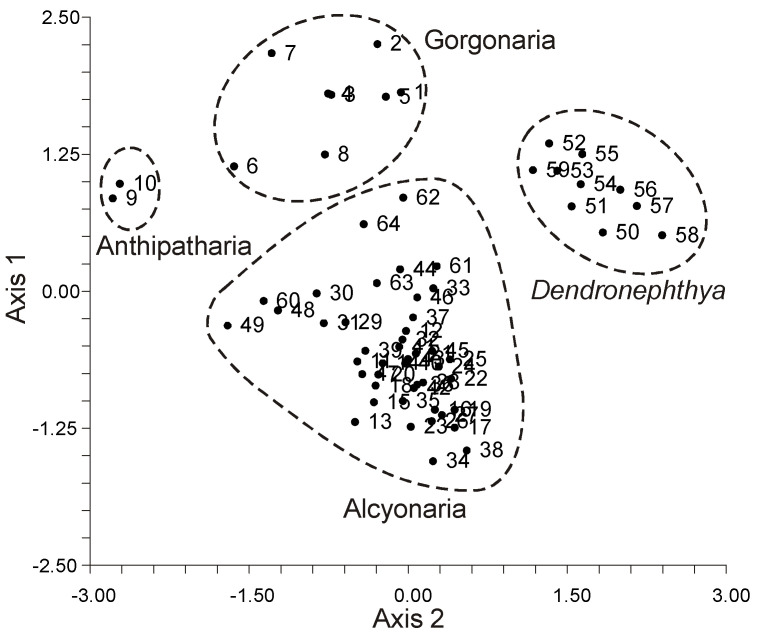
Principal component analysis (PCA) data on the total FA composition (square root of percentage) of 64 coral species from the coastal water of Vietnam. Curves were drawn manually to highlight each taxon domain. 1—*Euplexaura erecta*, 2—*Bebryce studeri*, 3—*Plexauridae* sp.1, 4—*Plexauridae* sp.2, 5—*Subergorgia suberosa*, 6—*Junceella fragilis*, 7—*J. juncea*, 8—*Rumphella aggregata*, 9—*Anthipatharia* sp.1, 10—*Anthipatharia* sp.2, 11—*Sinularia flexibilis*, 12—*S. capillosa*, 13—*S. leptoclados*, 14—*Sinularia* sp., 15—*S. polydactyla*, 16—*S. cruciata*, 17—*S. gibberosa*, 18—*S. querciformis*, 19—*S. leptoclados*, 20—*S. flexibilis*, 21—*S. cruciata*, 22—*S. cruciata*, 23—*S.* aff. *deformis*, 24—*S.* aff. *deformis*, 25—*S. lochmodes*, 26—*S.* cf. *muralis*, 27—*S. densa*, 28—*S. notanda*, 29—*Sarcophyton crassocaule*, 30—*S. crassocaule*, 31—*Lobophytum batarum*, 32—*L. pusillum*, 33—*L.* cf. *delectum*, 34—*Sarcophyton cinereum*, 35—*S. cinereum*, 36—*S. crassocaule*, 37—*S. ehrenbergi*, 38—*Sarcophyton* sp., 39—*S. digitatum*, 40—*S. ehrenbergi*, 41—*S. buitendijki*, 42—*S. cinereum*, 43—*S.* aff. *crassum*, 44—*S buitendijki*, 45—*S. trocheliophorum*, 46—*S. acutum*, 47—*S. elegans*, 48—*S. crassocaule*, 49—*S.* aff. *glaucum*, 50—*Dendronephthya crystallina*, 51—*D. aurea*, 52—*Dendronephthya* sp.1, 53—*D. aurea*, 54—*D. gigantea*, 55—*D. crystallina*, 56—*Dendronephthya* sp. 2, 57—*Dendronephthya* sp. 3, 58—*Dendronephthya* sp. 4, 59—*D.* aff. *involuta*, 60—*Dampia* sp., 61—*Cladiella laciniosa*, 62—*Litophyton* sp., 63—*Cespitularia* sp., 64—*Clavularia* sp. [105].

**Figure 36 marinedrugs-21-00539-f036:**
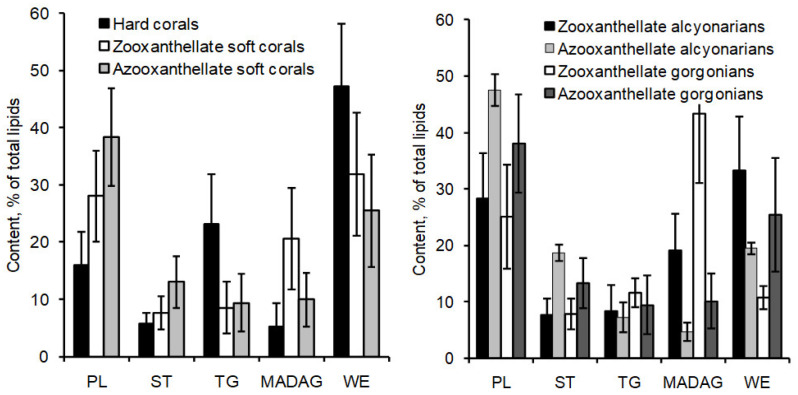
Content of main lipid classes (% of total lipids, mean ± SD) in different coral groups. WE, waxes; MADAG, monoalkyldiacyl-glycerols; TG, triacylglycerols; ST, sterols; PL, polar lipids.

**Figure 37 marinedrugs-21-00539-f037:**
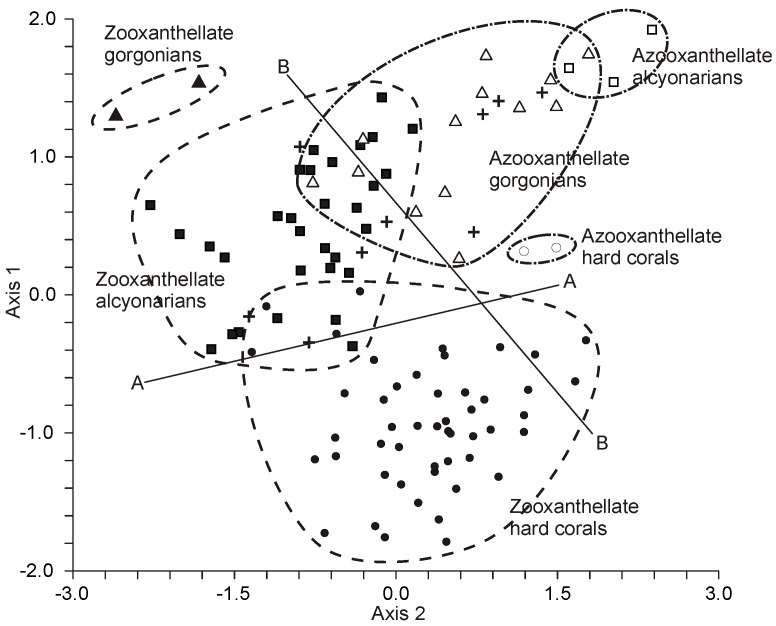
PCA of the data on principal lipid classes composition in 110 samples (93 species) of Vietnam corals (the South China Sea). Curves are drawn by hand to highlight the regions of each group. (●) zooxanthellate hard corals; (■) zooxanthellate alcyonarians; (▲) zooxanthellate gorgonians; (○) azooxanthellate hard corals; (□) azooxanthellate alcyonarians; (Δ) azooxanthellate gorgonians.

**Figure 38 marinedrugs-21-00539-f038:**
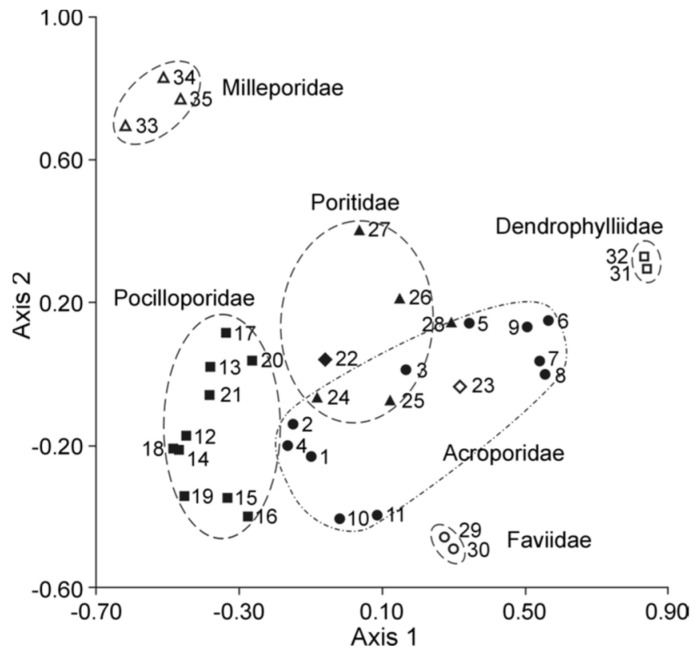
MSA data on variables (square root of 10 principal PUFA’s content values) measured for 35 hard coral samples from Vietnam and Seychelles. Numbers of coral samples are presented in the Appendix A. Lines, limiting domains of certain coral families, are made by hand. (●) Acroporidae; (■) Pocilloporidae; (▲) Poritidae; (○) Faviidae; (□) Dendrophylliidae; (Δ) Milleporidae; (♦) Pectiniidae; (◊) Fungiidae [103].

**Figure 39 marinedrugs-21-00539-f039:**
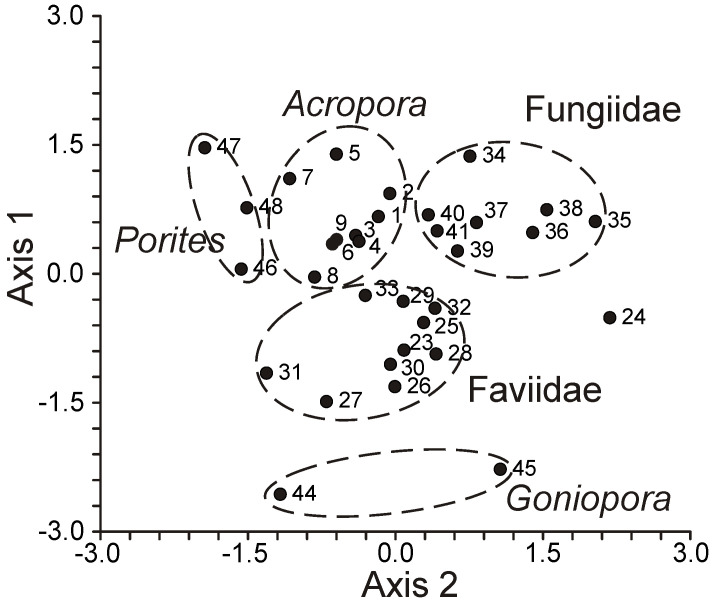
PCA data on nine variables (square roots of nine principal FA’s percentage) obtained for four hard coral families of Vietnam (Acroporidae, Poritidae, Faviidae, and Fungiidae (*Porites* + *Goniopora*)). Numbers of coral samples are presented in the Appendix A. Lines, limiting domains of separate coral families, were made by hand [68].

**Figure 40 marinedrugs-21-00539-f040:**
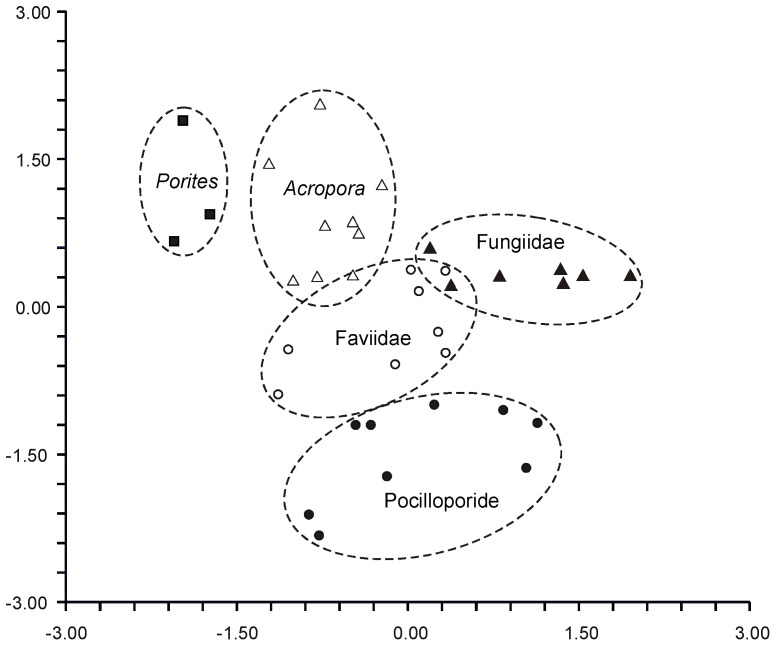
PCA data on nine variables (square roots of ten principal PUFA’s percentage) obtained for representatives of five hard coral families of Vietnam (Acroporidae, Poritidae, Faviidae, Fungiidae and Pocilloporidae). Lines, limiting domains of separate coral families, were made by hand.

**Figure 41 marinedrugs-21-00539-f041:**
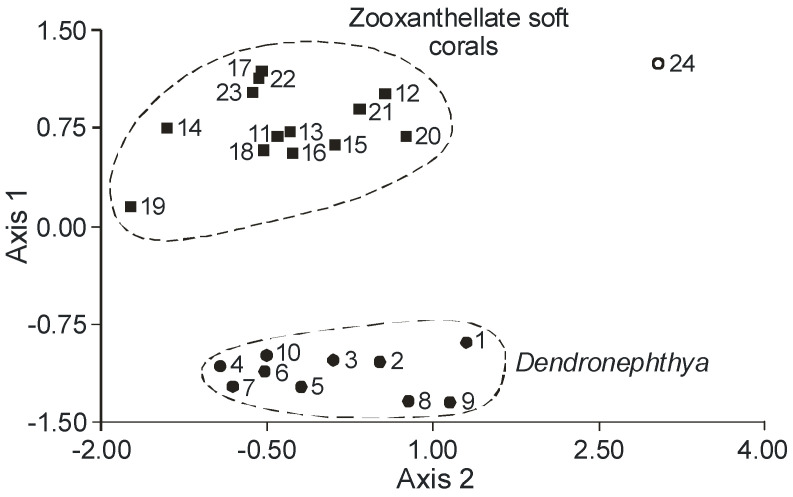
PCA data on nine variables (square root of nine principal FA’s content) measured for 24 soft coral samples of Vietnam. Lines, limiting domains of symbiotic and non-symbiotic soft corals, were made by hand. (●) *Dendronephthya*; (■) zooxanthellate species (*Sarcophyton*, *Lobophytum*, *Cladiella*, *Lytophyton*, *Cespitularia* and *Clavularia*); (○) symbiotic hard coral *Caulastrea tumida* [92].

**Figure 42 marinedrugs-21-00539-f042:**
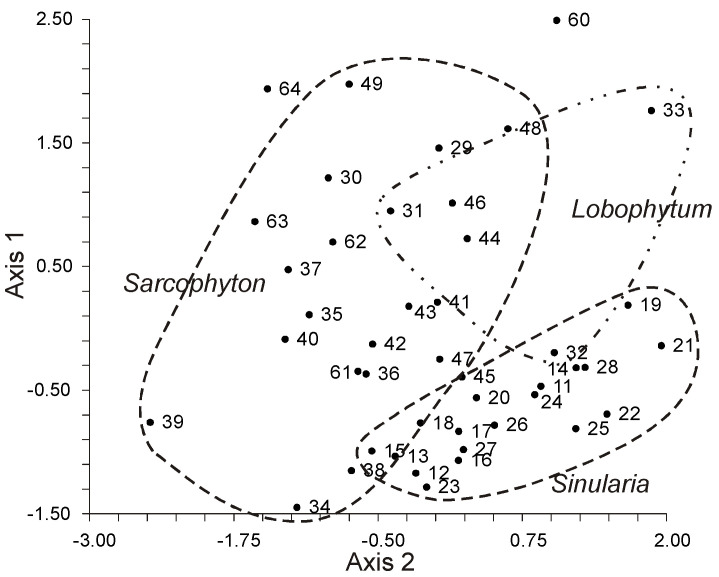
PCA data on 11 PUFAs (square roots of percentage value) of alcyonarian *Sinularia*, *Lobophytum*, and *Sarcophyton* from coastal water of Vietnam. Curves are made by hand to highlight domain of each taxon. 11—*Sinularia flexibilis*, 12—*S. capillosa*, 13—*S. leptoclados*, 14—*Sinularia* sp., 15—*S. polydactyla*, 16—*S. cruciata*, 17—*S. gibberosa*, 18—*S. querciformis*, 19—*S. leptoclados*, 20—*S. flexibilis*, 21—*S. cruciata*, 22—*S. cruciata*, 23—*S.* aff. *deformis*, 24—*S.* aff. *deformis*, 25—*S. lochmodes*, 26—*S.* cf. *muralis*, 27—*S. densa*, 28—*S. notanda*, 31—*Lobophytum batarum*, 32—*L. pusillum*, 33—*L.* cf. *delectum*, 29—*Sarcophyton crassocaule*, 30—*S. crassocaule*, 34—*S. cinereum*, 35—*S. cinereum*, 36—*S. crassocaule*, 37—*S. ehrenbergi*, 38—*Sarcophyton* sp., 39—*S. digitatum*, 40—*S. ehrenbergi*, 41—*S. buitendijki*, 42—*S. cinereum*, 43—*S.* aff. *crassum*, 44—*S. buitendijki*, 45—*S. trocheliophorum*, 46—*S. acutum*, 47—*S. elegans*, 48—*S. crassocaule*, 49—*S.* aff. *Glaucum*, 5, 60—*Dampia* sp., 61—*Cladiella laciniosa*, 62—*Litophyton* sp., 63—*Cespitularia* sp., 64—*Clavularia* sp. [105].

**Figure 43 marinedrugs-21-00539-f043:**
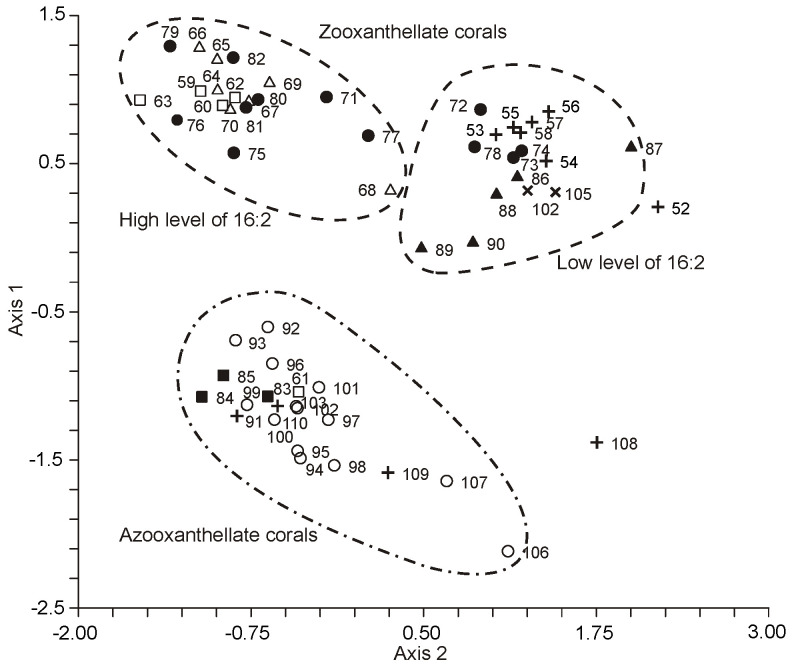
Results of PCA of 55 variables (square roots of 55 FA’s percentage values) for soft corals from the coastal water of Vietnam. Numbers of coral samples are presented in the Appendix A. Curves are made by hand to highlight each taxon domain. (●) *Sinularia*, (■) *Dendronephthya*, (▲) *Lemnalia* and *Nephthea*, (○) azooxanthellate gorgonians, (□) *Lobophytum*, (∆) *Sarcophyton*, (x) zooxanthellate gorgonians, (+) other species.

**Figure 44 marinedrugs-21-00539-f044:**
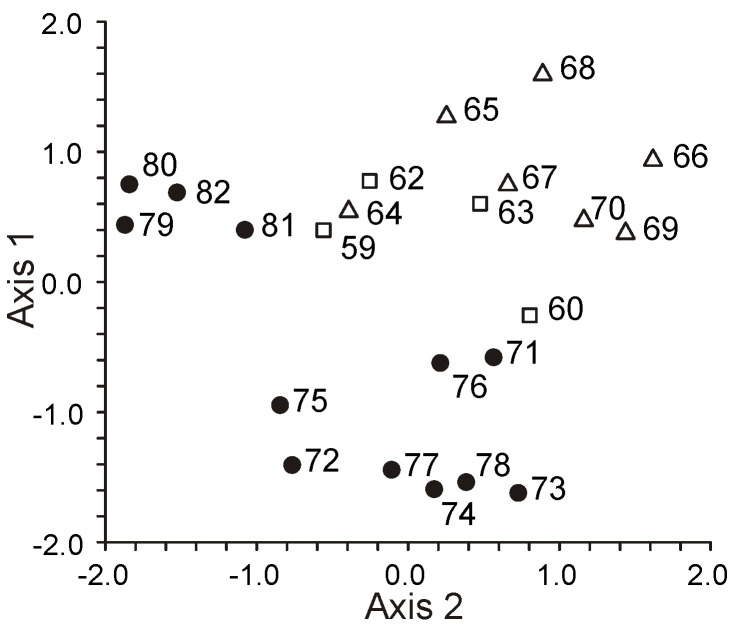
Results of PCA of 10 variables (square roots of 10 FA’s percentage values) for three genera of soft corals from the coastal water of Vietnam. Numbers of coral samples are presented in the Appendix A. (●) *Sinularia*, (□) *Lobophytum*, (∆) *Sarcophyton*.

**Figure 45 marinedrugs-21-00539-f045:**
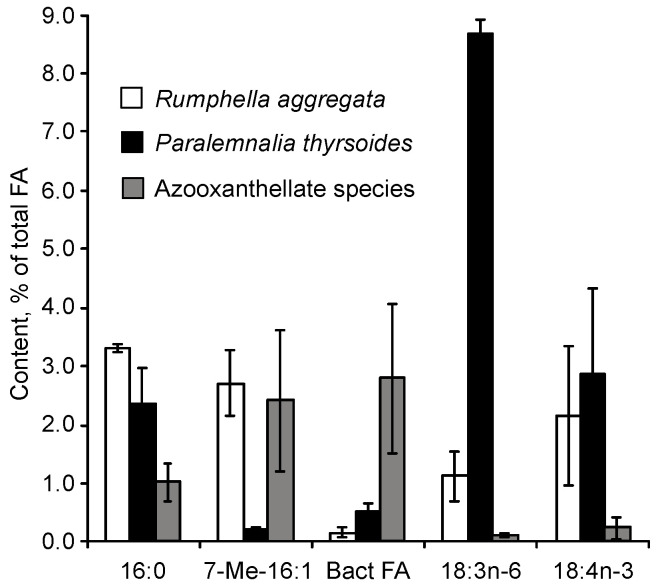
An average content of principal FAs (% of the total FA) in gorgonians *Rumphella aggregata*, *Paralemnalia thyrsoides* (zooxanthellate species), and azooxanthellate gorgonians (*Acabaria erithraea*, *Acanthogorgia isoxia*, *Chironephthya variabilis*, *Echinogorgia* sp., *Ellisella plexauroides*, *Menella praelonga*, *Bebryce studeri*). Bacterial FA—the sum of ai-15:0, ai-16:0, i-17:0, ai-17:0, 17:0, br-18:0, ai-19:0, and 19:0.

**Figure 46 marinedrugs-21-00539-f046:**
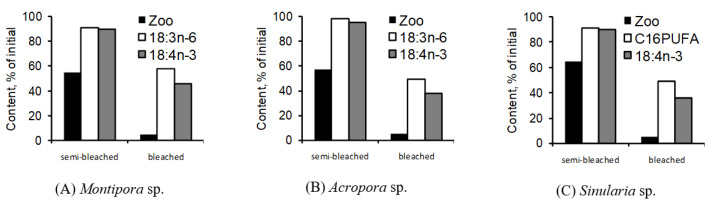
Change in zooxanthellae levels and their marker PUFA levels (% of the initial level) in total lipids of (**A**) *Montipora* sp., (**B**) *Acropora* sp. and (**C**) *Sinularia* sp. during the experimental colony bleaching (33 °C, 2 days).

**Figure 47 marinedrugs-21-00539-f047:**
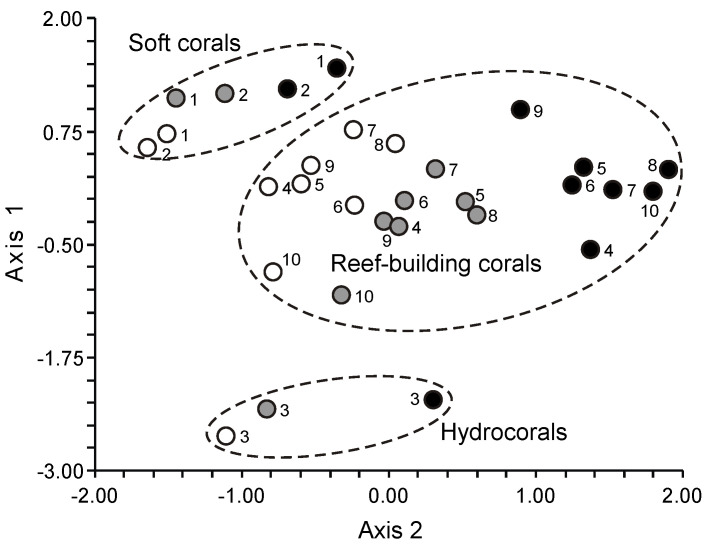
The result of PCA carried out on eight variables (16:3n-4, 16:4n-1, 18:3n-6, 18:4n-3, 20:4n-6, 20:5n-3, 22:5n-6, 22:6n-3; square roots of FA’s percentage value) for intact colonies, pure zooxanthellae, and the host tissues of ten cnidarian species from shallow water of Vietnam. 1, *Sinularia* cf. *capitalis*; 2, *Sinularia polydactila*; 3, *Millepora platyphylla*; 4, *Montipora faliosa*; 5, *Montipora digitata*; 6, *Pocillopora damicornis*; 7, *Acropora intermedia*; 8, *Acropora muricata*; 9, *Porites cylindrica*; 10, *Pavona decussata*. Objects: white, the host organism; grey, whole colonies; black, zooxanthellae [122]. Some species of these coelenterates are shown in Figure 48.

**Figure 48 marinedrugs-21-00539-f048:**
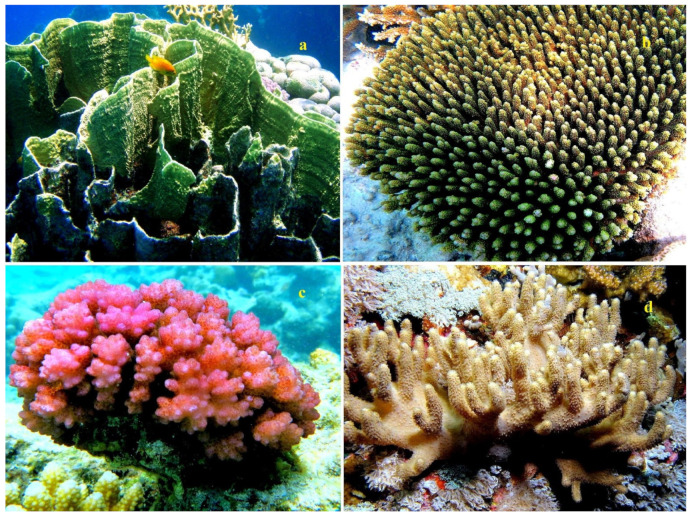
Some types of coral living in the Pacific Ocean off the coast of Vietnam: (**a**), *Millepora platyphylla*, (**b**), *Acropora millepora*, (**c**), *Pocillopora damicornis*, and (**d**), *Sinularia polydactyla*. Representatives of these corals produce various lipids, fatty acids, and other biologically active metabolites.

**Table 1 marinedrugs-21-00539-t001:** The total lipid content (an average ± SD, *n* = 5) (% DW) in the hard corals from the Caribbean (*) and Red Seas (**) [35].

Species	Lipid Content
*Porites porites* *	12.39 ± 1.06
*Montastrea annularis* *	31.87 ± 10.01
*Pocillopora verrucosa* **	10.70 ± 2.80
*Stylophora pistillata* **	17.03 ± 4.38
*Goniastrea retiformis* **	10.87 ± 3.47

**Table 2 marinedrugs-21-00539-t002:** The total lipid content (% DW) in hard corals (Okinawa Is., Japan) [36].

Species	Lipid Content
*Pocillopora damicornis*	30.8
*Pocillopora verrucosa*	14.1
*Stylophora pistillata*	20.8
*Montipora aequituberculata*	22.5
*Acropora microphthalma*	28.6
*Porites lutea*	20.1
*Porites cylindrical*	21.1
*Fungia fungites*	23.2
*Galaxea fascicularis*	37.0
*Goniastrea aspera*	29.6
*Oulastrea crispate*	19.3
*Tubastrea* sp.	15.6

**Table 3 marinedrugs-21-00539-t003:** The total lipid content (mg/g WW) in the soft corals from the Vietnamese shallow waters [46].

Subclass	Species	Lipid Content
Gorgonacea	*Euplexaura erecta*	4.6
	*Subergorgia suberosa*	12.7
	*Junceella fragilis*	14.2
	*Junceella juncea*	3.3
	*Rumphella aggregate*	4.5
	*Nicaule crucifera*	13.8
	*Plexauridae* spp. 1	9.8
	*Plexauridae* spp. 2	2.8
Alcyonacea	*Sinularia capillosa*	19.4
	*Sarcophyton crassocaule*	4.8
	*Sinularia leptoclados*	20.0
	*Sarcophyton* aff. *glaucum*	22.9

**Table 4 marinedrugs-21-00539-t004:** The lipid content (% DW ± SD, *n* = 7 ÷ 36) in some species of the Hawaiian corals depending on the season [52].

Species	Season			
	2 April 1982	6 July1983	2 April 1984	5 May 1984
*Montipora verrucosa*	-	46.85 ± 6.86	42.15 ± 7.03	43.86 ± 4.81
*Porites compressa*	-	33.00 ± 3.98	30.82 ± 5.85	29.89 ± 5.28
*Cyphastrea ocellina*	-	40.70 ± 5.95	-	-
*Pocillopora meandrina*	-	32.28 ± 4.96	-	33.50 ± 2.65
*Pocillopora damicornis* “Y” type	40.73 ± 7.55	41.32 ± 8.06	36.14 ± 8.06	-
*Pocillopora damicornis* “B” type	35.50 ± 6.20	36.89 ± 6.32	34.31 ± 4.56	-

**Table 5 marinedrugs-21-00539-t005:** The total lipid content (% DW ± SD, *n* = 10) in the hard corals *Porites porites*, *Montastrea annularis*, and *Siderastrea siderea* (Barbados Is.) depending on the depth of habitat [34].

Reef Name and Depth (m)	*Porites porites*	*Montastrea annularis*	*Siderastrea siderea*
Driftwood			
3	11.18 ± 1.44	31.13 ± 1.98	-
6	11.0 ± 1.99	29.01 ± 6.57	-
Bellairs			
3	11.50 ± 1.54	26.24 ± 3.69	27.02 ± 2.20
6	10.81 ± 1.43	27.17 ± 7.06	-
Barrier reef			
13	8.59 ± 1.36	32.70 ± 4.17	25.55 ± 3.28
20	-	24.35 ± 6.61	27.25 ± 1.31
30	-	30.80 ± 4.18	34.78 ± 3.23

**Table 6 marinedrugs-21-00539-t006:** The total lipid content (% DW) in deep-water cnidarians from regions of Newfoundland and Labrador (Atlantic Ocean) [53].

Species	Depth, m	Lipid Content
*Anthomastus grandiflorus*	578–1277	10.5
*Gersemia rubiformis*	51–1135	12.1
*Capnella florida*	59–1302	16.0
*Acanella arbuscula*	296–1154	6.9
*Paramuricea* spp.	644–1193	3.9
*Primnoa resedaeformis*	162–1157	6.5
*Paragorgia arborea*	448–1277	8.5
*Bathypates* spp.	876–1287	27.6
Sea feathers	663–969	16.2

**Table 7 marinedrugs-21-00539-t007:** The total lipid composition (% of the sum) in cnidarians, Okinawa Is., Japan [53].

Species	PL	ST	FFA	TG	MADAG	WE	SE
*Pocillopora damicornis*	18.8	9.6	10.1	21.2	1.5	26.4	0.2
*Pocillopora verrucosa*	27.8	13.1	12.6	14.9	1.5	15.8	0.7
*Stylophora pistillata*	21.9	8.5	10.6	19.2	1.0	21.8	0.2
*Montipora aequituberculata*	22.0	9.4	7.3	22.7	4.4	10.0	0.4
*Acropora microphthalma*	22.9	9.4	8.5	16.2	2.7	17.4	0.2
*Porites lutea*	17.7	10.3	11.7	21.0	7.5	18.0	0.2
*Porites cylindrica*	19.6	9.4	11.6	22.8	7.4	17.6	0.0
*Fungia fungites*	21.4	10.3	9.1	15.0	2.4	31.0	0.0
*Galaxea fascicularis*	14.3	11.3	9.1	30.4	3.9	21.3	0.0
*Goniastrea aspera*	17.9	10.5	9.8	16.1	3.0	31.4	0.2
*Oulastrea crispata*	18.9	11.3	11.0	22.2	4.2	20.0	0.9
*Tubastrea* sp.	20.3	12.8	12.7	21.9	6.6	9.1	7.5
*Lobophytum crassum*	23.7	12.6	7.6	8.9	9.5	14.6	0.4
*Millepora murrayi*	25.1	10.4	12.3	13.9	9.3	15.3	0.0

**Table 8 marinedrugs-21-00539-t008:** Lipid composition (% of total) in deep-water cnidarians from regions of Newfoundland and Labrador (the Atlantic Ocean) [53].

Species	Depth (m)	PL	ST	FFA	TG	MADAG	WE	SE
*Anthomastus grandiflorus*	578–1277	41.2	9.2	9.4	8.1	13.7	12.1	6.3
*Gersemia rubiformis*	51–1135	40.4	10.4	10.4	5.6	15.1	14.6	3.5
*Capnella florida*	59–1302	31.0	9.4	9.8	10.4	19.7	14.5	5.3
*Acanella arbuscula*	296–1154	40.5	12.8	14.9	6.1	11.4	10.1	4.2
*Paramuricea* spp.	644–1193	29.0	14.6	14.8	3.9	12.2	12.3	13.2
*Primnoa resedaeformis*	162–1157	34.9	11.3	11.7	6.8	12.0	15.0	8.7
*Paragorgia arborea*	448–1277	34.5	10.7	8.5	8.9	10.6	10.0	16.8
*Bathypates* spp.	876–1287	25.1	9.8	11.7	12.3	15.4	14.6	11.1
Sea pens	663–969	25.1	17.0	12.3	10.3	8.6	21.7	5.0

**Table 9 marinedrugs-21-00539-t009:** Composition of alkylglycerols and aliphatic alcohols (in reference to C_18_ alcohol) of the fraction of unsaponifiable lipids of gorgonians [43].

Species	Alkylglycerols			Aliphatic Alcohols		
	C_16_	C_18_	C_18:1_	C_16_	C_18_	C_18:1_
*Eunicea laciniata*	0.062	1	0.02	1.6	1	0.57
*Eunicea tournejorte*	0.34	1	0.02	1.6	1	0.12
*Gorgonia flabellum*	0.89	1	0	2.1	1	0.12
*Gorgonia ventalina*	0.95	1	0	3.2	1	0.49
*Muricea atlantica*	0.93	1	0.015	1.6	1	0.032
*Muriceopsis flarida*	0.22	1	0	7.8	1	0.42
*Plexaura homomalla*	0.39	1	0.008	0.36	1	0.058
*Pseudopterogorgia acerosa*	0.25	1	0	2.9	1	0.65
*Pterogorgia anceps*	1.4	1	0	9.5	1	1.4
*Rumphella antipathes*	0.77	1	0	3.4	1	0.37

**Table 10 marinedrugs-21-00539-t010:** Lipid composition (% of total) of three species of Vietnam gorgonian corals [87].

Content	*Psammogorgia nodosa*	*Bebryce indica*	*Mopsella aurantia*
Lipids in WW, %		0.95	1.03
Lipids in DW, %		1.24	3.80
Phospholipids, %	14.6	28.8	32.7
Diphosphatidylglycerol	-	-	1.1 ± 0.1
Phosphatidylinositol	3.1 ± 0.8	3.9 ± 0.4	3.4 ± 0.3
Phophatidylethanolamine	27.7 ± 1.4	5.4 ± 0.9	13.6 ± 0.4
Phosphatidylserine	19.8 ± 1.3	10.1 + 0.4	15.7 ± 0.5
Phophatidylcholine	26.2 ± 0.8	15.1 ± 0.4	21.4 ± 1.1
Ceramideaminoethyl- phosphonate	10.5 ± 0.8	13.7 ± 1.6	9.8 ± 0.5
Lysophophatidylcholine	2.4 ± 1.1	8.7 ± 1.0	8.6 ± 0.4
Lysophosphatidylserine	-	11.7 ± 0.8	2.7 ± 0.5
X1	-	-	1.4 ± 0.1
X2	-	21.84 ± 1.4	14.7 ± 1.3
X3	10.3 ± 1.0	9.6 ± 1.0	7.6 ± 1.3

**Table 11 marinedrugs-21-00539-t011:** Phospholipid composition (% of total) of corals of the genus *Sinularia* from the Vietnam shallow waters (the South China Sea) (*n* = 5, SD ≤ 1.5%) [88].

Species	PC	LPC	PE	LPE	PS	LPS	PI	DPG	CAEP	PA	X1
	Summer										
*S. polydactyla*	34.9	9.9	17.8	4.3	15.0	0	4.9	5.0	1 4.6	3.4	0
*S. gibberosa*	36.4	6.3	18.3	3.1	14.2	1.1	5.2	4.1	8.1	1.1	0
*S. cruciata*	28.1	5.8	30.4	2.8	13.4	0	4.2	2.5	8.8	1.0	3.0
*S. abrupta*	30.9	12.9	16.6	7.5	15.5	0	4.4	6.6	5.5	0	0
*S. rigida*	40.4	13.2	19.2	2.5	11.3	0	3.0	2.6	7.8	0	0
	Winter										
*S. polydactyla*	3.5	18.8	6.9	6.8	17.6	9.0	2.9	1.0	20.6	1.9	10.7
*S. gibberosa*	9.8	127	3.9	14.3	12.0	11.4	2.0	1.1	19.8	0.7	0
*S. cruciata*	3.0	14.7	6.2	9.8	17.5	14.4	1.2	0.4	6.8	1.1	25.5
*S. capillosa*	15.6	29.3	17.1	2.2	18.0	0	3.9	1.8	9.0	3.1	0
*S. flexibilis*	8.6	25.9	14.3	2.1	17.8	2.4	3.1	0.9	23.1	0.6	2.0
*S. querciformis*	25.6	10.1	24.4	4.4	19.3	0	2.6	0.6	11.3	0	1.7

**Table 12 marinedrugs-21-00539-t012:** Phospholipid composition (% of total) of corals of the genus *Lobophytum* from the Vietnam shallow waters (the South China Sea) (*n* = 5, SD ≤ 1.5%) [88].

Species	PC	PC *	LPC	PE	PE *	LPE	PS	PS *	LPS	PI	DPG	CAEP	PA	X2
	Summer													
*L. carnatum*	22.1	7.0	4.7	3.1	12.8	3.1	7.9	5.3	3.7	4.1	4.0	13.3	4.5	4.4
*L. batarum*	24.2	6.2	5.0	7.6	10.9	4.0	11.5	5.4	0	4.5	4.7	7.1	6.3	2.5
*L. undatum*	20.5	5.3	5.5	1.1	16.2	4.1	11.8	5.5	0	3.1	4.2	16.0	4.3	2.3
	Winter													
*L. carnatum*	24.6	-	10.0	18.8	-	9.1	15.7	-	3.4	2.1	1.1	10.6	0.7	3.8
*L. batarum*	28.5	-	8.4	28.0	-	2.6	16.8	-	3.0	3.4	2.0	1.8	1.5	2.3
*L. undatum*	26.7	-	7.4	27.0	-	3.0	16.9	-	3.9	3.3	1.5	3.7	1.7	4.9
*L. roxasi*	23.7	-	18.4	10.7	-	3.8	15.8	-	4.4	2.0	1.5	9.3	0.6	9.2
*L. sarcophytoides*	27.1	-	7.5	23.2	-	4.8	11.6	-	3.6	7.1	3.3	6.5	0	5.2

(*)—plasmalogen form of phospholipids.

**Table 13 marinedrugs-21-00539-t013:** Phospholipid composition (% of total) of corals of the genus *Sarcophyton* from the Vietnam shallow waters (the South China Sea) in the summer period (*n* = 5, SD ≤ 1.5%) [88].

Species	PC	PC *	LPC	PE	PE *	LPE	PS	PS *	LPS	PI	DPG	CAEP	PA	X1	X2
*S. ehrenbergi*	7.0	3.9	29.6	3.7	9.4	6.5	9.9	8.9	0	5.2	1.2	9.2	3.2	2.4	0
*S. molle*	29.2	0	4.1	5.3	0	5.2	16.1	0	8.4	2.6	2.5	6.6	9.9	10.1	0
*S. tereum*	24.3	3.8	5.8	4.3	26.1	2.8	10.6	2.9	0	3.0	4.2	2.5	8.3	0.8	0
*S. crassocaule*	32.8	3.8	7.6	2.0	12.5	4.8	11.3	3.2	3.6	1.6	2.1	8.3	3.1	2.8	0
*S. puertogalerae*	24.2	2.7	6.8	1.5	15.5	6.3	12.5	3.6	0	2.3	2.7	13.9	4.5	3.6	0
*S. moseri*	28.9	6.3	3.5	2.3	18.1	5.9	10.2	5.5	0	1.9	2.6	11.0	3.8	0.4	0
*S. digitatum*	28.1	4.6	6.4	2.9	15.5	5.1	13.7	4.4	0	3.1	1.3	9.0	1.8	2.9	1.2
*S. poculiforme*	34.2	5.7	4.5	3.8	19.8	1.8	10.1	5.4	0	2.7	2.0	7.0	1.3	0	1.7
*S. cinereum*	24.5	5.4	9.1	3.2	19.1	1.8	10.6	6.8	0	5.6	2.5	8.4	2.2	0	1.7

*—plasmalogen form of phospholipids.

**Table 14 marinedrugs-21-00539-t014:** Phospholipid composition (% of total, mean ± SD, *n* = 5) of the soft coral *Gersemia rubiformis* (Bering Sea) [48].

Phospho- and Phosphonolipids	Content
Phophatidylcholine	31.4 ± 6.2
Phophatidylethanolamine	25.6 ± 1.5
Phosphatidylserine	14.1 ± 3.2
Ceramideaminoethylphosphonate	15.6 ± 1.4
Ceramide-2-N-methylaminoethylphosphonate	13.3 ± 3.0

**Table 15 marinedrugs-21-00539-t015:** The total lipid content (%) of *Millepora dichotoma* and *Millepora platyphylla* from the coastal waters of Vietnam (the South China Sea) and *Millepora murray* from Okinawa Is. [36,68].

Species	PL	ST	FFA	TG	MADAG	WE
*Millepora murrayi*	25.1	10.4	12.3	13.9	9.3	15.3
*Millepora dichotoma*	19.2	5.6	0.7	22.8	17.4	26.7
*Millepora platyphylla*	15.1	5.4	0.9	22.0	18.6	37.1

**Table 16 marinedrugs-21-00539-t016:** The total lipid FA composition (%) of *Millepora* hydrocorals of Vietnam (B) and Seychelles (C) [100].

Fatty Acid	*Millepora* sp.	*Millepora* *platyphylla*	*Millepora* *dichotoma*
	B	C	C
14:0	0.6	0.8	1.4
16:0	6.3	17.6	18.9
16:1n-7	-	0.2	-
16:2	0.2	0.8	0.4
18:0	7.1	21.3	19.4
18:1n-9	1.4	2.4	3.2
18:2n-6	0.6	0.4	0.7
18:3n-6	-	0.2	0.4
18:4n-3	6.9	5.2	4.8
20:1	0.3	6.3	6.1
20:2n-6	-	0.5	0.4
20:3n-6	0.4	-	-
20:4n-6	0.4	0.3	1.0
20:4n-3	0.4	0.2	-
20:5n-3	1.1	0.4	0.6
22:4n-6	3.8	4.6	3.7
22:5n-6	8.3	10.0	8.5
22:5n-3	-	0.7	0.9
22:6n-3	61.5	27.5	28.0

**Table 18 marinedrugs-21-00539-t018:** Ratio of 18:1n-9 to 18:4n-3 in total lipids of thylakoid membranes of heat-sensitive and heat-tolerant clones of *Symbiodinium* spp. [118].

Clone	Sensitivity	18:1n-9/18:4n-3
CCMP 828	heat-tolerant	2.05
CCMP 830	heat-tolerant	0.87
CCMP 421	heat-tolerant	1.22
EIL2	heat-tolerant	2.68
CCMP 1633	heat-sensitive	0.28
CCMP 827	heat-sensitive	0.26
CCMP 831	heat-sensitive	0.24

**Table 19 marinedrugs-21-00539-t019:** Percentage of main fatty acids extracted from (A) whole cells and (B) a fraction enriched in photosynthetic membranes of *Symbiodinium microadriaticum* A1 and *Symbiodinium* sp. C1, grown at 24 °C.

Fatty Acid	*Symbiodinium* sp. (C1)		*S. microadriaticum* (A1)	
	A	B	A	B
16:0	30.0	32.6	34.8	40.7
16:1n-9	2.1	2.2	1.8	2.5
18:0	19.9	21.4	20.9	20.2
*trans*-18:1n-9	4.7	6.0	3.1	1.8
*cis*-18:1n-9	11.1	10.7	10.2	5.1
*cis*-18:1n-7	4.7	5.2	2.0	2.8
18:2n-6	3.1	2.8	5.3	1.5
18:3n-3	14.0	8.5	4.6	4.8
18:4n-3	4.9	5.4	9.7	10.4
22:6n-3	4.4	4.8	7.1	9.9

**Table 20 marinedrugs-21-00539-t020:** The main lipid classes content (% of total lipids) in intact colonies, zooxanthellae, and the host tissue of the soft coral *Sinularia* sp. (mean ± SD, *n* = 3) [68].

Lipid Class	Intact Colonies	Zooxanthellae	Host Tissue
WE *	19.8 ± 3.3	13.9 ± 1.6	31.7 ± 0.9
MADAG *	18.4 ± 1.1	0.8 ± 0.4	35.0 ± 2.3
TG	7.3 ± 2.1	7.4 ± 2.2	10.7 ± 0.5
FFA	4.8 ± 3.1	7.7 ± 3.2	3.5 ± 0.4
ST	13.9 ± 3.7	13.9 ± 5.1	5.4 ± 0.8
PL*	35.9 ± 2.9	56.2 ± 1.6	13.6 ± 1.0

* Significant difference between zooxanthellae and the host tissue (*p* < 0.01).

**Table 21 marinedrugs-21-00539-t021:** List of the host species sampled their identification and clade designation of the symbiont isolated from the host species [122].

Systematic Affiliation of Cnidarians	Host Species Sampled	Clade Designation of Symbiont
Hydrozoa (hydrocorals)	*Millepora platyphylla* Hemprich & Ehrenberg, 1834	C66a
Anthozoa: Octocorallia (soft corals)	*Sinularia* cf. *capitalis* (Pratt, 1903)	C71a
	*Sinularia polydactila* Ehrenberg, 1834	C71a
Anthozoa: Hexacorallia (hard corals)	*Acropora intermedia* (Brook, 1891)	C71a
	*Acropora muricata* (Linnaeus, 1758)	C1f/C27
	*Montipora digitata* (Dana, 1846)	D/D1a
	*Montipora foliosa* (Pallas, 1766)	C60
	*Pavona decussata* (Dana, 1846)	C72, C1b
	*Pocillopora damicornis* (Linnaeus, 1758)	D/D1a
	*Porites cylindrica* Dana, 1846	C15, C16, C17

**Table 22 marinedrugs-21-00539-t022:** Results of gas-chromatographic and mass-spectrometric (*m*/*z*, electron impact ionization, 70 eV) analysis of F-acids methyl esters from Vietnam gorgonian corals. Full forms of acid names are provided in Appendix A [111].

Fatty Acid	Me-F18	diMe-F18-3	Me-F20	diMe-F20	diMe-F20-3	Me-F22	diMe-F22
	(**43**)	(**44**)	(**45**)	(**46**)	(**47**)	(**48**)	(**49**)
n *	7	9	9	9	11	11	11
m *	3	1	3	3	1	3	3
GC retention time, min	27.3	31.2	35.5	38.4	39.8	43.5	46.3
M^+^	322	336	350	364	364	378	392
[M−MeOH+H]^+^	291	305	319	333	333	347	361
[M−alkyl(C_2_H_5_)]^+^	-	307	-	-	335	-	-
[M−alkyl(C_4_H_9_)]^+^	265	-	293	307	-	321	335
[M−alkylester]^+^ base peak	165	151	165	179	151	165	179
Furan fragment	109	123	109	123	123	109	123

* n, number of carbon atoms in alkyl chain; m, number of carbon atoms in acyl chain (Figure 15).

**Table 23 marinedrugs-21-00539-t023:** Content of zooxanthellae marker FA (18:3n-6 and 18:4n-3) in total FAs (%) and the ratio of these acids in pure zooxanthellae and polyp tissues from hard corals and *Millepora* [122]. SD, symbiotic dinoflagellates (zooxanthellae).

Species	18:3n-6			18:4n-3		
	SD	Host	SD/Host	SD	Host	SD/Host
*Millepora platyphylla*	0.1	-	-	17.3	0.9	19.2
*Montipora faliose*	23.4	2.4	9.8	10.3	0.6	17.2
*Montipora digitata*	17.2	3.6	4.8	15.0	1.3	11.5
*Pocillopora damicornis*	4.8	4.4	1.1	10.3	0.7	14.7
*Acropora intermedia*	8.2	2.7	3.0	13	0.9	14.4
*Acropora muricata*	11.4	2.7	4.2	14.1	1.3	10.8
*Porites cylindrica*	8.2	2.8	2.9	13.1	0.9	14.6
*Pavona decussata*	12.4	0.3	41.3	29.5	1.3	22.7

**Table 24 marinedrugs-21-00539-t024:** Content of zooxanthellae marker FAs (16:3n-4, 16:4n-1, and 18:4n-3), as well as 16:2n-7 and 18:2n-7, (%) and the ratio of these acids in total FAs of pure zooxanthellae and polyp tissues of Vietnamese soft corals [69,134]. SD, symbiotic dinoflagellates (zooxanthellae).

Fatty Acids	*Sinularia* sp.			*Sinularia* cf. *capitalis*			*Sinularia polydactila*		
	SD	Host	SD/Host	SD	Host	SD/Host	SD	Host	SD/Host
16:2n-7	6.0	4.2	1.4	5.7	3.8	1.5	4.8	4.1	1.2
16:3n-4	4.5	1.1	4.1	4.1	1.1	3.7	4.2	1.1	3.8
16:4n-1	4.4	0.4	11.0	3.7	0.5	7.4	1.5	0.4	3.8
18:4n-3	13.3	1.3	10.2	10.6	1.2	8.8	9.1	1.3	7.0
18:2n-7	0.8	4.0	0.2	0.9	2.0	0.5	1.9	4.0	0.5

## Data Availability

Not applicable.

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
