# Peer review of "Coral Lipids"

_marinedrugs, 2023, doi:10.3390/md21100539_

Round 1

Reviewer 1 Report

Line 9 Probably meant symbiotic and _heterotrophic_

 Line 60 It is advisable to decipher abbreviations according to the text of the article the first time they are used

 Line 352 misprint “squalen”

 Line 374-375  “It is possible that it was caused by the method of analysis of the total lipid using TLC method.”

repeated word “method”

 Line 461-463 “High content of monoenoic FA was detected in the waxes of P. damicornis, P. verrucosa, Stylophora pistillata, P. lutea, O. crispata (37.3%) and Tubastrea sp. (41.8%).”

According to the table, the amount оf total MUFA in O. crispate =(32.7+37.3+1.7%), in Tubastrea sp. =(8.1+41.8+5.4%)

 Line 620 “Phospholipids Bebryce indica contained merely 5.4% of PE under full”

“of” missed

 Line 685 hereinafter - the term molecular species is usually used, not “molecule species”

 Line 687-689 “Separation and quantitative analysis of lipid molecule species is a very complicated task, which is solved using a combination of several analytical methods or gas chromatography-mass spectrometry method with tandem mass-detector.“

The main method for molecular species separation is HPLC, not GC.

 Line 1024 misprint “methyldokcosa”

 Line 1111/table 50. It would probably be more correct to say “FA in the polar lipids”, because these corals clearly contain zooxanthellae, and therefore, glycolipids.

 Line 1993 “. It is known that the composition of total lipids of animals

Perhaps it would be more accurate to say " lipid classes composition of animals"

 Line 2149 “Axis 2 was a discriminant

According to fig.29 Axis 1 is a discriminant.

 Line 2382 misprint “25:4n-6”, here should be 22:4n-6.

 Line 2585 misprint “Ration of 18:4n-3” , should be “Ratio”.

Author Response

Reviewer 1

Thank you for your detailed analysis of our article presented in Marine Drugs.

  We carefully read all your comments and carefully made changes to the text.

  All typos that you pointed out, as well as those we additionally found, have been corrected, both in the main text and in the Tables. All changes and additions are highlighted in blue.

  In agreement with the Marine Drugs office, we have removed more than 40 Tables from the main text in the Supporting material, and now the article is easy to read.

In addition, at our request, our colleagues from the Department of English Philology worked on the article, and therefore the quality of the article became good.

Reviewer 2 Report

Abstract ln 15  : "betaine" is not a lipid, not sure what you are thinking of, choline 

Please define all abbreviations used the first time they are used in the text   ln 60 HC ST, hydrocarbon and sterols?

define CAEP

ln 74 DW dry weight? Ln 76 FA

the abbreviation list at the end is missing several.

Ln 81 "used method" should be "method used" 

Table 1 legend  should be "mg/g DW of colony"?  can this be converted to % so it is easier to compare with data in tables 2 and 3

Ln 129: should read " soft corals is published much less often than"

Just checking that the coral pictures in figures 1-3, 36 are all the authors own images and not taken from other sources, if the latter then they should be  cited?

Table 4 is not mentions=ed in text  ln 134, also in legend define WW wet weight.

Table 5 cn the columns of the season please be put into the correct chronological order.

Ln 359. would C18 and C27, be C18 to C27?

Table 12 , what are the numbers % of total?

Ln 472 should read "depends upon"

Ln 645-8 organism names need to be in italics. and multiple places through out manuscript 

Ln 671 CMAEP. the full name should have "N-"methyl-......

Ln 697 define DMOX DMDS

Tables 27 and 28 and elsewhere  ai-fatty acid please define ai and i in legend.

Figure 13 and 20 "*" sig Dif  is shown, but by what statistical method?

Table 65 has no box around like the others? in the legend state that the numbers in bold refer to structures in Fig 14

Ln 1524  odd character between 1H and 13C.

Ln 1634.. autor citation rather than numbered citation

p88 where the authors are describing the elongate and desaturate possibilities   ln 1653-4, can I suggest they name the as "scheme n " and then they can be referred to properly from the text .

ln 2011 I think fig 26 should be 25 

ln 2164  fig 19 should be 29?

ln 2415 "species"

Author Response

Reviewer 2

Thank you for your detailed analysis of our article presented in Marine Drugs, “Coral Lipids”.

  We carefully read all your comments and carefully made changes to the text.

  All typos that you pointed out, as well as those we additionally found, have been corrected, both in the main text and in the Tables. All changes and additions are highlighted in blue.

We cannot bring Tables 1, 2 and 3 to a single denominator since there is not enough source data. Therefore, we left everything as it is in the original articles.

All abbreviations that are in the text are placed at the end of the article by the editors.

In Table 5 we have made changes and put everything in the correct order.

The names of organisms are in Italian.

In Tables 28 and 29, additions have been made about iso- and anteiso-fatty acids.

Figures 13 and 20 indicate the statistical method of analysis.

In agreement with the Marine Drugs office, we have removed more than 40 Tables from the main text in the Supporting material, and now the article is easy to read.

In addition, at our request, our colleagues from the Department of English Philology worked on the article, and therefore the quality of the article became very good.

Reviewer 3 Report

I have reviewed the manuscript entitled " Coral Lipids". This review touches on the coral-derived lipids, and furthermore, the author examined the gamut of fatty acids and their acylderivatives. However, I think it can not be published in Marine Drugs in this edition for the following reasons.

1. The whole manuscript is too long and full of nonessential information, such as in parts of “2. The content of total lipids in corals”, these data will be more readable from tables1-7, but the authors have explained them one by one in the manuscript, and need to rewrite it.

2. Authors need to give the correct literature references, such as in Table 7, marked as [53], however, it seems the wrong reference, because I didn’t screen the content of Table 7 published in [53].

3. The paper is not well organized, and much of the content is redundant, for example, many tables need to be shown in the Supporting Information rather than the manuscript. The whole article is hardly readable and understandable.

4. The title is too broad and lacks focus.

Author Response

Reviewer 3

Thank you for your general analysis of our article presented in Marine Drugs, which we called Coral Lipids.

On your recommendation and in agreement with the Marine Drugs office, we have removed more than 40 Tables from the main text in the Supporting material, and now the article is easy to read.

We checked Table 7, which you mentioned, and link 53. Yes, the link is correct, we confirm it.

We have removed redundant material from the review. All changes and additions are highlighted in blue.

We left the name Coral Lipids unchanged, as both simple and complex coral lipids are presented in the flock. Three years earlier, we had already published an article about coral steroids, and this article is a legitimate continuation of this topic.

In addition, at our request, our colleagues from the Department of English Philology worked on the article, and therefore the quality of the article became very good.

Round 2

Reviewer 3 Report

this edition is well written